# Taming the Recent-Data Bias:
# Towards Robust Time Series Forecasting with Global Context

Longlong Xu [1]    Zeyan Li [2]    Xiao He [2]    Zhaoyang Yu [2]
Changhua Pei [3]    Zhe Xie [1]    Zijun Dou [1]    Tieying Zhang [2]    Dan Pei [1]

## Abstract

Time series forecasting plays a vital role in numerous domains. However, real-world time series are frequently contaminated by noise, missing values, and anomalies, posing significant challenges to reliable forecasting. In this work, we first systematically investigate a fundamental limitation prevalent in existing forecasting methods: an excessive reliance on the most recent observations—termed "recent-data bias". This bias renders forecasts highly vulnerable to perturbations in recent data, severely undermining prediction reliability. To address this issue, we propose TameR, a novel approach for robust time series forecasting that effectively mitigates recent-data bias via enhancing the utilization of global context. Specifically, it employs a basis-aligned randomized sampling strategy to reduce dependence on any specific recent data. Furthermore, TameR incorporates a learnable periodicity extraction module coupled with a two-stage learning protocol to robustly separate periodic patterns from the sampled residual components. Comprehensive experiments demonstrate that TameR significantly outperforms state-of-the-art methods in robustness against diverse perturbation scenarios, while achieving comparable accuracy on clean data. Code is available at https://github.com/NetManAIOps/TameR.

## 1. Introduction

Time series data are prevalent across domains including economics, energy, environment, traffic, and AIOps (Wen et al., 2022; Zhao et al., 2023; Qiu et al., 2025). In these areas, time series forecasting (TSF), which uses historical data to predict future values, plays a crucial role (Qiu et al., 2024). Accurate forecasting depend on effectively modeling temporal dependencies, with recent data being particularly significant. This principle is reflected in classical methods like exponential smoothing (Holt, 1957) and modern deep learning models such as LogTrans (Li et al., 2019), both of which inherently emphasize recent observations.

However, in real-world scenarios, time series data are often contaminated with noise, missing values, and anomalies (Cheng et al., 2024; Cini et al., 2021). This raises an important question: *how robust are existing TSF methods when recent data are perturbed?*

Fig. 1 illustrates predictions of a simple *single-layer linear model* on ETTh2 (Zhou et al., 2021). The input dimension of this model corresponds to the historical horizon, while the output dimension corresponds to the forecasting horizon. When a real-world anomaly occurs in recent data, predictions diverge significantly from the ground truth (Fig. 1a). Moreover, while the model predicts accurately on clean data (Fig. 1b), it performs poorly when an out-of-distribution perturbation is applied to the last historical data point (Fig. 1c). Fig. 1d shows the normalized weight matrix of the model, where the absolute values indicate the relative contributions to the forecast. **Vertical high-weight bands** are concentrated in the rightmost columns, indicating that recent data, particularly the last data point, exert a greater influence on future predictions than others. This phenomenon presents an overly **"recent-data bias"**, an excessive reliance on recent data that leaves TSF fragile to perturbations there.

Motivated by this observation, we systematically investigate the recent-data bias of more existing TSF methods with various architectures, particularly popular deep learning-based methods (see Sec. 3.1). Utilizing a gradient-based input importance scoring method to quantify the influence of each input point (Appendix D), our empirical study reveals that existing methods tend to rely excessively on the most recent observations. It can make them highly susceptible to perturbations in recent data, thus compromising prediction reliability. This insight motivates us to mitigate the recent-

---

[1]Department of Computer Science and Technology, Tsinghua University, Beijing, China [2]ByteDance, Beijing, China [3]Computer Network Information Center, Chinese Academy of Sciences, Beijing, China. Correspondence to: Tieying Zhang <tieying.zhang@bytedance.com>.

*Proceedings of the 43$^{rd}$ International Conference on Machine Learning*, Seoul, South Korea. PMLR 306, 2026. Copyright 2026 by the author(s).

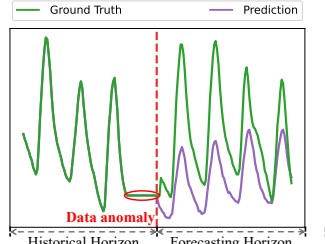 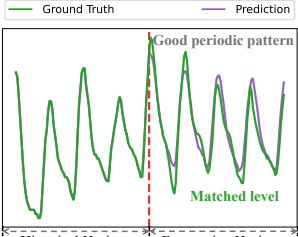 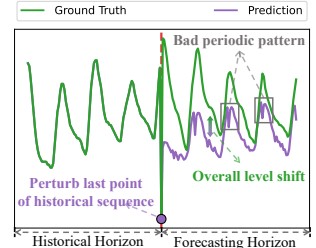 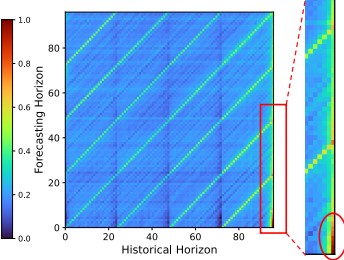

*(a)* Forecast with real anomaly.    *(b)* Forecast w/o perturbations.    *(c)* Forecast with perturbations.    *(d)* Model's weight matrix.

*Figure 1.* Motivational cases of a *single-layer linear model* on ETTh2 are analyzed. Fig. 1a shows a real-world anomaly in ETTh2. The purple point in Fig. 1c represents an out-of-distribution perturbation injected in clean data (Fig. 1b), resulting in poor forecasting. The normalized weight matrix in Fig. 1d shows high weights in the right columns, indicating these data points strongly affect forecasting.

data bias and develop a TSF method more robust against **perturbations in recent data**.

Existing study on robustness of TSF commonly utilize seasonal-trend decomposition (Wen et al., 2019; Zhou et al., 2022b; He et al., 2023). However, these methods do not fully address perturbations in recent data, as such perturbations can still persist in the seasonal component, the trend component, or both. Furthermore, some studies (Liu et al., 2022; 2023b) focus on enhancing robustness in non-stationary time series, which may introduce more reliance on recent data. Approaches (Connor et al., 1994; Cheng et al., 2024) centered on TSF with anomalies (TSFA) are relevant to robustness against perturbations. Classic TSFA strategies generally follow a detection-imputation-retraining pipeline (Connor et al., 1994). Nevertheless, its effectiveness is constrained by the cumulative errors introduced at each step (detection, imputation, and retraining). Overall, existing approaches for TSF commonly either lack robustness against perturbations in recent data or compromise forecasting accuracy. Achieving enhanced robustness to such perturbations while maintaining accuracy remains a significant challenge.

In this paper, we aim to develop a robust TSF method that is resistant to perturbations in recent data while preserving overall forecasting accuracy. The key idea is to *comprehensively exploit the global context* rather than relying heavily on recent data. Specifically, we propose to randomly sample time points from the historical input sequence during model training. This strategy compels the model to learn predictive patterns from arbitrary, potentially non-contiguous combinations of sampled time points, thereby enhancing the utilization of global context. However, developing such a sampling-based approach for robust and accurate forecasting faces two challenges:

**(1) Irregular representation.** Random sampling makes each point in time series correspond to irregular time intervals, so treating the sampled series as a vector misrepresents temporal structure. Imputation or padding introduces artificial values, distorting the original data distribution.

**(2) Information loss.** Random sampling can inadvertently remove key temporal information that is crucial for accurate forecasting. Important patterns or dependencies may be omitted if the sampled points fail to capture critical segments of the sequence, leading to incomplete context for the model and ultimately degrading forecasting accuracy.

To address challenge 1, we propose **B**asis-**A**ligned **R**andomized **S**ampling (**BARS**), which represents the randomly sampled time series in a predetermined basis function domain. Unlike direct vectorization or imputation in the time domain, BARS projects any subset of sampled time points onto a fixed set of trigonometric basis functions spanning various frequencies. Therefore, BARS provides a unified and consistent representation for irregularly sampled time series.

To address challenge 2, recognizing that periodicity is crucial for accurate time series forecasting (Wu et al., 2023; Lin et al., 2024), we incorporate a **L**earnable **P**eriodicity **E**xtraction (**LPE**) module which is inherently robust to perturbations. This module extracts the periodic component from historical data before sampling, thus preserving the underlying periodicity. Sampling is necessary only in the decycled residual space. The ultimate forecast combines periodicity forecasting through LPE and decycled residual forecasting using BARS. To improve the accuracy and stability of both periodicity and residual learning, we further propose a two-stage training protocol.

Ultimately, we propose **TameR**, a robust TSF approach designed to **Tame R**ecent-data bias via enhancing the utilization of global context. By integrating basis-aligned randomized sampling, learnable periodicity extraction, and two-stage training, TameR performs accurate and robust forecasting. The contributions are summarized as follows:

- To the best of our knowledge, we are the **first** to systematically investigate the "recent-data bias" of mainstream TSF methods.
- We introduce TameR, a sampling-based TSF method which is robust to perturbations in recent data. A BARS

strategy is designed to achieve robust and consistent representation of sampled time series.

- Extensive experiments on 8 datasets spanning 5 domains demonstrate that TameR performs accurate forecasting while remaining robust against diverse perturbations.

## 2. Related Work

**Robust time series forecasting** is focused on developing models that can maintain accurate predictions despite data anomalies or inherent variability. Seasonal-trend decomposition is well-established to enhance forecast against natural fluctuations in time series patterns. Deep learning models (Oreshkin et al., 2020; Zhou et al., 2022b; Zeng et al., 2023) have successfully integrated these decomposition techniques, whereas some methods (Wen et al., 2019) specifically target robust seasonal-trend decomposition. Additionally, some methods aim to robustly capture correlations among multiple variables for TSF, mainly employing graph neural networks (Shang et al., 2021; Yu et al., 2022) or domain adaptation (DARF (Cheng et al., 2023)). Furthermore, many studies, such as Non-stationary Transformer (Liu et al., 2022) and Koopa (Liu et al., 2023b), are dedicated to processing non-stationarity for robust forecasting. Time series forecasting with anomalies (TSFA) closely resembles perturbations we investigate. Classic TSFA methods employs a detection-imputation-retraining pipeline (Connor et al., 1994; Bohlke-Schneider et al., 2020), while RobustTSF (Cheng et al., 2024) provides a comprehensive theoretical analysis of how anomalies affect time series forecasting. Similarly to our work, an existing study (Yoon et al., 2022) examines the robustness to input perturbations by introducing a randomized smoothing technique, primarily from an adversarial perspective over *entire* time series. Conversely, our work focuses on enhancing robustness against perturbations in recent data, a common occurrence in real-world time series due to factors such as measurement noise, missing values, and anomalies.

## 3. Recent-Data Bias

### 3.1. Empirical Study

In this section, we systematically explore whether existing complex TSF approaches with diverse architectures exhibit recent-data bias. Unlike the *single-layer linear model* which can offer an interpretable weight matrix, we propose a gradient-based point importance scoring method (Appendix D) for complex TSF approaches. Specifically, we evaluate the importance score distribution of four representative architectures (see Appendix A) on ETTh2 (Zhou et al., 2021) and ETTm2 (Zhou et al., 2021), containing PatchTST (Nie et al., 2023) (attention-based, time domain), TimeMixer (Wang et al., 2024) (linear-based, time domain),

FiLM (Zhou et al., 2022a) (attention-based, mathematics domain), and FreTS (Yi et al., 2023) (linear-based, frequency domain). Using these models, we aim to cover key design paradigms in TSF, using linear and attention-based architectures across time, frequency, and mathematics domains.

Fig. 2 presents the resulting importance score distribution. Across all models and datasets, importance scores progressively rise for the recent few historical data points, with the highest scores typically assigned to the last point. Current TSF methods do not effectively leverage global context, instead show a significant recent-data bias, regardless of any specific input time series. Although recent data are theoretically beneficial for maintaining continuity and accuracy in forecasting, assigning excessive importance may reduce model robustness against perturbations in recent data.

### 3.2. Theoretical Analysis

Assume time series satisfies vector autoregressive process of order 1, *i.e.*, VAR(1), we conduct a theoretical analysis of the optimization process to explain the recent-data bias (Proposition 3.1). For simplicity, we focus on linear predictors here. However, recent-data bias also applies to non-linear deep models, because the core insight of theoretical analysis relies on the intrinsic decay of temporal correlations in stationary time series, thus affecting the gradient flow in model optimization.

**Proposition 3.1** (Recent-Data Bias). *Minimizing MSE on VAR(1) leads to an exponential decay in gradient magnitudes with increasing time lag, making the model assign greater weights to recent data relative to distant ones.*

*Proof.* Consider a multivariate time series $\{\mathbf{x}_t\}$ generated by $\mathbf{x}_t = \mathbf{A}\mathbf{x}_{t-1} + \boldsymbol{\epsilon}_t$, where the spectral radius $\rho(\mathbf{A}) < 1$ ensures stationarity. We define a linear forecaster $\mathcal{F}$ with a lookback window $H$: $\hat{\mathbf{x}}_{t+1} = \sum_{k=0}^{H-1} \mathbf{W}_k \mathbf{x}_{t-k}$, where $\mathbf{W}_k$ represents the learnable weight matrix for the input at lag $k$ (*i.e.*, $\mathbf{x}_{t-k}$). The model parameters $\boldsymbol{\theta} = \{\mathbf{W}_0, \ldots, \mathbf{W}_{H-1}\}$ are optimized by minimizing a loss function $\mathcal{L}$, typically MSE ($L_2$ norm).

During training via gradient descent, the update to the weight $\mathbf{W}_k$ at step $i$ is proportional to the expected gradient of the loss function. For MSE loss $\mathcal{L} = \|\mathbf{x}_{t+1} - \hat{\mathbf{x}}_{t+1}\|_2^2$, the gradient with respect to the $k$-th lag component is:

$$\nabla_{\mathbf{W}_k}\mathcal{L} = -2(\mathbf{x}_{t+1} - \hat{\mathbf{x}}_{t+1})\mathbf{x}_{t-k}^\top. \qquad (1)$$

Assuming initialization close to zero ($\hat{\mathbf{x}} \approx \mathbf{0}$), the initial learning signal is dominated by the correlation between the target $\mathbf{x}_{t+1}$ and the historical input $\mathbf{x}_{t-k}$:

$$\mathbb{E}[\nabla_{\mathbf{W}_k}\mathcal{L}] \propto -\mathbb{E}[\mathbf{x}_{t+1}\mathbf{x}_{t-k}^\top] = -\mathbf{\Gamma}(k+1). \qquad (2)$$

**Exponential Weight Decay.** For a VAR(1) process, the

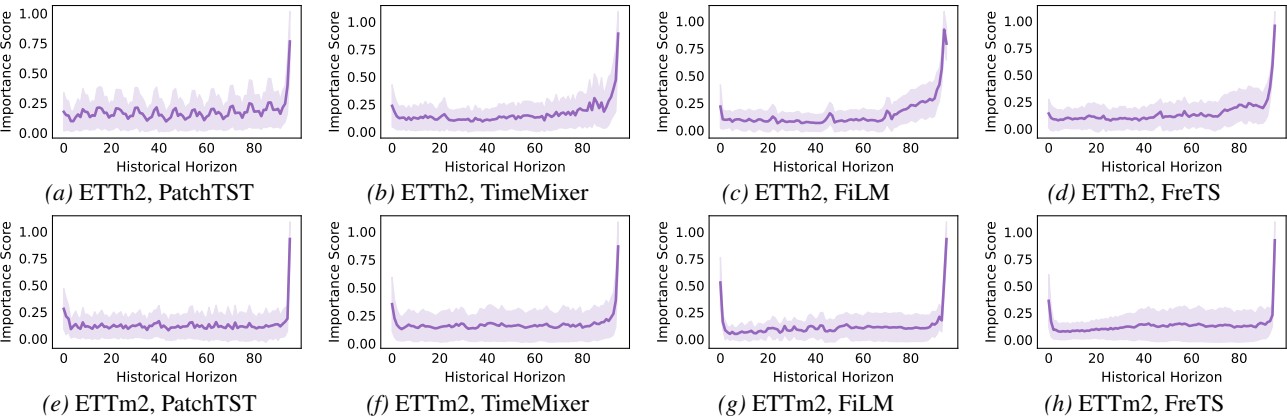

*Figure 2.* Importance score distribution of historical points on ETTh2 and ETTm2 across different model architectures. The central line indicates mean of scores ($\mu(t)$), and the shaded band refers to the 95% confidence interval ($\text{CI}_{95\%}(t)$).

auto-covariance at lag $k$ is given by $\mathbf{\Gamma}(k) = \mathbf{A}^k \mathbf{\Gamma}(0)$. Taking the norm, we have:

$$\|\mathbf{\Gamma}(k)\| \leq \|\mathbf{A}^k\| \|\mathbf{\Gamma}(0)\|. \tag{3}$$

Given that the upper bound of $\|\mathbf{A}^k\|$ increases exponentially with $k$ (detailed proof see Appendix C.1), when this is applied to Eq. 3 and Eq. 2, the gradient magnitude $\|\nabla_{\mathbf{W}_k} \mathcal{L}\|$ exponentially decreases as the lag $k$ increases. This leads the model to update weights $\mathbf{W}_0$, which correspond to the most recent data $\mathbf{x}_t$, with significantly larger steps than weights related to more distant data. This creates a strong inductive bias during optimization, guiding the model towards solutions that primarily rely on recent data. Even as the model converges, this bias often results in a predictor where distant historical data is under-utilized, as the error signals from long-range dependencies are overshadowed by the stronger gradients from recent data. $\qquad \square$

## 4. Problem Definition

In TSF, the primary objective is to predict future values based on historical time series data. Specifically, given a historical time series of $H$ time points $\mathbf{X} = \{\mathbf{x}_{t-H+1}, \mathbf{x}_{t-H+2}, \dots, \mathbf{x}_t\} \in \mathbb{R}^{H \times C}$, where $\mathbf{x}_t = [x_t^{(1)}, x_t^{(2)}, \dots, x_t^{(C)}]^\top$ is the observation vector at time $t$, and $C$ is the number of variables (or channels), the forecasting task aims to predict the future $F$-step time series $\hat{\mathbf{Y}} = \{\hat{\mathbf{x}}_{t+1}, \hat{\mathbf{x}}_{t+2}, \dots, \hat{\mathbf{x}}_{t+F}\} \in \mathbb{R}^{F \times C}$.

However, existing TSF approaches exhibit a recent-data bias, which undermines their robustness against perturbations in recent data. Given that real-world time series data frequently contain noise, missing values, and anomalies, achieving robustness against such perturbations is crucial for practical applications. Therefore, our aim is to develop a TSF method that maintains robustness amidst these challenges while preserving overall forecasting accuracy. To achieve this goal, it is crucial to prioritize global context

over recent data, thereby ensuring a relatively balanced importance distribution across various historical data points.

## 5. Methodology

Based on the idea of "comprehensively exploiting the global context", we propose TameR (Fig. 3), a robust TSF approach for taming recent-data bias. Firstly, we introduce BARS to dynamically utilize the global context window. Next, we present LPE to facilitate the robust decomposition of periodicity and residuals. On this basis, we design a two-stage training technique to enhance the accuracy and stability of dual-space learning (periodic and residual space).

### 5.1. BARS

To dynamically leverage the entire context window, we propose Basis-Aligned Randomized Sampling (**BARS**). The core insight is to stochastically sample data points within the historical sequence during training, compelling the model to learn predictions based on randomly selected points. Consequently, during inference, the trained model can effectively utilize all available time points without relying heavily on specific points. Unlike standard dropout (Baldi & Sadowski, 2013), BARS operates directly on raw temporal data instead of hidden features, and is coupled with basis projection to keep irregularly sampled data points in a fixed representation space. From an optimization perspective, we theoretically prove that such a sampling strategy attenuates the influence of each specific data point, effectively mitigating the recent-data bias (see Proposition C.3 in Appendix C.2). The complete process of BARS is detailed in Appendix B.1.

As illustrated in the left panel of Fig. 3, BARS operates through random data point sampling governed by a hyper-parameter sampling rate $\rho$. Given a historical time series $\mathbf{X} \in \mathbb{R}^{H \times C}$ spanning $H$ data points with $C$ channels, the sampled subsequence contains $\lfloor \rho H \rfloor$ points while preserv-

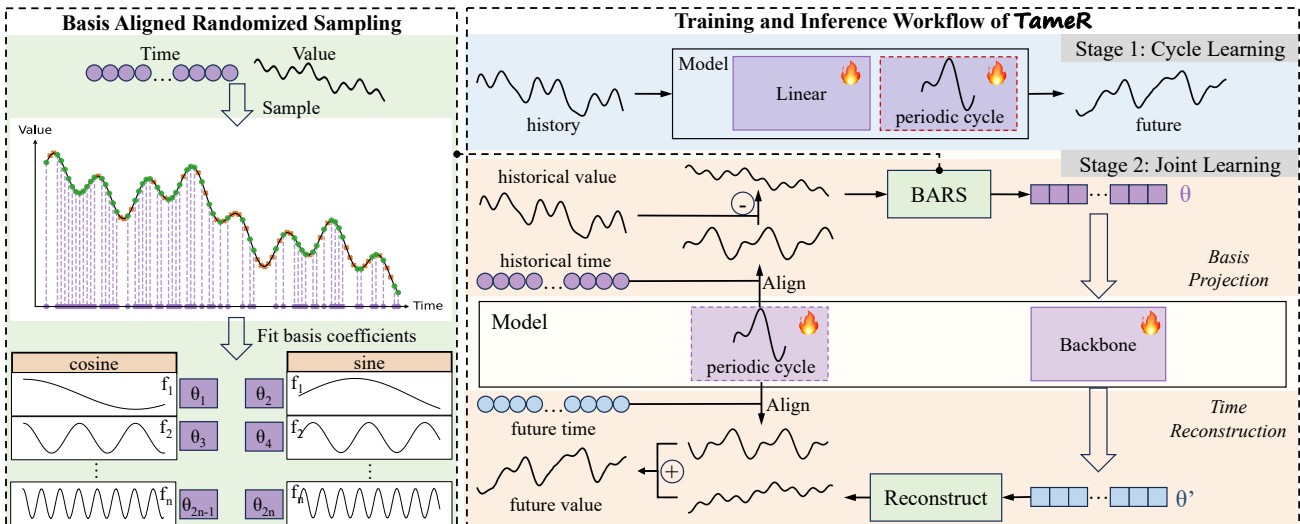

*Figure 3.* The overall TameR forecasting approach (Sec. 5.3). The left part is the BARS strategy (Sec. 5.1), which performs random time point sampling and transforms sampled time series into basis domain. The right part is the training and forecasting workflow incorporating BARS, Learnable Periodicity Extraction (Sec. 5.2) and 2-Stage Training (Sec. 5.4).

ing the channel dimension. During each training iteration, the random omission of temporal points prevents the model from over-relying on any particular points.

Traditional techniques (Connor et al., 1994) for processing unsampled data typically involve imputation in the time domain, such as zero-padding, forward/backward filling, or linear interpolation. However, these imputation techniques can introduce artificial biases, thereby distorting predictions. A well-established mathematical principle (Itō, 1993) demonstrates that any continuous function can be represented as a linear combination of basis functions, similar to vector decomposition in linear algebra. Inspired by this principle, we treat time series as a continuous function which maps temporal indices to observed values. Through function fitting, we project irregularly sampled time series onto a meticulously constructed basis function domain to avoid the need for imputation in the time domain. Furthermore, we theoretically demonstrate that this basis projection disperses local perturbation across global coefficients, thereby enhancing robustness (Proposition C.4 in Appendix C.2).

Specifically, considering the inherent characteristics of time series, we select Fourier trigonometric functions to capture temporal patterns. We define a set of 179 frequencies $\{f_i\}_{i=1}^{179}$ corresponding to minutely, hourly, daily, and weekly periodicity. Further implementation details are available in Appendix B.1, and the choice of frequencies is provided in Appendix I. Each frequency contributes two orthogonal basis functions (sine and cosine), yielding 358 functions that capture both short-term and long-term temporal dynamics. For data with different sampling intervals, the sampled indices are first converted to physical elapsed time in the same unit as the selected periods before evaluat-

ing the basis functions. For a given set of sampled indices and corresponding observed values, we solve for the basis coefficients $\theta$ via regularized least-squares:

$$\min_{\theta} \|\Phi\theta - \mathbf{X}_{\text{sampled}}\|_2^2 + \lambda\|\theta\|_2^2$$

where $\Phi$ denotes the basis matrix evaluated at the sampled indices $\tau$, and $\lambda$ is a regularization parameter to ensure numerical stability by mitigating ill-conditioned systems.

## 5.2. Learnable Periodicity Extraction

Random sampling can compromise key temporal information, leading to reduced forecasting accuracy. Recent studies (Wang et al., 2024; Wu et al., 2023; Lin et al., 2024) have thoroughly investigated periodicity modeling in time series forecasting, underscoring its crucial importance in enhancing forecasting accuracy. In response, we propose to extract periodicity from raw time series data before sampling, thus avoiding information loss in periodicity modeling.

Classic moving average (MOV) uses sliding windows for trend-periodicity decomposition, which suffer from two issues (Li et al., 2023; Lin et al., 2024): (1) Padding causes edge distortion, leading to forecasting artifacts. (2) It struggles to separate periodic patterns from perturbations.

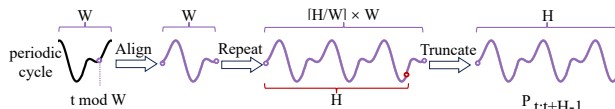

*Figure 4.* The LPE module. It extracts the periodic component with timestamp $t$ and length $H$ based on the periodic cycle.

To ensure the robustness of periodicity extraction against

perturbations, we propose a learnable periodicity extraction (LPE) module inspired by CycleNet (Lin et al., 2024). Specifically, for each channel in a dataset, we initialize a learnable periodic cycle whose length $W$ corresponds to the dataset's inherent periodicity (*e.g.*, 24 hours for daily cycles). This cycle is then optimized during training. In particular, for domains such as economics or finance that lack clear periodicity, $W$ is set to $W \geq H$ to capture patterns across broader contexts. The learnable cycle for all channels is parameterized as a per-channel table defined in $\mathbb{R}^{W \times C}$. Fig. 4 illustrates the workflow of the LPE module, which consists of three distinct processes. Firstly, the alignment process adjusts the cycle's starting point to correspond with the input timestamp. Secondly, the repetition process replicates the cycle to ensure it spans the entire historical and forecasting horizon. Lastly, the truncation process removes any cycle portions that exceed the necessary length.

### 5.3. Overall Forecasting Approach

By combining the LPE and the BARS module, We present a unified forecasting approach called TameR (Fig. 3). The LPE module generates the future periodic component, while BARS operates in the residual space. Consequently, TameR executes TSF through a dual-space learning mechanism.

Given an historical input sequence $\mathbf{X} \in \mathbb{R}^{H \times C}$, historical timestamps $\mathbf{T}_{\text{hist}}$ and future timestamps $\mathbf{T}_{\text{future}}$, the LPE module first generates corresponding periodic components:

$$\mathbf{P}_{\text{hist}} = \text{LPE}(\mathbf{T}_{\text{hist}}), \ \mathbf{P}_{\text{future}} = \text{LPE}(\mathbf{T}_{\text{future}})$$

The historical residual component is then computed as $\mathbf{R}_{\text{hist}} = \mathbf{X} - \mathbf{P}_{\text{hist}}$. This residual sequence serves as input to the BARS module, and is transformed to a basis coefficient representation, *i.e.*, $\theta_{\text{hist}} = \text{BARS}(\mathbf{R}_{\text{hist}}, \rho, \lambda)$, where $\theta_{\text{hist}} \in \mathbb{R}^{2K \times C}$ represents the basis coefficients capturing the non-periodic residual patterns. Here, random sampling is utilized during the training phase, whereas data points are not sampled during the inference phase for a global temporal information.

A deep learning backbone model $\mathcal{M}$ (*e.g.*, Transformer, CNN, or MLP) then processes these coefficients to predict future residual coefficients, *i.e.*, $\hat{\theta}_{\text{future}} = \mathcal{M}(\theta_{\text{hist}})$. Here, each channel utilizes a shared backbone for modeling through parameter sharing, also known as the Channel Independent strategy (Han et al., 2024). The future residual sequence is reconstructed using basis function, *i.e.*, $\hat{\mathbf{R}}_{\text{future}} = \Phi_{\text{future}}\hat{\theta}_{\text{future}}$, where $\Phi_{\text{future}}$ is the basis matrix evaluated at future timestamps. The final forecast combines the predicted residuals with the periodic component ($\hat{\mathbf{Y}} = \hat{\mathbf{R}}_{\text{future}} + \mathbf{P}_{\text{future}}$).

Note that we deliberately avoid directly reconstructing future residuals from $\theta_{\text{hist}}$ for two primary reasons. Firstly, the basis coefficients $\theta_{\text{hist}}$ are tailored to fit the historical

residuals uniquely, thereby leading to overfitting to past data patterns. Secondly, real-world time series exhibit fluctuations that do not adhere to strict mathematical function relationships, hindering the straightforward extrapolation of coefficients into future values.

### 5.4. 2-Stage Training

The incorporation of learnable periodic cycle, random sampling and basis function transformation in TameR introduces additional complexity to the learning process. To ensure the accuracy and stability of dual-space learning, we develop a meticulously crafted two-stage training protocol (detailed in Appendix B.2).

**Stage 1: Periodic Cycle Learning.** As illustrated in the top-right panel of Fig. 3, the initial stage is designed to learn accurate periodic cycles using a simple architecture. We replace the BARS module in TameR with a single-layer linear model that maps historical residuals directly to future residuals. The future periodic component keeps to be predicted using the LPE module. The straightforward architecture of this stage, operating directly in the time domain without intricate transformations, forces the LPE module to accurately learn periodic cycles.

**Stage 2: Joint space Learning.** As depicted in the lower-right panel of Fig. 3, this stage incorporates the periodic cycles learned in Stage 1 into TameR by initializing the LPE module using the learned cycle parameters. Then, we train TameR which contains the LPE module for periodicity space learning and the BARS module for residual space learning. All parameters are jointly optimized during this stage, enabling fine-tuning of the periodic cycle to complement the basis domain.

**Loss Function.** Building upon Proposition 3.1, and inspired by RobustTSF (Cheng et al., 2024), which demonstrated that Mean Absolute Error (MAE) offers improved robustness against anomalies over Mean Squared Error (MSE), we employ MAE loss across both stages.

## 6. Evaluation

We comprehensively evaluate TameR utilizing 8 widely-used datasets across 5 key domains. Our analysis involves a comparison of TameR with baseline models across various architectures and representation domains, employing two well-established metrics: MSE and MAE. We use a *dual-layer MLP* with ReLU activation as the backbone, $\mathcal{M}$. Comprehensive datasets, baselines, evaluation metrics and implementation details are provided in Appendix E. Due to space limitations, parameter sensitivity analysis and efficiency analysis are presented in Appendix H and J respectively.

*Table 1.* Robustness under perturbations (recent, single anomalous point). Results are averaged from prediction lengths $F \in \{96, 192, 336, 720\}$. **1$^{st}$ Count** is the total number of times a model achieved the best performance of two metrics across all datasets and prediction lengths. **Avg Rank** is a model's average rank of two metrics across all datasets and prediction lengths. **Bold** indicates best result, underlined second best per dataset and metric.

| Models | ETTh1 | | ETTh2 | | ETTm1 | | ETTm2 | | Weather | | Exchange | | Traffic | | Solar | | 1$^{st}$ Count | Avg Rank |
|---|---|---|---|---|---|---|---|---|---|---|---|---|---|---|---|---|---|---|
| | MSE | MAE | MSE | MAE | MSE | MAE | MSE | MAE | MSE | MAE | MSE | MAE | MSE | MAE | MSE | MAE | | |
| DLinear | 0.566 | 0.493 | 0.448 | 0.451 | 0.623 | 0.477 | 0.331 | 0.366 | 0.371 | 0.345 | 0.373 | 0.434 | 0.902 | 0.422 | 3.175 | 0.851 | 5 | 5.38 |
| TimeMixer | 0.552 | 0.486 | 0.407 | 0.419 | 0.534 | 0.452 | 0.318 | 0.352 | 0.298 | 0.311 | 0.439 | 0.465 | 0.568 | 0.346 | 0.569 | 0.449 | 0 | 3.46 |
| CycleNet | 0.565 | 0.483 | 0.434 | 0.436 | 0.619 | 0.479 | 0.320 | 0.353 | 0.349 | 0.331 | 0.429 | 0.452 | 0.588 | 0.340 | 0.664 | 0.405 | 0 | 4.25 |
| Crossformer | 0.524 | 0.503 | 0.980 | 0.690 | 0.514 | 0.467 | 0.571 | 0.496 | 0.274 | 0.303 | 0.612 | 0.591 | 0.593 | **0.289** | **0.327** | **0.288** | 8 | 4.13 |
| PatchTST | 0.554 | 0.486 | 0.400 | 0.415 | 0.645 | 0.491 | 0.346 | 0.369 | 0.306 | 0.316 | 0.468 | 0.485 | 0.601 | 0.353 | 0.725 | 0.507 | 0 | 4.39 |
| iTransformer | 0.500 | 0.470 | 0.404 | 0.418 | 0.575 | 0.475 | 0.327 | 0.359 | 0.317 | 0.322 | 0.410 | 0.445 | 0.583 | 0.365 | 0.754 | 0.470 | 1 | 3.68 |
| Leddam | 0.505 | 0.467 | 0.405 | 0.418 | 0.585 | 0.471 | 0.350 | 0.372 | 0.340 | 0.347 | 0.441 | 0.461 | 0.625 | 0.351 | 2.340 | 0.761 | 0 | 4.70 |
| MICN | 0.474 | 0.467 | 0.474 | 0.459 | 0.484 | 0.441 | 0.309 | 0.360 | 0.295 | 0.313 | **0.325** | **0.398** | 0.623 | 0.318 | 0.359 | 0.345 | 6 | 2.51 |
| TimesNet | 0.472 | 0.453 | 0.396 | 0.409 | 0.418 | 0.411 | 0.296 | 0.331 | **0.262** | **0.284** | 0.435 | 0.448 | 0.700 | 0.360 | 0.382 | 0.354 | 9 | 2.20 |
| FEDformer | 0.467 | 0.459 | 0.408 | 0.426 | 0.789 | 0.591 | 0.348 | 0.388 | 0.314 | 0.346 | 0.530 | 0.497 | 0.666 | 0.389 | 0.657 | 0.607 | 3 | 4.97 |
| FreTS | 0.677 | 0.532 | 0.576 | 0.492 | 0.745 | 0.519 | 0.404 | 0.404 | 0.310 | 0.336 | 0.423 | 0.456 | 0.792 | 0.405 | 1.369 | 0.586 | 0 | 5.93 |
| TimeKAN | 0.501 | 0.461 | 0.404 | 0.418 | 0.521 | 0.450 | 0.323 | 0.357 | 0.286 | 0.302 | 0.451 | 0.467 | 0.678 | 0.379 | 0.803 | 0.482 | 1 | 3.48 |
| FiLM | 0.492 | 0.453 | 0.389 | 0.406 | 0.511 | 0.438 | 0.315 | 0.348 | 0.303 | 0.305 | 0.380 | 0.421 | 1.281 | 0.708 | 1.173 | 0.629 | 0 | 3.24 |
| RobustTSF | 0.608 | 0.510 | 0.503 | 0.483 | 0.739 | 0.509 | 0.362 | 0.389 | 0.471 | 0.382 | 0.411 | 0.435 | 0.926 | 0.450 | 2.071 | 0.761 | 0 | 6.36 |
| TameR | **0.461** | **0.442** | **0.379** | **0.401** | **0.416** | **0.404** | **0.278** | **0.319** | 0.263 | 0.286 | 0.421 | 0.460 | **0.550** | 0.319 | 0.340 | 0.308 | **31** | **1.33** |

**Perturbation Scenarios.** To thoroughly evaluate the robustness of TameR against diverse real-world data perturbation scenarios, we include perturbations from 2 temporal positions (*recent* and *random*) and 4 perturbation types (*single anomalous point*, *continuous anomalous sequence*, *single missing point*, and *multiple anomalous points*). Perturbations are independently and randomly introduced into each test sample. We refer to specific scenarios as, for example, "perturbations (recent, single anomalous point)" in the following sections. More details on perturbation scenarios can be found in Appendix E.3.

### 6.1. Robustness under Perturbations in Recent Data

In this section, we analyze the results of perturbations in recent data. Tbl. 1 presents experimental results of perturbations (recent, single anomalous point). Due to space constraints, results for perturbations of continuous anomalous sequence and a single missing point are detailed in Appendix F. Across all three recent data perturbation scenarios, TameR exhibits a substantial performance advantage. In particular, most existing baselines perform poorly even when the anomaly is limited to the last data point (Tbl. 1). Conversely, TameR remains robust under such condition, achieving an average rank of **1.33** and a 1$^{st}$ Count of **31** across all datasets and prediction lengths. Note that TameR consistently achieves either the best or second-best robustness performance across all datasets, except for the Exchange dataset. As illustrated by Tbl. 3 in Appendix E.1, Exchange exhibits a pronounced shifting characteristic. Since TameR relies partially on reconstructing basis functions for predictions, its robustness slightly decreases when the time series demonstrates significant shifts.

**Temporal importance of TameR.** To further analyze the temporal importance (see Sec. 3.1) of TameR, we present the impact of different data points on forecasting performance, using TameR with a historical sequence length $H = 96$ and prediction length $F = 96$ across four ETT datasets, as depicted in Fig. 5. Compared to other models (Fig. 2), TameR exhibits a relatively uniform distribution of importance across all historical points, with marginally higher weights assigned to the most recent points. This balanced temporal utilization allows for robust predictions that withstand local perturbations while effectively capturing recent trends. Overall, TameR shows enhanced robustness of TSF against perturbations in recent data.

### 6.2. Robustness under Different Perturbation Positions

In addition to perturbations in recent data, TameR consistently exhibits robust performance against perturbations at random positions, achieving the lowest average rank compared to baselines across the evaluated datasets (Detailed in Appendix F). To quantify the proportional rise in MSE after perturbations, we introduce $MSE \Uparrow$ which is calculated by $\frac{\text{MSE}_{\text{pert}} - \text{MSE}_{\text{clean}}}{\text{MSE}_{\text{clean}}} \times 100\%$, where $\text{MSE}_{\text{clean}}$ and $\text{MSE}_{\text{pert}}$ represent MSE on clean and perturbed data respectively. A higher $MSE \Uparrow$ indicates a greater decline in accuracy.

Fig. 6 presents the $MSE$ and $MSE \Uparrow$ across different perturbation scenarios on ETTm1. Existing baseline models show greater robustness to perturbations occurring at *random* temporal positions than with *recent* data. This observation suggests that most current TSF methods rely heavily on recent data within the historical window, rather than utilizing information from the global context uniformly. We

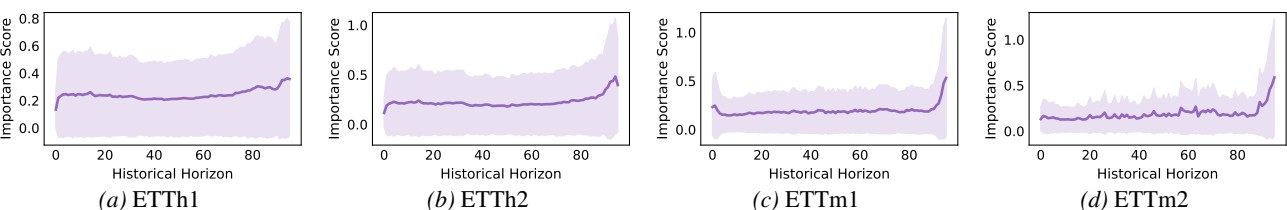

*Figure 5.* Importance score distribution of historical data points on four ETT datasets for TameR.

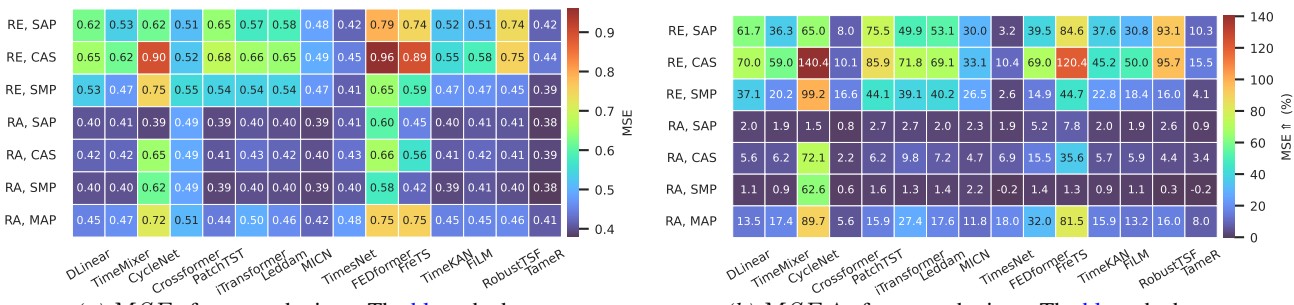

*(a)* $MSE$ after perturbations. The bluer the better. *(b)* $MSE \Uparrow$ after perturbations. The bluer the better.

*Figure 6.* Robustness evaluation under different perturbation scenarios on ETTm1. The Y-axis represents perturbation scenarios, denoted as (position, type). Perturbation positions: Recent (RE), Random (RA). Perturbation types: single anomalous point (SAP), continuous anomalous sequence (CAS), single missing point (SMP), multiple anomalous points (MAP). The top 3 rows represent perturbations in recent data, where current TSF methods demonstrate inferior performance compared to the bottom 4 rows, which perturb random points.

note that some methods (Crossformer, TimesNet) have a lower $MSE \Uparrow$ than TameR in some cases. However, their absolute MSE after perturbation is often higher, meaning TameR's predictions are actually more robust in the presence of perturbations. In extreme cases, a meaningless constant predictor achieves an $MSE \Uparrow$ of 0%.

### 6.3. Forecasting Accuracy on Unperturbed Data

Tbl. 2 illustrates the multivariate forecasting performance on unperturbed data, revealing TameR's competitive accuracy, comparable to SOTA baselines. TameR achieves the highest average rank and the most 1st Count across all datasets and prediction lengths. Due to its channel-independent mechanism, TameR performs slightly worse on datasets with greater correlation, such as Solar and Traffic, compared to lower-dimensional datasets like ETT series. The Exchange dataset, characterized by higher distribution shifts and significant non-stationarity, poses a challenge for the basis function reconstruction to maintain high accuracy. Notably, TameR, utilizing only a dual-layer MLP backbone, consistently outperforms more complex models, like TimesNet and TimeMixer, across most datasets. The trade-off between robustness to perturbations and accuracy on clean data is a common challenge. However, TameR demonstrates competitive accuracy relative to SOTA baselines, while significantly enhancing robustness to perturbations, which is our primary goal. To improve accuracy, increasing the sampling rate $\rho$ in BARS can be beneficial (see Appendix H).

### 6.4. Ablation Study

To evaluate the impact of each component in TameR, we conduct comprehensive ablation studies using five architectural variants: (1) **TameR w/o 2-Stage** utilizes only joint space learning, removing the periodic cycle learning stage; (2) **TameR w/o LPE** removes the LPE module; (3) **TameR Sample-0-Pad** substitutes basis transformation in BARS with zero-padding in the time domain; (4) **TameR Sample-Interp** replaces basis transformation in BARS with linear interpolation in the time domain; and (5) **TameR w/o Sample** eliminates time point sampling during training, thus utilizing all available time points. Experiments are conducted on 3 representative datasets with varying sampling frequency: ETTh1, ETTm1, and Weather. As presented in Fig. 7, we analyze both robustness ($MSE \Uparrow$) under perturbations (recent, single anomalous point) and forecasting accuracy (MSE) on unperturbed data.

Subfigure (a) of Fig. 7 illustrates the forecasting accuracy of different variants on unperturbed data. All variants, except TameR w/o Sample, exhibit an increase in MSE. This underscores the critical importance of the LPE module and two-stage training techniques for achieving optimal forecasting accuracy. The comparable or superior performance of TameR w/o Sample suggests that the approach of sampling and transforming into the basis domain successfully preserves predictive capability while enhancing robustness. It is noteworthy that time-domain sampling variants, particularly TameR Sample-Interp and TameR Sample-0-Pad, suffer significant accuracy degradation. In particular, the

*Table 2.* Results of TSF on unperturbed data (averaged from all prediction lengths of $F \in \{96, 192, 336, 720\}$).

| Models | ETTh1 | | ETTh2 | | ETTm1 | | ETTm2 | | Weather | | Exchange | | Traffic | | Solar | | 1st Count | Avg Rank |
|---|---|---|---|---|---|---|---|---|---|---|---|---|---|---|---|---|---|---|
| | MSE | MAE | MSE | MAE | MSE | MAE | MSE | MAE | MSE | MAE | MSE | MAE | MSE | MAE | MSE | MAE | | |
| DLinear | 0.443 | 0.433 | 0.452 | 0.446 | 0.396 | 0.387 | 0.286 | 0.330 | 0.271 | 0.290 | 0.318 | 0.387 | 0.665 | 0.352 | 0.328 | 0.308 | 0 | 4.62 |
| TimeMixer | 0.461 | 0.439 | 0.374 | 0.395 | 0.398 | 0.388 | 0.278 | 0.319 | 0.254 | 0.270 | 0.381 | 0.414 | 0.495 | 0.304 | 0.334 | 0.321 | 0 | 3.59 |
| CycleNet | **0.429** | **0.418** | 0.375 | 0.394 | 0.386 | 0.383 | 0.273 | 0.312 | 0.258 | 0.276 | 0.390 | 0.419 | 0.489 | 0.292 | 0.292 | 0.266 | 6 | 2.52 |
| Crossformer | 0.518 | 0.498 | 0.976 | 0.688 | 0.484 | 0.445 | 0.554 | 0.486 | 0.247 | 0.275 | 0.606 | 0.585 | 0.593 | 0.300 | 0.226 | **0.218** | 8 | 5.54 |
| PatchTST | 0.431 | 0.424 | **0.367** | **0.389** | 0.383 | 0.383 | 0.278 | 0.318 | 0.256 | 0.272 | 0.373 | 0.409 | 0.495 | 0.292 | 0.272 | 0.305 | 5 | 2.40 |
| iTransformer | 0.448 | 0.439 | 0.381 | 0.401 | 0.394 | 0.389 | 0.281 | 0.321 | 0.254 | 0.271 | 0.365 | 0.406 | **0.450** | 0.282 | 0.249 | 0.245 | 6 | 3.08 |
| Leddam | 0.437 | 0.425 | 0.371 | 0.391 | 0.390 | 0.383 | 0.277 | 0.317 | 0.243 | 0.272 | 0.362 | 0.403 | 0.491 | **0.278** | 0.220 | 0.256 | 7 | 2.07 |
| MICN | 0.435 | 0.441 | 0.463 | 0.450 | 0.381 | 0.388 | 0.285 | 0.337 | 0.248 | 0.274 | **0.309** | **0.379** | 0.619 | 0.311 | 0.283 | 0.279 | 10 | 3.76 |
| TimesNet | 0.490 | 0.464 | 0.417 | 0.419 | 0.406 | 0.403 | 0.289 | 0.325 | 0.256 | 0.278 | 0.430 | 0.444 | 0.679 | 0.348 | 0.364 | 0.336 | 0 | 5.80 |
| FEDformer | 0.439 | 0.446 | 0.417 | 0.431 | 0.569 | 0.520 | 0.341 | 0.383 | 0.306 | 0.339 | 0.529 | 0.497 | 0.649 | 0.376 | 0.548 | 0.549 | 0 | 6.65 |
| FreTS | 0.511 | 0.462 | 0.541 | 0.469 | 0.415 | 0.404 | 0.300 | 0.340 | **0.240** | **0.267** | 0.394 | 0.425 | 0.519 | 0.304 | 0.263 | 0.261 | 2 | 4.59 |
| TimeKAN | 0.445 | 0.427 | 0.378 | 0.397 | 0.388 | 0.384 | 0.281 | 0.321 | 0.249 | 0.269 | 0.394 | 0.422 | 0.633 | 0.349 | 0.321 | 0.291 | 1 | 3.66 |
| FiLM | 0.444 | 0.427 | 0.382 | 0.400 | 0.401 | 0.387 | 0.286 | 0.322 | 0.282 | 0.287 | 0.376 | 0.417 | 1.233 | 0.687 | 0.377 | 0.362 | 0 | 5.11 |
| RobustTSF | 0.446 | 0.438 | 0.435 | 0.438 | 0.395 | 0.390 | 0.284 | 0.332 | 0.270 | 0.293 | 0.374 | 0.396 | 0.639 | 0.359 | 0.342 | 0.343 | 3 | 4.80 |
| TameR | 0.432 | 0.423 | 0.369 | 0.392 | **0.379** | **0.379** | **0.267** | **0.309** | 0.248 | 0.269 | 0.358 | 0.403 | 0.537 | 0.308 | 0.243 | 0.244 | 16 | 1.82 |

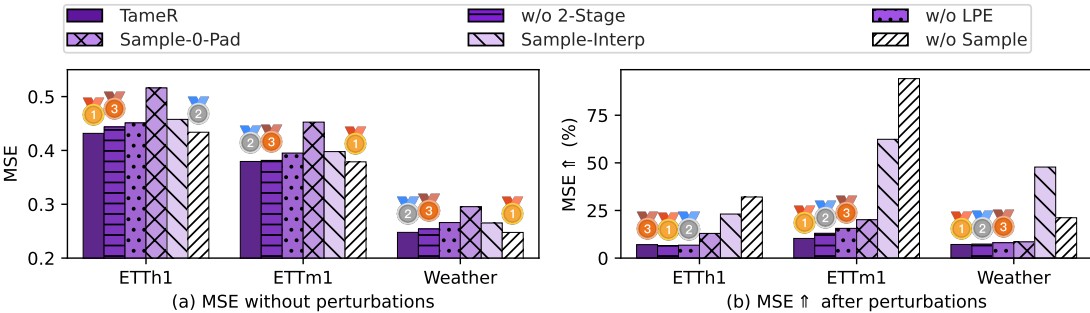

*Figure 7.* Ablation study on ETTh1, ETTm1 and Weather. $MSE \Uparrow$ is the proportional rise in MSE after perturbations. The results are averaged from all prediction lengths of $F \in \{96, 192, 336, 720\}$.

0-padding method performs worse due to its introduction of artificial values that disrupt temporal patterns.

Subfigure (b) of Fig. 7 evaluates robustness against perturbations in recent data. It shows that different variants exhibit unique patterns of increase in MSE. Although TameR w/o Sample demonstrates strong performance in unperturbed time series forecasting, its MSE significantly increases after perturbations. This highlights time point sampling as the critical role of perturbation robustness. TameR Sample-Interp also exhibits a high $MSE \Uparrow$, showing poor robustness due to preserved temporal patterns. By randomly filtering some time points, TameR Sample-0-Pad achieves a lower $MSE \Uparrow$ compared to the aforementioned linear interpolation variant. Notably, TameR, even without utilizing the LPE module or undergoing 2-stage training, maintains relatively low $MSE \Uparrow$, indicating that the BARS strategy significantly contributes to robustness.

Overall, the integration of random sampling, basis transformation, learnable periodicity extraction, and the two-stage training process allows TameR to effectively perform accurate forecasting that remains robust to perturbations.

## 7. Conclusion

In this paper, we systematically investigate the "recent-data bias" of mainstream TSF approaches. They show excessive reliance on recent data, thus hinders robustness against perturbation in these data. To address this issue, we introduce TameR, a robust time series forecasting approach to tame recent-data bias via enhancing the utilization of global context. TameR integrates a BARS module, which randomly samples data points during training, thereby ensuring the utilization of global context during inference. Through a LPE module, we prevent periodic components from perturbations and sampling. Overall, TameR forecasts in a dual space, *i.e.*, periodicity forecasting via the LPE and residual forecasting via BARS. Additionally, it employs a two-stage learning protocol to ensure the accuracy and stability of dual-space learning. Experiments across 8 datasets from 5 key domains validate TameR's high forecasting accuracy and robustness to diverse perturbations, with low computational costs and high training speed. Limitations and directions for future research are discussed in Appendix K.

## Acknowledgements

This work is supported by the National Key Research and Development Program of China (No.2024YFB4505903).

## Impact Statement

This work advances the field of machine learning, specifically in improving the robustness of time series forecasting methods. The primary goal is to address a methodological limitation (recent-data bias) to enhance model reliability in the presence of data perturbations. There are many potential societal consequences of our work, none of which we feel must be specifically highlighted here.

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

# A. Time Series Forecasting Approaches

Traditional statistical methods, such as ARIMA (Box & Pierce, 1970) and ETS (Hyndman et al., 2008), are highly effective in forecasting linear patterns in time series data but struggle with nonlinear dynamics. Machine learning models like XGBoost (Zhang et al., 2021) and Random Forests (Mei et al., 2014) have been extensively used to address nonlinear relationships and complex patterns, yet they require manual feature engineering and model design. Recently, deep learning techniques have emerged as powerful alternatives, employing deep neural networks (DNNs) to automatically extract complex temporal patterns through representation learning. These modern deep learning methods are systematically organized based on two primary aspects: *model architecture* and *representation domain*.

## MODEL ARCHITECTURES

**Attention-based models** utilize self-attention and cross-attention mechanisms to capture long-range dependencies across time points, such as Informer (Zhou et al., 2021) (sparse attention for efficiency), Autoformer (Wu et al., 2021) (decomposition-enhanced attention), Crossformer (Zhang & Yan, 2022) (cross-dimension dependency modeling), iTransformer (Liu et al., 2023a) (inverted architecture), PatchTST (Nie et al., 2023) (patch-based processing), and Leddam (Yu et al., 2024) (learnable decomposition). **Linear-based models** employ efficient architectures with minimal parameters, achieving competitive performance through carefully designed linear operations. Representative models include N-BEATS (Oreshkin et al., 2020) (interpretable basis expansions), DLinear (Zeng et al., 2023) (decomposition + linear projection), TSMixer (Chen et al., 2023) (all-MLP architecture with cross-variate mixing), and TimeMixer (Wang et al., 2024) (Multi-scale decomposition). **Convolution-based models** capture local patterns through specialized convolution operations. Representative models include TCN (Bai et al., 2018) (temporal dilated convolution), MICN (Wang et al., 2023) (multi-scale isometric convolution) and TimesNet (Wu et al., 2023) (2D convolution).

## REPRESENTATION DOMAINS

**Time domain models** process raw time series directly, preserving the original temporal structure. Most deep learning methods (Zhang & Yan, 2022; Nie et al., 2023; Zeng et al., 2023; Wang et al., 2024; Wu et al., 2023), operate primarily in time domain. **Frequency domain models** employ frequency analysis to extract periodic patterns through Fourier or wavelet analysis, such as FEDformer (Zhou et al., 2022b) (frequency attention), FreTS (Yi et al., 2023) (frequency-domain MLPs) and TimeKAN (Huang et al., 2025) (KAN-based frequency decomposition). **Mathematical transform domains** utilize functional projections to represent time series in alternative mathematical spaces, like FiLM (Zhou et al., 2022a) (Legendre polynomial representation) and DAM (Darlow et al., 2024) (trigonometric basis domain).

Despite architectures and representation domains, most models exhibit imbalanced temporal dependency, which rely heavily on recent data. Therefore, they exhibit limited robustness to perturbations in recent data. TameR utilizes a random sampling technique in basis domain, enabling balanced context utilization and perturbation robustness.

# B. Details of TameR

## B.1. Pseudocode of BARS

The pseudocode of the BARS strategy is shown in Alg. 1. The selected frequencies for the basis functions are categorized into four tiers based on temporal scales, encompassing a total of 179 distinct frequencies. The specifics are as follows (the right endpoint for each level is excluded.):

- **Minute-level** (5-minute interval): Frequencies range from 1 minute to 60 minutes, with a 5-minute step size. This tier comprises 12 frequencies.
- **Hour-level** (15-minute interval): Frequencies range from 60 minutes to 1440 minutes (24 hours), with a 15-minute step size. This tier comprises 92 frequencies.
- **Day-level** (6-hour interval): Frequencies range from 24 hours to 168 hours (7 days), with a 6-hour step size. This tier comprises 24 frequencies.
- **Week-level** (1-week interval): Frequencies range from 1 week to 52 weeks, with a 1-week step size. This tier comprises 51 frequencies.

---

**Algorithm 1** Basis-Aligned Randomized Sampling (BARS)

---

**Input:** Historical time series $\mathbf{X} \in \mathbb{R}^{H \times C}$, sampling rate $\rho$, regularization $\lambda$
**Output:** Basis coefficients $\theta \in \mathbb{R}^{2K \times C}$

1: $\mathcal{I} \leftarrow \{1, 2, \ldots, H\}$            *# Full time indices*
2: $\tau \leftarrow \text{RandomSample}(\mathcal{I}, \lfloor \rho H \rfloor)$            *# Random time point sampling*
3: $\mathbf{X}_{\text{sampled}} \leftarrow \mathbf{X}[\tau, :]$            *# Time series values*
4: ───────────────────────────────────────
5: **Basis Transformation**
6: $K \leftarrow 179$            *# Frequencies: min/hr/day/week*
7: Initialize $\Phi \leftarrow \mathbf{0}_{\lfloor \rho H \rfloor \times 2K}$            *# Basis matrix*
8: **for** $k \in \{1, 2, \ldots, K\}$ **do**
9:     $\Phi[:, 2k-1] \leftarrow \sin(2\pi f_k \tau)$            *# Sine basis*
10:     $\Phi[:, 2k] \leftarrow \cos(2\pi f_k \tau)$            *# Cosine basis*
11: **end for**
12: ───────────────────────────────────────
13: **Per-Channel Basis Coefficient Solution**
14: Initialize $\theta \leftarrow \mathbf{0}_{2K \times C}$            *# Coefficients matrix*
15: $\mathbf{A} \leftarrow \Phi^\top \Phi + \lambda I$
16: **for** $c = 1$ **to** $C$ **do**
17:     $\mathbf{b}_c \leftarrow \Phi^\top \mathbf{X}_{\text{sampled}}[:, c]$
18:     $\theta[:, c] \leftarrow \mathbf{A}^{-1} \mathbf{b}_c$
19: **end for**
20: **return** $\theta$

---

### B.2. Pseudocode of 2-Stage Training

The pseudocode for the BARS strategy is presented in Alg. 2. During stage 1, only the parameters of the LPE module are updated. In stage 2, both the backbone model and the LPE module are optimized simultaneously.

## C. Theoretical Analysis

In this section, we provide a theoretical analysis for understanding the vulnerability of standard TSF methods to perturbations in recent data and how TameR mitigates this problem.

### C.1. Decay of $\|\mathbf{A}^k\|$ in Eq. 3

To analyze the decay of $\|\mathbf{A}^k\|$, we introduce Gelfand's Formula (Dunford & Schwartz, 1964).

**Theorem C.1** (Gelfand's Formula). *For any matrix $\mathbf{A} \in \mathbb{C}^{n \times n}$, the spectral radius $\rho(\mathbf{A})$ satisfies:*

$$\rho(\mathbf{A}) = \lim_{k \to \infty} \|\mathbf{A}^k\|^{1/k}. \tag{4}$$

**Lemma C.2.** *If $\rho(\mathbf{A}) < 1$, then $\|\mathbf{A}^k\|$ decays exponentially as $k \to \infty$. Specifically, for any $\epsilon > 0$, there exists a constant $C_\epsilon$ such that $\|\mathbf{A}^k\| \leq C_\epsilon (\rho(\mathbf{A}) + \epsilon)^k$.*

*Proof.* From Theorem C.1, for any $\epsilon > 0$, there exists an integer $N$ such that for all $k \geq N$, $\|\mathbf{A}^k\|^{1/k} < \rho(\mathbf{A}) + \epsilon$. Consequently, $\|\mathbf{A}^k\| < (\rho(\mathbf{A}) + \epsilon)^k$ for large $k$. Assuming stationarity ($\rho(\mathbf{A}) < 1$), we can choose a sufficiently small $\epsilon$ such that $\rho(\mathbf{A}) + \epsilon < 1$. Thus, $\|\mathbf{A}^k\|$ follows an exponential decay rate governed by $\rho(\mathbf{A})$. □

### C.2. Robustness of BARS

The analysis in Sec. 3.2 demonstrates that standard training results in a model with larger weights in relation to recent data. TameR utilizes BARS during training to mitigate this issue. BARS fundamentally alters the optimization, resulting in a more robust TSF model during inference. Let $\mathbf{m} \in \{0, 1\}^H$ be the binary sampling mask where $m_i \sim \text{Bernoulli}(\rho)$. The

---

**Algorithm 2** 2-Stage Training

---

**Input:** Stage-1 single-layer linear model Linear, Stage-2 backbone model $\mathcal{M}$, Dataset $\mathcal{D}$, historical length $H$, forecast length $F$, sampling rate $\rho$, regularization $\lambda$

**Output:** Trained model parameters $\Theta_{\text{LPE}}, \Theta_{\mathcal{M}}$

 1: **Stage 1: Periodic Cycle Learning**
 2: **for** each batch $(\mathbf{X}, \mathbf{Y}, \mathbf{T}_{\text{hist}}, \mathbf{T}_{\text{future}}) \in \mathcal{D}$ **do**
 3: $\quad$ $\mathbf{P}_{\text{hist}}, \mathbf{P}_{\text{future}} \leftarrow \text{LPE}(\mathbf{T}_{\text{hist}}), \text{LPE}(\mathbf{T}_{\text{future}})$
 4: $\quad$ $\mathbf{R}_{\text{hist}} \leftarrow \mathbf{X} - \mathbf{P}_{\text{hist}}$
 5: $\quad$ $\hat{\mathbf{R}}_{\text{future}} \leftarrow \text{Linear}(\mathbf{R}_{\text{hist}})$
 6: $\quad$ $\hat{\mathbf{Y}} \leftarrow \hat{\mathbf{R}}_{\text{future}} + \mathbf{P}_{\text{future}}$
 7: $\quad$ $\mathcal{L} \leftarrow \text{MAE}(\mathbf{Y}, \hat{\mathbf{Y}})$
 8: $\quad$ Update $\Theta_{\text{LPE}}$ and linear layer parameters
 9: **end for**
10: $\quad$ ――――――――――――――――――――――――――――――――――――
11: **Stage 2: Joint Space Learning**
12: $\text{LPE}_{\text{init}} \leftarrow \Theta_{\text{LPE}}$ $\hfill$ *# Initialize with Stage 1 cycles*
13: **for** each batch $(\mathbf{X}, \mathbf{Y}, \mathbf{T}_{\text{hist}}, \mathbf{T}_{\text{future}}) \in \mathcal{D}$ **do**
14: $\quad$ $\mathbf{P}_{\text{hist}}, \mathbf{P}_{\text{future}} \leftarrow \text{LPE}(\mathbf{T}_{\text{hist}}), \text{LPE}(\mathbf{T}_{\text{future}})$
15: $\quad$ $\mathbf{R}_{\text{hist}} \leftarrow \mathbf{X} - \mathbf{P}_{\text{hist}}$
16: $\quad$ $\theta_{\text{hist}} \leftarrow \text{BARS}(\mathbf{R}_{\text{hist}}, \rho, \lambda)$
17: $\quad$ $\hat{\theta}_{\text{future}} \leftarrow \mathcal{M}(\theta_{\text{hist}})$
18: $\quad$ $\hat{\mathbf{R}}_{\text{future}} \leftarrow \Phi_{\text{future}} \hat{\theta}_{\text{future}}$
19: $\quad$ $\hat{\mathbf{Y}} \leftarrow \hat{\mathbf{R}}_{\text{future}} + \mathbf{P}_{\text{future}}$
20: $\quad$ $\mathcal{L} \leftarrow \text{MAE}(\mathbf{Y}, \hat{\mathbf{Y}})$
21: $\quad$ Update $\Theta_{\text{LPE}}, \Theta_{\mathcal{M}}$
22: **end for**

---

model is trained to minimize the global objective function $\mathcal{J}$, defined as the expected loss over the sampling distribution: $\mathcal{J} = \mathbb{E}_{\mathbf{m}}[\mathcal{L}(\mathcal{F}(\mathbf{X} \odot \mathbf{m}), \mathbf{Y})]$.

**Proposition C.3.** *Consider a specific input point* $\mathbf{x}_t$ *(e.g., the most recent data point), the effective gradient of the global objective* $\mathcal{J}$ *with respect to* $\mathbf{x}_t$ *is:*

$$\nabla_{\mathbf{x}_t} \mathcal{J} = \rho \cdot \mathbb{E}_{\mathbf{m}_{-t}} \left[ \nabla_{\mathbf{x}_t} \mathcal{L} \mid m_t = 1 \right], \tag{5}$$

*where* $\mathbf{m}_{-t}$ *denotes the set of mask variables excluding index* $t$ *(i.e.,* $\{m_i\}_{i \neq t}$*).*

This relationship indicates that the learning signal derived from a specific $\mathbf{x}_t$ is inherently reduced in strength. Furthermore, optimizing $\mathcal{J}$ compels the model to avoid exclusive dependence on $\mathbf{x}_t$, as it must also effectively manage loss in situations where $m_t = 0$.

*Proof.* The global objective $\mathcal{J}$ measures the weighted average risk across all possible sampling masks. We can decompose this expectation based on whether the specific point $x_t$ is visible ($m_t = 1$) or masked ($m_t = 0$):

$$\mathcal{J} = P(m_t = 0) \underbrace{\mathbb{E}_{\mathbf{m}_{-t}}[\mathcal{L} \mid m_t = 0]}_{\mathcal{L}_{\text{without } x_t}} + P(m_t = 1) \underbrace{\mathbb{E}_{\mathbf{m}_{-t}}[\mathcal{L} \mid m_t = 1]}_{\mathcal{L}_{\text{with } x_t}}. \tag{6}$$

Standard training only optimizes the second term (with $P(m_t = 1) = 1$), allowing the model to greedily assign all weight to $\mathbf{x}_t$ if it has high correlation. In contrast, BARS (with $P(m_t = 0) = 1 - \rho > 0$) introduces the first term $\mathcal{L}_{\text{without } x_t}$. In this scenario, $\mathbf{x}_t$ is zeroed out, so the model **must** rely on other context data points (contained in $\mathbf{m}_{-t}$) to minimize the error. By jointly facilitating both terms, the learned parameters are forced to capture dependencies from the global context rather than overfitting to the recent point $\mathbf{x}_t$. The derived gradient $\nabla_{\mathbf{x}_t} \mathcal{J} = \rho \cdot \nabla_{\mathbf{x}_t} \mathcal{L}_{\text{with } x_t}$ (since $\nabla_{\mathbf{x}_t} \mathcal{L}_{\text{without } x_t} = 0$) mathematically reflects this regularization: the *relative* strength of the gradient from $\mathbf{x}_t$ versus context is reduced, leading to a distributed importance distribution during inference. $\hfill \square$

While sampling reduces the probability of relying on corrupted data points, the basis projection $\boldsymbol{\theta} = \mathcal{P}_{\boldsymbol{\Phi}}(\mathbf{X}_{\text{sampled}})$ acts as a smoothing operator. For local perturbations $\boldsymbol{\delta}$ in recent data (*e.g.*, a specific segment or point), the induced change in the basis coefficient space $\Delta\boldsymbol{\theta}$ is mitigated.

**Proposition C.4.** *Consider a perturbation vector $\boldsymbol{\delta}$ localized in a short time window. Under the regularized least-squares projection in BARS with sample size $L = \lfloor \rho H \rfloor$, the variation on any single basis dimension $k$ scales inversely with $L$:*

$$|\Delta\theta_k| \leq O\left(\frac{1}{L}\right) \|\boldsymbol{\delta}\|_1. \tag{7}$$

*Proof.* In BARS, the coefficients are computed via the regularized least-squares solution $\boldsymbol{\theta} = (\Phi^\top \Phi + \lambda I)^{-1}\Phi^\top \mathbf{X}_{\text{sampled}}$ (see Alg. 1). Let $\mathbf{P} = (\Phi^\top \Phi + \lambda I)^{-1}\Phi^\top$ denote the effective projection operator. A perturbation $\boldsymbol{\delta}$ in the input leads to a change $\Delta\boldsymbol{\theta} = \mathbf{P}\boldsymbol{\delta}$. While BARS utilizes standard sine/cosine basis functions bounded by 1 (*i.e.*, $|\Phi_{tk}| \leq 1$), the least-squares formulation introduces an implicit normalization. Due to the approximate orthogonality of trigonometric bases, the Gram matrix satisfies $\Phi^\top \Phi \approx \frac{L}{2}I$. Consequently, the regularization term $(\Phi^\top \Phi + \lambda I)^{-1}$ acts approximately as a scaling factor proportional to $\frac{2}{L}$. The effective projection elements $P_{kt}$ thus have a magnitude of order $O(1/L)$. The shift in the $k$-th coefficient is $\Delta\theta_k = \sum_t P_{kt}\delta_t$, bounded by $|\Delta\theta_k| \leq \max |P_{kt}| \|\boldsymbol{\delta}\|_1 \approx O(\frac{1}{L})\|\boldsymbol{\delta}\|_1$. This confirms that the least-squares mechanism scales down local perturbations by spreading them across the global coefficient space, enabling the model to perceive local perturbations in time domain as gradual global adjustments in basis function domain. □

## D. Point Importance Scoring

Techniques such as Grad-CAM (Selvaraju et al., 2017) from the computer vision area and methods (Assaf & Schumann, 2019) in the time series analysis area employ gradient-based interpretability strategies. Grad-CAM generates gradient-weighted class activation maps to highlight influential regions in input images, whereas Assaf & Schumann (2019) quantifies feature importance through input gradient magnitudes within multivariate time series. Inspired by these methods, we propose a gradient-based point importance scoring method for complex time series forecasting models.

Specifically, given a historical time series $\mathbf{X} = \{\mathbf{x}_{t-H+1}, \mathbf{x}_{t-H+2}, \ldots, \mathbf{x}_t\} \in \mathbb{R}^{H \times C}$ with historical length $H$ and $C$ channels, we first compute the raw temporal importance for each channel as:

$$w_t^{(c)} = \left| \frac{\partial \mathcal{L}_{\text{MSE}}}{\partial \mathbf{x}_t^{(c)}} \right| \tag{8}$$

where $\mathcal{L}_{\text{MSE}}$ refers to Mean Squared Error which is well established to assess the prediction accuracy. To enable temporal comparisons across all channels, we apply channel-wise normalization:

$$\tilde{w}_t^{(c)} = \frac{w_t^{(c)} - \min\limits_{k \in [1,H]} w_k^{(c)}}{\max\limits_{k \in [1,H]} w_k^{(c)} - \min\limits_{k \in [1,H]} w_k^{(c)}} \tag{9}$$

yielding normalized importance scores $\tilde{w}_t^{(c)} \in [0, 1]$ along the temporal dimension per channel. This represents each time point's relative contribution to the prediction within its channel context.

To generate a global temporal importance profile, we apply statistical interval estimation (Neyman, 1937) by aggregating $\tilde{w}_t^{(c)}$ from the test dataset $\mathcal{D}_{\text{test}}$, which comprises $N$ sequence samples used for forecasting. Because we only focus on the importance of different time points in each sequence, we treat the sequence from each channel as an individual sample. Consequently, the number of all samples for interval estimation achieving $N \times C$. For each time point $t$, we compute the mean of $\tilde{w}_t^{(c)}$, *i.e.*, $\mu(t)$, as follows:

$$\mu(t) = \frac{1}{N \times C} \sum_{n=1}^{N} \sum_{c=1}^{C} \tilde{w}_{t,n}^{(c)} \tag{10}$$

where $\tilde{w}_{t,n}^{(c)}$ denotes the normalized importance at time $t$ for channel $c$ in sequence $n$. The standard error of the mean is given by:

$$\text{SE}(t) = \frac{s(t)}{\sqrt{N \times C}} \tag{11}$$

with

$$s(t) = \sqrt{\frac{1}{N \times C - 1} \sum_{n=1}^{N} \sum_{c=1}^{C} \left( \tilde{w}_{t,n}^{(c)} - \mu(t) \right)^2}$$

According to the central limit theorem (Kwak & Kim, 2017), the sampling distribution of $\mu(t)$ converges to a normal distribution as the number of samples increases. We therefore construct the 95% confidence interval:

$$\text{CI}_{95\%}(t) = [\mu(t) - z_{0.975} \cdot \text{SE}(t), \ \mu(t) + z_{0.975} \cdot \text{SE}(t)] \tag{12}$$

where $z_{0.975} \approx 1.96$ is the $97.5th$ percentile of the standard normal distribution. Based on the line plot of $\mu(t)$ and $\text{CI}_{95\%}(t)$, we can analyze the importance score distribution of historical time points for complex models.

# E. Experimental Details

## E.1. Datasets

*Table 3.* Statistics of datasets

| Dataset | Domain | Frequency | Lengths | Dim | Split | Shifting | Correlation | Cycle $W$ |
|---|---|---|---|---|---|---|---|---|
| ETTh1 (Zhou et al., 2021) | Electricity | 1 hour | 14,400 | 7 | 6:2:2 | -0.06 | 0.63 | 24 |
| ETTh2 (Zhou et al., 2021) | Electricity | 1 hour | 14,400 | 7 | 6:2:2 | -0.40 | 0.51 | 24 |
| ETTm1 (Zhou et al., 2021) | Electricity | 15 mins | 57,600 | 7 | 6:2:2 | -0.06 | 0.61 | 96 |
| ETTm2 (Zhou et al., 2021) | Electricity | 15 mins | 57,600 | 7 | 6:2:2 | -0.41 | 0.50 | 96 |
| Weather (Wu et al., 2021) | Environment | 10 mins | 52,696 | 21 | 7:1:2 | 0.21 | 0.69 | 144 |
| Exchange (Lai et al., 2018) | Economic | 1 day | 7,588 | 8 | 7:1:2 | 0.33 | 0.57 | 512 |
| Solar (Lai et al., 2018) | Energy | 10 mins | 52,560 | 137 | 6:2:2 | 0.20 | 0.79 | 144 |
| Traffic (Wu et al., 2021) | Traffic | 1 hour | 17,544 | 862 | 7:1:2 | 0.07 | 0.81 | 168 |

We conduct comprehensive evaluations using multiple datasets following a popular benchmark, TFB (Qiu et al., 2024), adopting its standardized data partitioning scheme and rolling forecasting protocol. Our selected datasets encompass five key domains to ensure broad coverage of real-world forecasting scenarios: **Electricity** (ETTh1 (Zhou et al., 2021), ETTh2 (Zhou et al., 2021), ETTm1 (Zhou et al., 2021), ETTm2 (Zhou et al., 2021)), **Environment** (Weather (Wu et al., 2021)), **Energy** (Solar (Lai et al., 2018)), **Traffic** (Traffic (Wu et al., 2021)), and **Economic** (Exchange (Lai et al., 2018)). Tbl. 3 provides detailed information of the selected datasets. These datasets exhibit significant diversity across sampling frequencies, channel dimensions, and whole sequence lengths.

In Tbl. 3, we also present the cycle length $W$ configuration for different datasets. For domains with natural periodicity, like electricity or weather, the cycle length $W$ is set to the natural period (*e.g.*, 24 hours for daily cycles). In particular, for domains such as economics or finance that lack clear periodicity, we configure $W \geq H$ to capture patterns across broader contexts.

TFB also defines datasets with distinct attributes: *Shifting* indicates a change in the probability distribution of a time series over time, with higher severity as the shifting value approaches 1. *Correlation* denotes the dependency among different variables in a multivariate time series. We employ TFB's unified evaluation protocol with four prediction lengths $F$ (96, 192, 336, 720). For a fair comparison, the historical sequence length $H$ is uniformly set to 96 for both TameR and all baseline models.

## E.2. Baselines

We select representative deep learning models for time series forecasting spanning diverse architectures and operational domains for comprehensive comparison. Approaches operating in the time domain include: linear-based models (DLinear (Zeng et al., 2023), TimeMixer (Wang et al., 2024), CycleNet (Lin et al., 2024)), attention-based models (Crossformer (Zhang & Yan, 2022), PatchTST (Nie et al., 2023), iTransformer (Liu et al., 2023a) and Leddam (Yu et al., 2024)), and convolution-based models (MICN (Wang et al., 2023), TimesNet (Wu et al., 2023)). Complementary approaches operating in different domains include frequency-domain models (FEDformer (Zhou et al., 2022b), FreTS (Yi et al., 2023) and TimeKAN (Huang et al., 2025)), and mathematical-transform-domain models (FiLM (Zhou et al., 2022a)). We also include RobustTSF (Cheng et al., 2024) which aims for time series forecasting with anomalies. Detailed information of these models can be found in Appendix A.

### E.3. Perturbation Scenarios

We categorize perturbation scenarios based on two dimensions: *temporal position* and *perturbation type*. The temporal position distinguishes between:

- **Recent**: Affecting the most recent data points in the historical window
- **Random**: Affecting randomly selected data points throughout the sequence

The perturbation magnitude is controlled by a perturbation ratio $\alpha$, which quantifies the extent of perturbation relative to the input sequence's inherent variability. During robustness evaluation, we randomly inject perturbations into each test sample independently. For a specific historical time series $\mathbf{x} = \{x_1, x_2, \ldots, x_T\}$, we calculate the standard deviation $\sigma_{\text{hist}}$ independently for each channel to preserve channel-specific scale characteristics. The perturbation $\delta$ applied to an anomalous point is sampled from $\mathcal{N}(0, (\alpha \cdot \sigma_{\text{hist}})^2)$. We set $\alpha = 3.0$ for all experiments involving anomalous perturbations. In order to simulate a variety of perturbation scenarios common in real-world data, such as noise, anomalies, and missing values, we implement distinct types of perturbations. We distinguish anomalies and noise by the perturbation ratio $\alpha$, with $\alpha = 1$ signifying noise and $\alpha = 3$ denoting anomalies. In the primary experiments, we concentrate on investigating anomalies and missing values, while noise is addressed briefly for clarity (Fig. 9b). This is because anomalies and missing values typically exert a more significant influence on model performance compared to general background noise.

Combining with different perturbation types, the perturbation scenarios are defined as follows:

Recent temporal position:

- **Single anomalous point:** The final time point is perturbed: $x_T^{\text{pert}} = x_T + \delta$
- **Continuous anomalous sequence:** A continuous segment of length $L$ (randomly sampled from $\{2, 3, 4, 5\}$) ending at $T$ is perturbed: $x_t^{\text{pert}} = x_t + \delta_t$ for $t = T - L + 1, \ldots, T$, where each $\delta_t$ is independently sampled
- **Single missing point:** The final time point is simulated as missing by setting the raw value to zero:: $x_T^{\text{pert}} = 0$

Random temporal position:

- **Single anomalous point:** A random time point $t$ (uniformly sampled from $\{1, \ldots, T\}$) is perturbed: $x_t^{\text{pert}} = x_t + \delta$
- **Continuous anomalous sequence:** A continuous segment of length $L$ (randomly sampled from $\{2, 3, 4, 5\}$) starting at random position $s$ (uniformly sampled from $\{1, \ldots, T - L + 1\}$) is perturbed: $x_t^{\text{pert}} = x_t + \delta_t$ for $t = s, \ldots, s + L - 1$
- **Single missing point:** A random time point $t$ (uniformly sampled from $\{1, \ldots, T\}$) is missing: $x_t^{\text{pert}} = 0$
- **Multiple anomalous points:** K isolated points (randomly sampled from $\{2, 3, 4, 5\}$) are selected without replacement and each perturbed: $x_{t_k}^{\text{pert}} = x_{t_k} + \delta_{t_k}$ for $k = 1, \ldots, K$

Note that we exclude the "recent, multiple anomalous points" scenario as it is not well-defined. Recent multiple points would typically form a continuous segment, which is already covered by the continuous anomalous sequence scenario. Finally, we achieve 7 distinct perturbation scenarios that cover common real-world data quality issues. All experimental results are averaged across prediction lengths $F \in \{96, 192, 336, 720\}$ to ensure comprehensive evaluation.

### E.4. Evaluation Metrics

We evaluate forecasting accuracy using two well-established metrics: MSE (Mean Squared Error) and MAE (Mean Absolute Error). Both metrics are calculated on normalized datasets. For perturbation robustness evaluation, we evaluate the impact of perturbations by reporting post-perturbation MSE and MAE.

### E.5. Implementation Details

We implement TameR within both Time Series Library (Wu et al., 2023) and TFB (Qiu et al., 2024), with model and experimental code publicly available at `https://github.com/NetManAIOps/TameR`. All experiments were conducted using PyTorch (Paszke et al., 2019) with the Adam optimizer (Kingma & Ba, 2014), executed on one NVIDIA Tesla V100 GPU with 32GB memory. For TameR, we employ RevIN (Kim et al., 2021) normalization, a sampling rate $\rho = 0.75$ and a regularization $\lambda = 0.1$ in BARS. For different datasets, we assign distinct cycle lengths $W$ based on their inherent periodicity (see Tbl. 3). Notably, in the case of the Exchange dataset, which lacks an intrinsic period, we set $W = 512$ to represent a prolonged cycle length. We select a *dual-layer MLP* with ReLU activation as the backbone architecture. Hyperparameter searching is performed for hidden size of MLP from 64 to 512, learning rate from 0.0001 to 0.01, batch size from 8 to 64. Baseline models utilize hyperparameters derived from TFB, and all models are trained using MAE loss

to ensure fair robustness comparisons. All experiments are repeated with multiple random seeds and the reported results are the mean values. Notably, to prevent suboptimal performance due to the loss function shift (from MSE to MAE), we validate the learning rates and convergence behaviors for all baselines, making minor adjustments where necessary to ensure they are fully optimized for the MAE loss. The initial batch size is set according to the default TFB settings. In the event of an Out-Of-Memory (OOM) error, the batch size can be reduced by half, with a minimum size of 4.

## F. Experimental Analysis of Different Perturbation Scenarios

As indicated in Tbl. 1, 4, 5, TameR demonstrates a notable performance advantage over perturbations in recent data, achieving more than **30** 1$^{st}$ Count. In contrast, existing baseline models exhibit limited robustness when dealing with perturbations in recent data. Regarding perturbations occurring at random temporal positions, Tbl. 6, 7, 8, 9 reveal that TameR maintains robustness in such scenarios. Existing baselines perform comparatively better with perturbations at random positions than they do with recent data. This underscores that most current TSF methods depend predominantly on recent data within the historical window rather than leveraging information uniformly from the global context.

We perform a comprehensive analysis of recent-data bias in existing TSF methods by examining their performance under data perturbations in recent data. By analyzing Tbl. 1, 4, 5 and Fig. 6, we find different robustness patterns associated with various *model architectures* and *representation domains*. **Findings:**

1. *Linear-based* models such as DLinear, TimeMixer, and CycleNet show increased vulnerability to perturbations due to a heavy reliance on recent observations.
2. *Attention-based* models, such as PatchTST, iTransformer and Leddam, remain susceptible to perturbations in recent data despite having complex architectures with many parameters. Crossformer experiences a slight increase in MSE following perturbations, but its absolute forecasting MSE is higher than TameR in most datasets.
3. *Convolution-based* models demonstrate greater robustness, with TimesNet showing minimal performance degradation under perturbed conditions. However, the absolute forecasting accuracy of TimesNet post-perturbation is generally lower than that of TameR across most datasets.
4. *Frequency-domain* models like FreTS and mathematical-transform-domain models such as FiLM also experience significant performance declines under perturbations. While FEDformer shows minor performance deterioration, its overall accuracy is constrained by inadequate attention to recent temporal patterns, limiting its practical applicability.
5. RobustTSF, designed for *time series forecasting with anomalies*, suffers substantial performance degradation with perturbations in recent data. This is mainly because RobustTSF is not specifically optimized to handle anomalies in recent data within historical windows, and the anomaly detection technique it used may introduce errors during model training.

*Table 4.* Robustness under perturbations (recent, continuous anomalous sequence).

| Models | ETTh1 | | ETTh2 | | ETTm1 | | ETTm2 | | Weather | | Exchange | | Traffic | | Solar | | 1$^{st}$ Count | Avg Rank |
|---|---|---|---|---|---|---|---|---|---|---|---|---|---|---|---|---|---|---|
| | MSE | MAE | MSE | MAE | MSE | MAE | MSE | MAE | MSE | MAE | MSE | MAE | MSE | MAE | MSE | MAE | | |
| DLinear | 0.594 | 0.507 | 0.456 | 0.457 | 0.654 | 0.488 | 0.337 | 0.371 | 0.388 | 0.351 | 0.373 | 0.435 | 1.113 | 0.481 | 3.403 | 0.877 | 5 | 5.01 |
| TimeMixer | 0.617 | 0.514 | 0.434 | 0.436 | 0.617 | 0.480 | 0.344 | 0.370 | 0.324 | 0.326 | 0.484 | 0.490 | 0.610 | 0.378 | 1.073 | 0.593 | 0 | 4.05 |
| CycleNet | 0.589 | 0.495 | 0.450 | 0.446 | 0.895 | 0.596 | 0.436 | 0.441 | 0.423 | 0.380 | 0.459 | 0.472 | 0.686 | 0.386 | 1.156 | 0.576 | 0 | 5.34 |
| Crossformer | 0.536 | 0.512 | 0.895 | 0.658 | 0.522 | 0.472 | 0.583 | 0.503 | 0.282 | 0.309 | 0.619 | 0.596 | 0.681 | 0.352 | 0.342 | 0.295 | 10 | 3.99 |
| PatchTST | 0.583 | 0.500 | 0.405 | 0.419 | 0.681 | 0.501 | 0.350 | 0.371 | 0.311 | 0.317 | 0.483 | 0.493 | 0.631 | 0.365 | 0.850 | 0.546 | 0 | 3.62 |
| iTransformer | 0.542 | 0.491 | 0.424 | 0.432 | 0.655 | 0.505 | 0.349 | 0.374 | 0.327 | 0.328 | 0.439 | 0.465 | 0.633 | 0.398 | 0.968 | 0.543 | 1 | 3.86 |
| Leddam | 0.535 | 0.482 | 0.421 | 0.428 | 0.647 | 0.493 | 0.372 | 0.383 | 0.397 | 0.367 | 0.461 | 0.473 | 0.674 | 0.383 | 3.153 | 0.909 | 0 | 4.66 |
| MICN | 0.503 | 0.475 | 0.475 | 0.460 | 0.495 | 0.446 | 0.312 | 0.363 | 0.315 | 0.326 | 0.329 | 0.401 | 0.625 | 0.320 | 0.370 | 0.354 | 7 | 2.01 |
| TimesNet | 0.501 | 0.468 | 0.412 | 0.419 | 0.445 | 0.428 | 0.309 | 0.342 | 0.292 | 0.308 | 0.443 | 0.454 | 0.766 | 0.397 | 0.439 | 0.399 | 3 | 2.32 |
| FEDformer | 0.487 | 0.472 | 0.410 | 0.427 | 0.959 | 0.642 | 0.353 | 0.393 | 0.357 | 0.373 | 0.530 | 0.497 | 0.691 | 0.408 | 0.823 | 0.680 | 4 | 4.61 |
| FreTS | 0.734 | 0.553 | 0.509 | 0.473 | 0.894 | 0.564 | 0.422 | 0.413 | 0.307 | 0.334 | 0.445 | 0.472 | 0.837 | 0.436 | 1.541 | 0.627 | 0 | 5.66 |
| TimeKAN | 0.535 | 0.478 | 0.414 | 0.426 | 0.551 | 0.462 | 0.337 | 0.367 | 0.304 | 0.315 | 0.465 | 0.477 | 0.751 | 0.412 | 1.037 | 0.573 | 2 | 3.37 |
| FiLM | 0.530 | 0.474 | 0.400 | 0.414 | 0.584 | 0.465 | 0.330 | 0.359 | 0.342 | 0.326 | 0.400 | 0.435 | 1.321 | 0.722 | 1.673 | 0.730 | 0 | 3.63 |
| RobustTSF | 0.639 | 0.525 | 0.512 | 0.488 | 0.749 | 0.514 | 0.364 | 0.391 | 0.479 | 0.383 | 0.417 | 0.440 | 1.034 | 0.491 | 2.073 | 0.762 | 0 | 6.02 |
| TameR | 0.492 | 0.459 | 0.391 | 0.410 | 0.436 | 0.434 | 0.290 | 0.329 | 0.276 | 0.296 | 0.458 | 0.483 | 0.596 | 0.340 | 1.618 | 0.609 | 32 | 1.85 |

*Table 5.* Robustness under perturbations (recent, single missing point).

| Models | ETTh1 | | ETTh2 | | ETTm1 | | ETTm2 | | Weather | | Exchange | | Traffic | | Solar | | 1st Count | Avg Rank |
|---|---|---|---|---|---|---|---|---|---|---|---|---|---|---|---|---|---|---|
| | MSE | MAE | MSE | MAE | MSE | MAE | MSE | MAE | MSE | MAE | MSE | MAE | MSE | MAE | MSE | MAE | | |
| DLinear | 0.520 | 0.486 | 0.451 | 0.478 | 0.532 | 0.475 | 0.409 | 0.433 | 77.463 | 2.232 | 19.005 | 2.652 | 0.729 | 0.383 | 0.771 | 0.440 | 2 | 5.28 |
| TimeMixer | 0.534 | 0.490 | 0.416 | 0.443 | 0.474 | 0.446 | 0.391 | 0.422 | 340.153 | 4.787 | 29.535 | 4.570 | 0.516 | 0.317 | 0.343 | 0.328 | 3 | 4.41 |
| CycleNet | 0.514 | 0.473 | 0.451 | 0.470 | 0.747 | 0.587 | 0.494 | 0.483 | 68.055 | 2.266 | 10.530 | 2.924 | 0.517 | 0.310 | 0.442 | 0.341 | 1 | 5.22 |
| Crossformer | 0.548 | 0.521 | 0.887 | 0.655 | 0.548 | 0.489 | 0.554 | 0.494 | 0.488 | 0.438 | 1.851 | 1.109 | 0.664 | 0.338 | 0.310 | 0.265 | 6 | 4.02 |
| PatchTST | 0.543 | 0.498 | 0.399 | 0.427 | 0.539 | 0.492 | 0.425 | 0.454 | 2.503 | 0.561 | 10.197 | 3.026 | 0.588 | 0.333 | 0.309 | 0.324 | 1 | 3.91 |
| iTransformer | 0.504 | 0.480 | 0.406 | 0.439 | 0.536 | 0.496 | 0.415 | 0.448 | 31.469 | 2.012 | 7.377 | 2.582 | 0.534 | 0.332 | 0.401 | 0.316 | 1 | 4.25 |
| Leddam | 0.494 | 0.468 | 0.408 | 0.441 | 0.538 | 0.488 | 0.423 | 0.448 | 69.210 | 2.582 | 10.800 | 3.123 | 0.595 | 0.341 | 0.321 | 0.280 | 0 | 4.23 |
| MICN | 0.481 | 0.473 | 0.467 | 0.473 | 0.471 | 0.448 | 0.331 | 0.396 | 27.866 | 1.569 | 5.791 | 2.253 | 0.622 | 0.314 | 0.357 | 0.315 | 1 | 3.38 |
| TimesNet | 0.471 | 0.454 | 0.393 | 0.412 | 0.414 | 0.410 | 0.303 | 0.344 | 28.109 | 1.622 | 2.353 | 1.188 | 0.682 | 0.350 | 0.350 | 0.328 | 0 | 2.84 |
| FEDformer | 0.464 | 0.457 | 0.416 | 0.432 | 0.649 | 0.562 | 0.365 | 0.404 | 26.379 | 2.979 | 0.523 | 0.507 | 0.696 | 0.406 | 1.043 | 0.742 | 5 | 4.45 |
| FreTS | 0.613 | 0.532 | 0.495 | 0.490 | 0.587 | 0.512 | 0.537 | 0.513 | 4.172 | 0.631 | 8.267 | 2.574 | 0.619 | 0.360 | 0.327 | 0.290 | 0 | 4.91 |
| TimeKAN | 0.495 | 0.462 | 0.396 | 0.427 | 0.468 | 0.442 | 0.366 | 0.404 | 156.218 | 2.945 | 13.858 | 3.206 | 0.663 | 0.366 | 0.349 | 0.305 | 1 | 3.72 |
| FiLM | 0.474 | 0.448 | 0.385 | 0.408 | 0.467 | 0.436 | 0.362 | 0.392 | 15.618 | 1.114 | 1.260 | 0.791 | 1.251 | 0.693 | 0.503 | 0.420 | 3 | 3.29 |
| RobustTSF | 0.474 | 0.463 | 0.623 | 0.559 | 0.454 | 0.437 | 0.523 | 0.497 | 0.387 | 0.410 | 1.131 | 0.816 | 0.663 | 0.375 | 0.556 | 0.489 | 7 | 4.22 |
| TameR | 0.455 | 0.440 | 0.373 | 0.407 | 0.394 | 0.394 | 0.277 | 0.330 | 3.029 | 0.732 | 69.465 | 7.477 | 0.555 | 0.308 | 0.306 | 0.276 | 33 | 1.89 |

*Table 6.* Robustness under perturbations (random, single anomalous point).

| Models | ETTh1 | | ETTh2 | | ETTm1 | | ETTm2 | | Weather | | Exchange | | Traffic | | Solar | | 1st Count | Avg Rank |
|---|---|---|---|---|---|---|---|---|---|---|---|---|---|---|---|---|---|---|
| | MSE | MAE | MSE | MAE | MSE | MAE | MSE | MAE | MSE | MAE | MSE | MAE | MSE | MAE | MSE | MAE | | |
| DLinear | 0.454 | 0.442 | 0.418 | 0.430 | 0.403 | 0.392 | 0.287 | 0.331 | 0.272 | 0.291 | 0.318 | 0.387 | 0.723 | 0.372 | 0.427 | 0.336 | 0 | 4.90 |
| TimeMixer | 0.469 | 0.444 | 0.376 | 0.396 | 0.405 | 0.392 | 0.280 | 0.320 | 0.246 | 0.266 | 0.383 | 0.416 | 0.509 | 0.317 | 0.358 | 0.338 | 0 | 3.37 |
| CycleNet | 0.435 | 0.422 | 0.377 | 0.396 | 0.392 | 0.387 | 0.275 | 0.314 | 0.259 | 0.277 | 0.394 | 0.421 | 0.520 | 0.312 | 0.294 | 0.271 | 5 | 2.52 |
| Crossformer | 0.509 | 0.492 | 0.977 | 0.688 | 0.487 | 0.448 | 0.555 | 0.487 | 0.242 | 0.273 | 0.606 | 0.585 | 0.578 | 0.279 | 0.223 | 0.220 | 15 | 5.21 |
| PatchTST | 0.447 | 0.432 | 0.368 | 0.391 | 0.393 | 0.389 | 0.280 | 0.319 | 0.258 | 0.274 | 0.376 | 0.410 | 0.514 | 0.304 | 0.295 | 0.313 | 5 | 2.56 |
| iTransformer | 0.460 | 0.445 | 0.384 | 0.403 | 0.404 | 0.395 | 0.283 | 0.322 | 0.255 | 0.272 | 0.367 | 0.407 | 0.467 | 0.299 | 0.270 | 0.267 | 4 | 3.30 |
| Leddam | 0.443 | 0.430 | 0.372 | 0.393 | 0.398 | 0.388 | 0.279 | 0.318 | 0.239 | 0.262 | 0.364 | 0.404 | 0.510 | 0.295 | 0.366 | 0.285 | 4 | 2.09 |
| MICN | 0.438 | 0.444 | 0.463 | 0.451 | 0.389 | 0.392 | 0.285 | 0.338 | 0.247 | 0.276 | 0.309 | 0.379 | 0.613 | 0.310 | 0.313 | 0.309 | 10 | 3.69 |
| TimesNet | 0.470 | 0.452 | 0.394 | 0.407 | 0.414 | 0.408 | 0.292 | 0.327 | 0.259 | 0.281 | 0.432 | 0.446 | 0.701 | 0.360 | 0.379 | 0.352 | 0 | 5.49 |
| FEDformer | 0.454 | 0.451 | 0.407 | 0.425 | 0.599 | 0.530 | 0.342 | 0.383 | 0.305 | 0.337 | 0.530 | 0.497 | 0.654 | 0.380 | 0.548 | 0.551 | 1 | 6.62 |
| FreTS | 0.510 | 0.470 | 0.517 | 0.456 | 0.448 | 0.418 | 0.303 | 0.342 | 0.242 | 0.272 | 0.394 | 0.425 | 0.550 | 0.325 | 0.288 | 0.281 | 1 | 4.65 |
| TimeKAN | 0.454 | 0.433 | 0.380 | 0.398 | 0.396 | 0.390 | 0.282 | 0.322 | 0.250 | 0.271 | 0.397 | 0.423 | 0.647 | 0.360 | 0.347 | 0.311 | 0 | 3.68 |
| FiLM | 0.452 | 0.433 | 0.386 | 0.403 | 0.408 | 0.391 | 0.288 | 0.324 | 0.283 | 0.288 | 0.376 | 0.418 | 1.241 | 0.690 | 0.423 | 0.382 | 0 | 5.05 |
| RobustTSF | 0.455 | 0.443 | 0.436 | 0.439 | 0.405 | 0.395 | 0.285 | 0.333 | 0.272 | 0.294 | 0.375 | 0.397 | 0.672 | 0.375 | 0.357 | 0.352 | 3 | 4.99 |
| TameR | 0.436 | 0.426 | 0.371 | 0.394 | 0.383 | 0.382 | 0.269 | 0.310 | 0.250 | 0.272 | 0.358 | 0.404 | 0.546 | 0.316 | 0.290 | 0.265 | 16 | 1.86 |

*Table 7.* Robustness under perturbations (random, continuous anomalous sequence).

| Models | ETTh1 | | ETTh2 | | ETTm1 | | ETTm2 | | Weather | | Exchange | | Traffic | | Solar | | 1st Count | Avg Rank |
|---|---|---|---|---|---|---|---|---|---|---|---|---|---|---|---|---|---|---|
| | MSE | MAE | MSE | MAE | MSE | MAE | MSE | MAE | MSE | MAE | MSE | MAE | MSE | MAE | MSE | MAE | | |
| DLinear | 0.469 | 0.452 | 0.420 | 0.431 | 0.417 | 0.400 | 0.289 | 0.333 | 0.276 | 0.292 | 0.319 | 0.387 | 0.856 | 0.411 | 0.439 | 0.344 | 0 | 4.54 |
| TimeMixer | 0.485 | 0.453 | 0.383 | 0.401 | 0.422 | 0.402 | 0.282 | 0.322 | 0.248 | 0.268 | 0.391 | 0.421 | 0.538 | 0.343 | 0.483 | 0.399 | 0 | 3.53 |
| CycleNet | 0.452 | 0.432 | 0.381 | 0.398 | 0.651 | 0.520 | 0.383 | 0.409 | 0.323 | 0.333 | 0.403 | 0.426 | 0.573 | 0.337 | 0.365 | 0.316 | 3 | 4.27 |
| Crossformer | 0.515 | 0.497 | 0.892 | 0.656 | 0.493 | 0.451 | 0.557 | 0.488 | 0.243 | 0.274 | 0.606 | 0.585 | 0.670 | 0.344 | 0.237 | 0.230 | 13 | 5.26 |
| PatchTST | 0.466 | 0.445 | 0.369 | 0.392 | 0.406 | 0.397 | 0.282 | 0.321 | 0.262 | 0.277 | 0.384 | 0.415 | 0.586 | 0.339 | 0.363 | 0.338 | 7 | 2.63 |
| iTransformer | 0.483 | 0.458 | 0.391 | 0.408 | 0.431 | 0.411 | 0.288 | 0.326 | 0.258 | 0.275 | 0.373 | 0.411 | 0.508 | 0.332 | 0.327 | 0.314 | 4 | 3.52 |
| Leddam | 0.458 | 0.438 | 0.375 | 0.395 | 0.419 | 0.400 | 0.282 | 0.321 | 0.242 | 0.264 | 0.370 | 0.408 | 0.543 | 0.323 | 0.900 | 0.454 | 3 | 2.55 |
| MICN | 0.439 | 0.445 | 0.463 | 0.451 | 0.398 | 0.398 | 0.286 | 0.338 | 0.251 | 0.279 | 0.310 | 0.380 | 0.616 | 0.313 | 0.335 | 0.327 | 12 | 2.88 |
| TimesNet | 0.492 | 0.462 | 0.404 | 0.413 | 0.433 | 0.420 | 0.300 | 0.332 | 0.266 | 0.287 | 0.438 | 0.449 | 0.767 | 0.397 | 0.425 | 0.394 | 0 | 5.60 |
| FEDformer | 0.462 | 0.457 | 0.408 | 0.426 | 0.659 | 0.550 | 0.343 | 0.384 | 0.309 | 0.340 | 0.530 | 0.497 | 0.669 | 0.392 | 0.583 | 0.570 | 1 | 6.34 |
| FreTS | 0.525 | 0.478 | 0.432 | 0.429 | 0.562 | 0.464 | 0.305 | 0.346 | 0.245 | 0.276 | 0.394 | 0.426 | 0.603 | 0.358 | 0.348 | 0.315 | 0 | 4.45 |
| TimeKAN | 0.467 | 0.440 | 0.381 | 0.400 | 0.410 | 0.398 | 0.285 | 0.324 | 0.253 | 0.273 | 0.404 | 0.428 | 0.690 | 0.381 | 0.384 | 0.339 | 0 | 3.38 |
| FiLM | 0.466 | 0.442 | 0.390 | 0.406 | 0.424 | 0.399 | 0.290 | 0.325 | 0.284 | 0.289 | 0.382 | 0.420 | 1.257 | 0.696 | 0.437 | 0.390 | 0 | 4.75 |
| RobustTSF | 0.471 | 0.454 | 0.440 | 0.442 | 0.412 | 0.400 | 0.286 | 0.334 | 0.271 | 0.294 | 0.375 | 0.396 | 0.743 | 0.404 | 0.356 | 0.357 | 3 | 4.27 |
| TameR | 0.446 | 0.431 | 0.375 | 0.397 | 0.392 | 0.390 | 0.272 | 0.313 | 0.253 | 0.275 | 0.367 | 0.411 | 0.587 | 0.332 | 0.378 | 0.326 | 18 | 2.02 |

*Table 8.* Robustness under perturbations (random, single missing point).

| Models | ETTh1 | | ETTh2 | | ETTm1 | | ETTm2 | | Weather | | Exchange | | Traffic | | Solar | | 1st Count | Avg Rank |
|---|---|---|---|---|---|---|---|---|---|---|---|---|---|---|---|---|---|---|
| | MSE | MAE | MSE | MAE | MSE | MAE | MSE | MAE | MSE | MAE | MSE | MAE | MSE | MAE | MSE | MAE | | |
| DLinear | 0.451 | 0.442 | 0.417 | 0.431 | 0.400 | 0.392 | 0.287 | 0.333 | 1.691 | 0.372 | 0.571 | 0.457 | 0.678 | 0.360 | 0.340 | 0.314 | 2 | 4.25 |
| TimeMixer | 0.470 | 0.446 | 0.375 | 0.398 | 0.401 | 0.391 | 0.280 | 0.323 | 13.074 | 0.768 | 2.027 | 0.950 | 0.498 | 0.306 | 0.338 | 0.325 | 0 | 4.16 |
| CycleNet | 0.434 | 0.421 | 0.376 | 0.396 | 0.616 | 0.509 | 0.387 | 0.415 | 2.971 | 0.619 | 1.775 | 0.918 | 0.494 | 0.296 | 0.320 | 0.280 | 5 | 4.23 |
| Crossformer | 0.511 | 0.493 | 0.888 | 0.655 | 0.486 | 0.446 | 0.554 | 0.487 | 0.257 | 0.284 | 0.696 | 0.627 | 0.654 | 0.333 | 0.220 | 0.219 | 13 | 4.75 |
| PatchTST | 0.443 | 0.432 | 0.367 | 0.392 | 0.389 | 0.387 | 0.278 | 0.320 | 0.316 | 0.296 | 1.199 | 0.745 | 0.551 | 0.315 | 0.274 | 0.304 | 5 | 2.42 |
| iTransformer | 0.455 | 0.444 | 0.385 | 0.406 | 0.398 | 0.394 | 0.285 | 0.327 | 1.281 | 0.463 | 1.448 | 0.901 | 0.460 | 0.290 | 0.255 | 0.250 | 5 | 3.66 |
| Leddam | 0.444 | 0.431 | 0.373 | 0.395 | 0.395 | 0.388 | 0.279 | 0.322 | 8.217 | 0.838 | 1.300 | 0.766 | 0.504 | 0.287 | 0.232 | 0.232 | 6 | 2.64 |
| MICN | 0.439 | 0.445 | 0.461 | 0.451 | 0.389 | 0.392 | 0.283 | 0.337 | 0.825 | 0.377 | 0.509 | 0.460 | 0.612 | 0.309 | 0.284 | 0.282 | 2 | 3.51 |
| TimesNet | 0.465 | 0.450 | 0.399 | 0.413 | 0.404 | 0.403 | 0.297 | 0.334 | 13.479 | 1.168 | 1.454 | 0.897 | 0.681 | 0.349 | 0.365 | 0.338 | 0 | 5.81 |
| FEDformer | 0.453 | 0.450 | 0.406 | 0.426 | 0.577 | 0.524 | 0.340 | 0.383 | 2.670 | 0.910 | 0.470 | 0.469 | 0.656 | 0.380 | 0.670 | 0.597 | 1 | 5.72 |
| FreTS | 0.503 | 0.467 | 0.427 | 0.428 | 0.420 | 0.407 | 0.303 | 0.347 | 0.373 | 0.327 | 0.645 | 0.499 | 0.533 | 0.313 | 0.264 | 0.262 | 0 | 4.23 |
| TimeKAN | 0.452 | 0.432 | 0.377 | 0.398 | 0.391 | 0.388 | 0.281 | 0.323 | 8.772 | 0.611 | 1.854 | 0.896 | 0.642 | 0.355 | 0.323 | 0.293 | 0 | 3.88 |
| FiLM | 0.449 | 0.432 | 0.386 | 0.405 | 0.405 | 0.391 | 0.288 | 0.325 | 1.116 | 0.389 | 1.219 | 0.684 | 1.236 | 0.687 | 0.382 | 0.365 | 0 | 4.45 |
| RobustTSF | 0.447 | 0.440 | 0.443 | 0.445 | 0.396 | 0.392 | 0.289 | 0.338 | 0.271 | 0.294 | 0.401 | 0.411 | 0.641 | 0.362 | 0.343 | 0.344 | 7 | 3.88 |
| TameR | 0.434 | 0.425 | 0.369 | 0.394 | 0.379 | 0.380 | 0.267 | 0.311 | 1.301 | 0.531 | 4.390 | 1.151 | 0.552 | 0.305 | 0.246 | 0.245 | 18 | 2.40 |

*Table 9.* Robustness under perturbations (random, multi anomalous points).

| Models | ETTh1 | | ETTh2 | | ETTm1 | | ETTm2 | | Weather | | Exchange | | Traffic | | Solar | | 1st Count | Avg Rank |
|---|---|---|---|---|---|---|---|---|---|---|---|---|---|---|---|---|---|---|
| | MSE | MAE | MSE | MAE | MSE | MAE | MSE | MAE | MSE | MAE | MSE | MAE | MSE | MAE | MSE | MAE | | |
| DLinear | 0.500 | 0.470 | 0.427 | 0.436 | 0.447 | 0.419 | 0.295 | 0.338 | 0.282 | 0.297 | 0.324 | 0.392 | 1.078 | 0.487 | 0.749 | 0.442 | 0 | 4.49 |
| TimeMixer | 0.520 | 0.472 | 0.391 | 0.407 | 0.465 | 0.427 | 0.292 | 0.330 | 0.259 | 0.279 | 0.400 | 0.428 | 0.595 | 0.389 | 0.580 | 0.458 | 3 | 3.43 |
| CycleNet | 0.478 | 0.448 | 0.393 | 0.406 | 0.716 | 0.547 | 0.406 | 0.425 | 0.349 | 0.347 | 0.416 | 0.434 | 0.670 | 0.398 | 0.479 | 0.387 | 0 | 4.38 |
| Crossformer | 0.531 | 0.509 | 0.896 | 0.658 | 0.506 | 0.461 | 0.562 | 0.491 | 0.247 | 0.278 | 0.608 | 0.586 | 0.703 | 0.369 | 0.263 | 0.247 | 13 | 4.66 |
| PatchTST | 0.505 | 0.468 | 0.376 | 0.397 | 0.441 | 0.417 | 0.290 | 0.327 | 0.266 | 0.283 | 0.401 | 0.425 | 0.692 | 0.401 | 0.519 | 0.422 | 7 | 2.90 |
| iTransformer | 0.534 | 0.483 | 0.409 | 0.419 | 0.497 | 0.446 | 0.303 | 0.337 | 0.264 | 0.282 | 0.386 | 0.421 | 0.588 | 0.395 | 0.531 | 0.418 | 1 | 4.12 |
| Leddam | 0.487 | 0.456 | 0.381 | 0.400 | 0.460 | 0.425 | 0.289 | 0.328 | 0.257 | 0.276 | 0.372 | 0.409 | 0.633 | 0.387 | 2.407 | 0.784 | 1 | 2.78 |
| MICN | 0.448 | 0.451 | 0.466 | 0.453 | 0.422 | 0.415 | 0.289 | 0.341 | 0.262 | 0.291 | 0.310 | 0.381 | 0.630 | 0.322 | 0.631 | 0.536 | 15 | 2.69 |
| TimesNet | 0.539 | 0.483 | 0.428 | 0.427 | 0.477 | 0.445 | 0.314 | 0.344 | 0.282 | 0.300 | 0.448 | 0.457 | 0.900 | 0.465 | 0.519 | 0.460 | 0 | 5.67 |
| FEDformer | 0.481 | 0.468 | 0.410 | 0.427 | 0.752 | 0.585 | 0.345 | 0.386 | 0.316 | 0.345 | 0.530 | 0.497 | 0.693 | 0.409 | 0.643 | 0.604 | 1 | 5.78 |
| FreTS | 0.559 | 0.496 | 0.443 | 0.435 | 0.751 | 0.533 | 0.315 | 0.354 | 0.251 | 0.283 | 0.399 | 0.430 | 0.701 | 0.418 | 0.464 | 0.390 | 0 | 4.56 |
| TimeKAN | 0.503 | 0.460 | 0.391 | 0.406 | 0.448 | 0.422 | 0.292 | 0.331 | 0.261 | 0.282 | 0.432 | 0.445 | 0.745 | 0.426 | 0.527 | 0.430 | 1 | 3.53 |
| FiLM | 0.493 | 0.459 | 0.400 | 0.413 | 0.452 | 0.419 | 0.296 | 0.331 | 0.291 | 0.295 | 0.389 | 0.425 | 1.288 | 0.709 | 0.587 | 0.457 | 0 | 4.34 |
| RobustTSF | 0.508 | 0.475 | 0.450 | 0.448 | 0.456 | 0.421 | 0.294 | 0.340 | 0.287 | 0.302 | 0.380 | 0.401 | 0.884 | 0.474 | 0.574 | 0.430 | 2 | 4.59 |
| TameR | 0.465 | 0.442 | 0.383 | 0.402 | 0.409 | 0.402 | 0.281 | 0.320 | 0.260 | 0.282 | 0.368 | 0.415 | 0.654 | 0.380 | 0.554 | 0.427 | 20 | 2.05 |

## G. Additional Analysis of 2-Stage Training

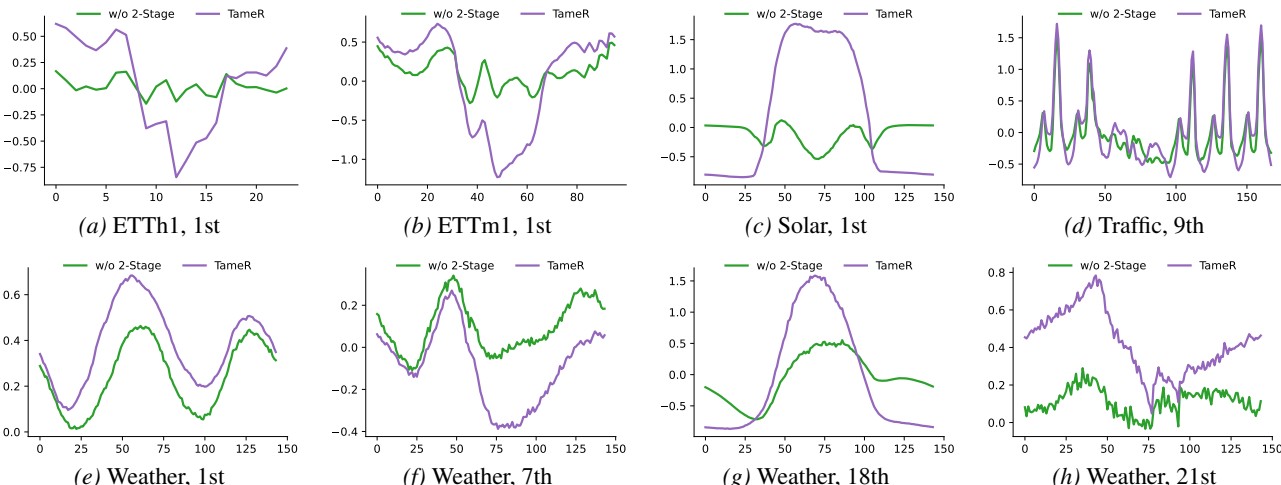

*Figure 8.* Visualization of the periodic cycles learned by TameR with and without 2-stage training. Panels (a-d) illustrate various periodic cycles extracted from different datasets. Conversely, panels (e-h) showcase diverse periodic cycles identified from different channels within the same dataset. The *i* th denotes the index of the channel within the dataset.

To further investigate the impact of 2-Stage-Training, we visualize the learned periodic cycles with and without this training approach. As illustrated in Fig. 8, without 2-Stage-Training, the learned periodic cycles tend to cluster around zero with minor fluctuations, which impairs the accurate representation of the data's inherent periodicity. Conversely, employing 2-Stage-Training allows TameR to derive more precise periodic cycles. In Fig. 8a and 8b, the periodic cycles extracted from ETTh1 and ETTm1 by TameR show a high degree of consistency, aligning with the expectation given that the primary difference between ETTh1 and ETTm1 lies in the sampling frequency. Fig. 8d highlights the distinct traffic data patterns observed on weekdays compared to weekends. Furthermore, as depicted in Fig. 8e, 8f, 8g, and 8h, different channels exhibit varying periodic patterns, warranting the use of an independent cycle for each channel when designing the cycle.

## H. Parameter Sensitivity

This section investigates how different parameters—including the sampling rate $\rho$, perturbation ratio $\alpha$, historical sequence length $H$, and prediction length $F$—affect the accuracy and robustness of TameR in forecasting tasks. We perform a series of experiments using the ETTm1 and ETTm2 datasets, and the results are depicted in Fig. 9. In each subfigure, MSE denotes the forecasting accuracy on unperturbed data, whereas $\hat{\text{MSE}}$ and $MSE \Uparrow$ indicate robustness against perturbations (recent, single anomalous point).

**Sampling rate** $\rho$. TameR enhances the use of global context information by employing randomized sampling during training. Fig. 9a reveals key insights on the effects of sampling rate $\rho$. Firstly, forecasting errors on unperturbed data slightly decrease with a higher $\rho$, although the difference between $\rho \in \{0.60, 0.75, 0.90\}$ is minimal. Secondly, perturbation robustness decreases as $\rho$ increases; specifically, $\rho = 0.60$ exhibits the lowest degradation under perturbations (recent, single anomalous point), while $\rho = 0.90$ shows the highest degradation. This is because fewer sampled time points during training encourage TameR to learn predictions based on complete contextual information rather than isolated points. Among the three sampling rates, the $\rho = 0.75$ configuration effectively balances forecasting accuracy and robustness. In practical applications, the parameter $\rho$ can be adjusted to align with the data characteristics and practical needs. Specifically, a lower $\rho$ (*e.g.*, 0.5–0.7) is preferable when the data involves frequent anomalies, or when the practical need demands extreme robustness. A higher $\rho$ (*e.g.*, 0.8–1.0) is recommended for relatively clean data or when permanent distribution shifts occur frequently. Here, the practical need prioritizes adaptation to the "new normal" and high-frequency details for achieving high clean-data accuracy. Moderate values (*e.g.*, 0.7–0.8) serves as a universally default that strikes a balance between accuracy and perturbation robustness for general forecasting tasks.

**Perturbation ratio** $\alpha$. Fig. 9b demonstrates the effect of varying perturbation ratios on the robustness of TameR. Both the prediction $\hat{\text{MSE}}$ and $MSE \Uparrow$ after perturbations exhibit an approximately linear relationship with the perturbation ratio $\alpha$, increasing as $\alpha$ grows. Notably, under in-distribution perturbations ($\alpha = 1.0$), which simulate typical measurement

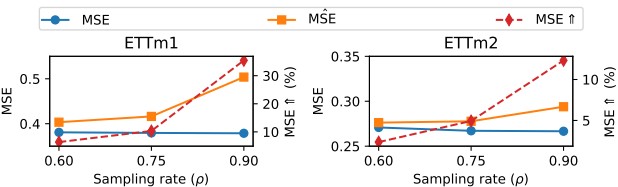
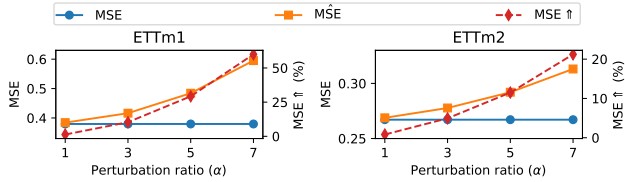

*(a)* Performance under different sampling rates $\rho$. The results are averaged from all prediction lengths of $F \in \{96, 192, 336, 720\}$.

*(b)* Performance under different perturbation ratios $\alpha$. The results are averaged from all prediction lengths of $F \in \{96, 192, 336, 720\}$.

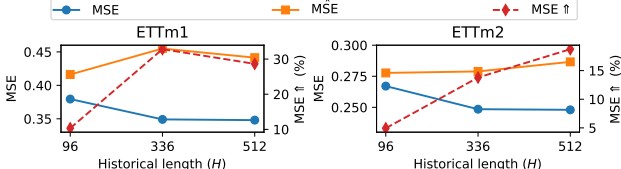
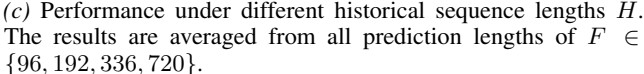
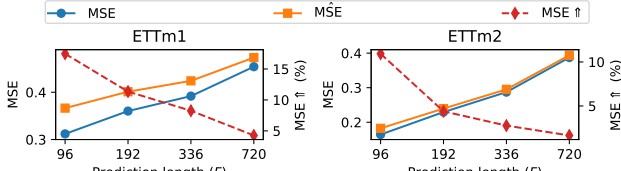

*(c)* Performance under different historical sequence lengths $H$. The results are averaged from all prediction lengths of $F \in \{96, 192, 336, 720\}$.

*(d)* Performance under different prediction lengths $F$.

*Figure 9.* Model analysis for TameR in cases of different sampling rates $\rho$, perturbation ratios $\alpha$, historical sequence lengths $H$ and prediction lengths $F$. Experiments are conducted on ETTm1 and ETTm2 with historical sequence length $H = 96$ as default. All perturbations here are (recent, single anomalous point), *i.e.*, perturbations applied to the last point of historical sequence. $\hat{MSE}$ refers to MSE after perturbations, and $MSE \Uparrow$ is the proportional rise in MSE after perturbations.

noise, TameR maintains nearly identical forecasting accuracy compared to scenarios without perturbations. This stability demonstrates TameR's inherent resilience to common data collection variability, highlighting its robustness for practical applications.

**Historical sequence length** $H$. As demonstrated in Fig. 9c, prediction accuracy peaks at $H = 336$ while diminishing at $H = 96$, indicating longer historical context windows are beneficial for latent temporal pattern learning. However, further extension to $H = 512$ does not result in improved accuracy on unperturbed data, suggesting there are diminishing returns beyond optimal context lengths. Under perturbations, models with $H = 96$ exhibit the least performance degradation, whereas longer sequences show increased sensitivity to such perturbations. This may be attributed to the fact that shorter sequences maintain tighter coupling between time points, thereby enhancing their resilience to perturbations when a single point is distorted.

**Prediction length** $F$. As illustrated in Fig. 9d, increasing the prediction length leads to a higher MSE on unperturbed data due to error accumulation over longer horizons. However, when considering perturbations, longer prediction horizons exhibit less relative performance degradation. This is because a perturbation in recent data mainly impacts predictions that are temporally closer. The farther the predicted point is from the perturbed data, the less affected it will be. As the prediction length increases, the perturbation of recent data has a smaller average effect on the overall prediction, resulting in a smaller decrease in prediction accuracy after perturbations.

## I. Frequency Design

This section studies the predefined frequency set used by BARS. A frequency is the reciprocal of a temporal period used in the sine and cosine basis functions. The default set contains 179 frequencies across four tiers: minute-level (12), hour-level (92), day-level (24), and week-level (51), *i.e.*, covering periods from 1 minute up to 1 year. More details can be found in Appendix B.1.

We construct 10 combinations (*e.g.*, "Minute–Day" includes minute- and hour-level frequencies, "Minute–Year" is our default 179-frequency set) and evaluate them on ETTh2 (hourly) and ETTm2 (15-min) under both clean and perturbed (recent, single anomalous point) data. Tbl. 10 shows that the full Minute-Year range provides the best or near-best performance on both clean and perturbed data. Removing low-frequency components (*e.g.*, Minute–Hour) tends to hurt clean-data accuracy, likely due to insufficient capacity to represent long-term trends. Removing high-frequency components (*e.g.*, Week–Year) tends to increase sensitivity to perturbations. A plausible explanation is that local anomaly has broad spectral support;

*Table 10.* Frequency set ablation on ETTh2 and ETTm2. Each cell reports MSE on clean / perturbed data.

| Frequency Sets | ETTh2 | ETTm2 |
|---|---|---|
| Minute–Hour | 0.392 / 0.398 | 0.286 / 0.291 |
| Minute–Day | 0.392 / 0.394 | 0.271 / 0.281 |
| Minute–Week | 0.368 / 0.380 | 0.268 / 0.279 |
| Minute–Year (Default) | 0.369 / **0.379** | **0.267 / 0.278** |
| Hour–Day | 0.405 / 0.407 | 0.278 / 0.294 |
| Hour–Week | 0.381 / 0.386 | 0.268 / 0.286 |
| Hour–Year | 0.379 / 0.385 | 0.268 / 0.284 |
| Day–Week | 0.373 / 0.410 | 0.271 / 0.287 |
| Day–Year | **0.368** / 0.429 | 0.269 / 0.282 |
| Week–Year | 0.370 / 0.383 | 0.272 / 0.279 |

without enough high-frequency bases, the least-squares fit may compensate by adjusting low-frequency coefficients, which can affect the global reconstruction. This intuition is consistent with our projection analysis in Proposition C.4.

*Table 11.* Comparison with a learnable-frequency variant, TameR-LF. Training Time denotes the average time required per epoch.

| Dataset | Model | MSE (Clean / Perturbed) | Training Time (s) |
|---|---|---|---|
| ETTh2 | TameR-LF | 0.392 / 0.401 | 27.4 |
| | TameR | **0.369 / 0.379** | **9.0** |
| ETTm2 | TameR-LF | 0.295 / 0.314 | 121.3 |
| | TameR | **0.267 / 0.278** | **38.8** |

We also explore a learnable-frequency variant (TameR-LF) with 256 learnable frequencies and a gating mechanism to control the number of activated frequencies. Experimental results are presented in Tbl. 11. Learnable frequencies degrade accuracy and introduce a noticeable computational overhead, since BARS relies on a closed-form least-squares projection and backpropagating through it increases training cost. Therefore, the fixed frequency set offers a stable and efficient inductive bias, while still being easy to extend for datasets with much finer or coarser sampling intervals.

## J. Efficiency

*Table 12.* Model efficiency comparison and their MSEs in forecasting on the ETTm1 dataset, with historical sequence length $H = 96$ and prediction length $F = 720$. Training Time denotes the average time required per epoch.

| Model | Parameters | GPU Memory (MB) | Training Time (s) | MSE |
|---|---|---|---|---|
| DLinear | 139.7K | **417** | 12.7 | 0.4688 |
| TimeMixer | 177.6K | 1779 | 64.5 | 0.4812 |
| CycleNet | **70.5K** | 495 | **10.7** | 0.4603 |
| Crossformer | 171.8K | 2143 | 68.2 | 0.8870 |
| PatchTST | 1.4M | 553 | 20.7 | 0.4588 |
| iTransformer | 205.1K | 436 | 17.4 | 0.4775 |
| Leddam | 5.0M | 606 | 24.5 | 0.4720 |
| MICN | 263.0K | 591 | 21.8 | 0.4578 |
| TimesNet | 666.0K | 3045 | 192.6 | 0.5450 |
| FEDformer | 16.8M | 3573 | 451.6 | 0.6584 |
| FreTS | 3.4M | 671 | 16.1 | 0.4893 |
| TimeKAN | 77.6K | 710 | 43.2 | 0.4588 |
| FiLM | 12.6M | 1301 | 114.5 | 0.4768 |
| RobustTSF | 139.7K | 417 | 13.1 | 0.4686 |
| TameR | 129.2K | 509 | 38.8 | **0.4540** |

As demonstrated in Tbl. 12, the TameR model achieves minimal forecasting error with low computational overhead. Compared to mainstream time series forecasting approaches, TameR operates with significantly fewer parameters—a key advantage stemming from its parameter count being solely determined by the number of selected frequencies rather than

scaling with input or prediction lengths. This architectural efficiency translates to substantially reduced GPU memory consumption during both training and inference phases. While the basis coefficient solution in BARS introduces moderate computational overhead relative to pure time-domain linear models, its overall training time still remains competitive. On ETTm1 dataset with $H = 96$ and $F = 720$, TameR completes training 1.7-11.6x faster than structurally complex models like TimeMixer, TimesNet, and FEDformer, while maintaining comparable to simpler linear baselines. This efficiency-performance balance positions TameR as a practical solution for resource-constrained forecasting scenarios requiring accuracy, robustness and computational economy.

## K. Limitation and Future Work

There are some limitations and future works for TameR. TameR employs a channel-independent technique, which exhibits limited efficacy on time series with a high number of variates. Future research could explore the potential of channel clustering within the BARS module. Another limitation lies in adapting to persistent concept drift. While TameR is designed to mitigate sensitivity to local perturbations by leveraging global context, a permanent distribution shift may require enhanced adjustments based on the most recent observations. In such scenarios, increasing the sampling rate $\rho$ can improve adaptation to clean data at the cost of reduced perturbation robustness. Additionally, TameR uses a fixed sampling rate for all time series inputs during training. We plan to explore dynamic sampling rates to enhance both forecasting accuracy and robustness in future research. Finally, the default selection of frequencies is designed for common forecasting datasets with sampling intervals ranging from minutes to weeks. Datasets with significantly finer or coarser sampling intervals may achieve benefit from expanding the predefined frequency set.

## L. Full Results

For robustness evaluation, Tbl. 13 presents the forecasting performance with perturbations (recent, single anomalous point). In this scenario, TameR demonstrates superior performance in the majority of settings.

Tbl. 14 presents the multivariate forecasting performance on unperturbed data for various prediction lengths, using a consistent historical length of $H = 96$ to ensure a fair comparison.

## M. Showcases

To assess the performance of different models, we perform a qualitative comparison by plotting a random dimension of the forecasting results from the test set of each dataset (Fig. 10, 11, 12, 13, 14, 15, 16). We select some models from different model architectures and representation domains. Among the various models, TameR demonstrates superior performance, particularly when anomalies or large noise occur in the last few points of the historical window.

*Table 13.* Detailed robustness results under perturbations (recent, single anomalous point).

| Models | F | ETTh1 | | ETTh2 | | ETTm1 | | ETTm2 | | Weather | | Exchange | | Traffic | | Solar | |
|---|---|---|---|---|---|---|---|---|---|---|---|---|---|---|---|---|---|
| | | MSE | MAE | MSE | MAE | MSE | MAE | MSE | MAE | MSE | MAE | MSE | MAE | MSE | MAE | MSE | MAE |
| DLinear | 96 | 0.606 | 0.483 | 0.345 | 0.384 | 0.717 | 0.492 | 0.222 | 0.296 | 0.384 | 0.326 | 0.312 | 0.405 | 0.981 | 0.450 | 3.883 | 0.945 |
| | 192 | 0.567 | 0.484 | 0.415 | 0.427 | 0.618 | 0.466 | 0.311 | 0.353 | 0.347 | 0.327 | 0.160 | 0.291 | 0.914 | 0.416 | 3.535 | 0.923 |
| | 336 | 0.569 | 0.494 | 0.461 | 0.460 | 0.601 | 0.474 | 0.353 | 0.383 | 0.375 | 0.352 | 0.265 | 0.385 | 0.846 | 0.400 | 2.964 | 0.830 |
| | 720 | 0.521 | 0.510 | 0.570 | 0.535 | 0.556 | 0.475 | 0.436 | 0.433 | 0.379 | 0.375 | 0.755 | 0.656 | 0.865 | 0.420 | 2.318 | 0.704 |
| | Avg | 0.566 | 0.493 | 0.448 | 0.451 | 0.623 | 0.477 | 0.331 | 0.366 | 0.371 | 0.345 | 0.373 | 0.434 | 0.902 | 0.422 | 3.175 | 0.851 |
| TimeMixer | 96 | 0.574 | 0.480 | 0.362 | 0.386 | 0.528 | 0.445 | 0.219 | 0.294 | 0.261 | 0.288 | 0.217 | 0.332 | 0.569 | 0.342 | 0.544 | 0.434 |
| | 192 | 0.527 | 0.470 | 0.394 | 0.406 | 0.503 | 0.434 | 0.288 | 0.338 | 0.256 | 0.286 | 0.218 | 0.340 | 0.571 | 0.348 | 0.510 | 0.437 |
| | 336 | 0.532 | 0.482 | 0.440 | 0.440 | 0.548 | 0.456 | 0.348 | 0.371 | 0.308 | 0.317 | 0.376 | 0.449 | 0.558 | 0.340 | 0.649 | 0.481 |
| | 720 | 0.574 | 0.511 | 0.433 | 0.446 | 0.555 | 0.472 | 0.416 | 0.405 | 0.365 | 0.353 | 0.944 | 0.737 | 0.575 | 0.356 | 0.575 | 0.446 |
| | Avg | 0.552 | 0.486 | 0.407 | 0.419 | 0.534 | 0.452 | 0.318 | 0.352 | 0.298 | 0.311 | 0.439 | 0.465 | 0.568 | 0.346 | 0.569 | 0.449 |
| CycleNet | 96 | 0.607 | 0.484 | 0.381 | 0.401 | 0.719 | 0.500 | 0.241 | 0.311 | 0.330 | 0.302 | 0.104 | 0.232 | 0.608 | 0.344 | 0.968 | 0.493 |
| | 192 | 0.566 | 0.479 | 0.435 | 0.430 | 0.602 | 0.466 | 0.284 | 0.334 | 0.328 | 0.316 | 0.241 | 0.357 | 0.592 | 0.337 | 0.689 | 0.418 |
| | 336 | 0.559 | 0.481 | 0.461 | 0.453 | 0.575 | 0.469 | 0.332 | 0.361 | 0.347 | 0.336 | 0.422 | 0.477 | 0.570 | 0.337 | 0.553 | 0.372 |
| | 720 | 0.530 | 0.486 | 0.461 | 0.461 | 0.580 | 0.481 | 0.421 | 0.408 | 0.391 | 0.368 | 0.950 | 0.740 | 0.580 | 0.344 | 0.445 | 0.335 |
| | Avg | 0.565 | 0.483 | 0.434 | 0.436 | 0.619 | 0.479 | 0.320 | 0.353 | 0.349 | 0.331 | 0.429 | 0.452 | 0.588 | 0.340 | 0.664 | 0.405 |
| Crossformer | 96 | 0.443 | 0.445 | 0.654 | 0.553 | 0.442 | 0.440 | 0.237 | 0.321 | 0.218 | 0.264 | 0.188 | 0.325 | 0.561 | 0.292 | 0.297 | 0.271 |
| | 192 | 0.470 | 0.462 | 0.761 | 0.601 | 0.421 | 0.422 | 0.353 | 0.405 | 0.230 | 0.270 | 0.380 | 0.469 | 0.593 | 0.282 | 0.271 | 0.265 |
| | 336 | 0.518 | 0.488 | 1.041 | 0.714 | 0.445 | 0.433 | 0.634 | 0.546 | 0.284 | 0.314 | 0.700 | 0.664 | 0.582 | 0.279 | 0.362 | 0.313 |
| | 720 | 0.665 | 0.619 | 1.465 | 0.892 | 0.748 | 0.572 | 1.060 | 0.711 | 0.362 | 0.364 | 1.181 | 0.906 | 0.635 | 0.304 | 0.379 | 0.302 |
| | Avg | 0.524 | 0.503 | 0.980 | 0.690 | 0.514 | 0.467 | 0.571 | 0.496 | 0.274 | 0.303 | 0.612 | 0.591 | 0.593 | 0.289 | 0.327 | 0.288 |
| PatchTST | 96 | 0.596 | 0.491 | 0.343 | 0.377 | 0.839 | 0.536 | 0.259 | 0.322 | 0.234 | 0.266 | 0.209 | 0.333 | 0.637 | 0.368 | 0.956 | 0.555 |
| | 192 | 0.531 | 0.473 | 0.398 | 0.406 | 0.624 | 0.484 | 0.306 | 0.349 | 0.278 | 0.299 | 0.267 | 0.379 | 0.594 | 0.350 | 0.649 | 0.504 |
| | 336 | 0.542 | 0.478 | 0.429 | 0.432 | 0.508 | 0.450 | 0.389 | 0.393 | 0.323 | 0.329 | 0.402 | 0.473 | 0.566 | 0.339 | 0.590 | 0.465 |
| | 720 | 0.547 | 0.500 | 0.430 | 0.444 | 0.608 | 0.492 | 0.428 | 0.412 | 0.390 | 0.370 | 0.993 | 0.756 | 0.607 | 0.354 | 0.704 | 0.503 |
| | Avg | 0.554 | 0.486 | 0.400 | 0.415 | 0.645 | 0.491 | 0.346 | 0.369 | 0.306 | 0.316 | 0.468 | 0.485 | 0.601 | 0.353 | 0.725 | 0.507 |
| iTransformer | 96 | 0.476 | 0.452 | 0.335 | 0.374 | 0.601 | 0.477 | 0.247 | 0.316 | 0.260 | 0.283 | 0.134 | 0.265 | 0.673 | 0.408 | 0.808 | 0.475 |
| | 192 | 0.491 | 0.463 | 0.418 | 0.420 | 0.588 | 0.476 | 0.288 | 0.337 | 0.305 | 0.314 | 0.225 | 0.344 | 0.572 | 0.361 | 0.562 | 0.416 |
| | 336 | 0.513 | 0.475 | 0.432 | 0.435 | 0.542 | 0.466 | 0.339 | 0.367 | 0.323 | 0.328 | 0.380 | 0.450 | 0.541 | 0.350 | 1.076 | 0.579 |
| | 720 | 0.518 | 0.491 | 0.430 | 0.445 | 0.567 | 0.483 | 0.433 | 0.415 | 0.379 | 0.363 | 0.903 | 0.720 | 0.544 | 0.343 | 0.571 | 0.412 |
| | Avg | 0.500 | 0.470 | 0.404 | 0.418 | 0.575 | 0.475 | 0.327 | 0.359 | 0.317 | 0.322 | 0.410 | 0.445 | 0.583 | 0.365 | 0.754 | 0.470 |
| Leddam | 96 | 0.494 | 0.455 | 0.350 | 0.381 | 0.593 | 0.464 | 0.262 | 0.323 | 0.296 | 0.318 | 0.156 | 0.288 | 0.749 | 0.388 | 2.197 | 0.703 |
| | 192 | 0.508 | 0.465 | 0.404 | 0.410 | 0.565 | 0.457 | 0.318 | 0.355 | 0.305 | 0.328 | 0.230 | 0.348 | 0.599 | 0.345 | 2.470 | 0.806 |
| | 336 | 0.525 | 0.473 | 0.433 | 0.435 | 0.580 | 0.473 | 0.362 | 0.380 | 0.351 | 0.353 | 0.425 | 0.476 | 0.576 | 0.330 | 2.697 | 0.836 |
| | 720 | 0.495 | 0.475 | 0.435 | 0.446 | 0.602 | 0.490 | 0.458 | 0.430 | 0.411 | 0.388 | 0.953 | 0.731 | 0.574 | 0.340 | 1.995 | 0.702 |
| | Avg | 0.505 | 0.467 | 0.405 | 0.418 | 0.585 | 0.471 | 0.350 | 0.372 | 0.340 | 0.347 | 0.441 | 0.461 | 0.625 | 0.351 | 2.340 | 0.761 |
| MICN | 96 | 0.470 | 0.454 | 0.318 | 0.368 | 0.512 | 0.442 | 0.220 | 0.306 | 0.269 | 0.291 | 0.104 | 0.231 | 0.595 | 0.314 | 0.300 | 0.314 |
| | 192 | 0.451 | 0.448 | 0.394 | 0.411 | 0.461 | 0.426 | 0.271 | 0.336 | 0.294 | 0.311 | 0.178 | 0.312 | 0.599 | 0.307 | 0.339 | 0.323 |
| | 336 | 0.474 | 0.463 | 0.471 | 0.465 | 0.454 | 0.431 | 0.315 | 0.361 | 0.301 | 0.319 | 0.259 | 0.387 | 0.616 | 0.313 | 0.388 | 0.351 |
| | 720 | 0.501 | 0.503 | 0.712 | 0.592 | 0.510 | 0.466 | 0.429 | 0.435 | 0.316 | 0.331 | 0.759 | 0.661 | 0.680 | 0.338 | 0.409 | 0.393 |
| | Avg | 0.474 | 0.467 | 0.474 | 0.459 | 0.484 | 0.441 | 0.309 | 0.360 | 0.295 | 0.313 | 0.325 | 0.398 | 0.623 | 0.318 | 0.359 | 0.345 |
| TimesNet | 96 | 0.419 | 0.421 | 0.313 | 0.353 | 0.355 | 0.378 | 0.191 | 0.268 | 0.171 | 0.213 | 0.112 | 0.239 | 0.684 | 0.351 | 0.310 | 0.312 |
| | 192 | 0.465 | 0.445 | 0.391 | 0.398 | 0.385 | 0.393 | 0.248 | 0.303 | 0.229 | 0.262 | 0.201 | 0.325 | 0.689 | 0.359 | 0.408 | 0.378 |
| | 336 | 0.503 | 0.468 | 0.444 | 0.438 | 0.412 | 0.410 | 0.322 | 0.348 | 0.288 | 0.306 | 0.383 | 0.445 | 0.697 | 0.361 | 0.375 | 0.339 |
| | 720 | 0.500 | 0.480 | 0.436 | 0.445 | 0.521 | 0.464 | 0.422 | 0.403 | 0.361 | 0.354 | 1.045 | 0.783 | 0.732 | 0.370 | 0.434 | 0.385 |
| | Avg | 0.472 | 0.453 | 0.396 | 0.409 | 0.418 | 0.411 | 0.296 | 0.331 | 0.262 | 0.284 | 0.435 | 0.448 | 0.700 | 0.360 | 0.382 | 0.354 |
| FEDformer | 96 | 0.392 | 0.415 | 0.329 | 0.371 | 0.646 | 0.536 | 0.251 | 0.336 | 0.226 | 0.287 | 0.125 | 0.256 | 0.653 | 0.388 | 0.432 | 0.473 |
| | 192 | 0.449 | 0.448 | 0.408 | 0.420 | 0.894 | 0.613 | 0.299 | 0.358 | 0.272 | 0.321 | 0.255 | 0.361 | 0.659 | 0.387 | 0.649 | 0.602 |
| | 336 | 0.486 | 0.470 | 0.444 | 0.451 | 0.764 | 0.593 | 0.375 | 0.407 | 0.345 | 0.372 | 0.477 | 0.506 | 0.655 | 0.380 | 0.969 | 0.803 |
| | 720 | 0.540 | 0.503 | 0.451 | 0.461 | 0.853 | 0.622 | 0.468 | 0.453 | 0.414 | 0.405 | 1.261 | 0.865 | 0.695 | 0.402 | 0.577 | 0.549 |
| | Avg | 0.467 | 0.459 | 0.408 | 0.426 | 0.789 | 0.591 | 0.348 | 0.388 | 0.314 | 0.346 | 0.530 | 0.497 | 0.666 | 0.389 | 0.657 | 0.607 |
| FreTS | 96 | 0.733 | 0.513 | 0.420 | 0.416 | 0.800 | 0.522 | 0.389 | 0.384 | 0.280 | 0.318 | 0.131 | 0.264 | 0.883 | 0.428 | 2.009 | 0.650 |
| | 192 | 0.667 | 0.517 | 0.440 | 0.430 | 0.732 | 0.506 | 0.377 | 0.388 | 0.284 | 0.317 | 0.218 | 0.347 | 0.868 | 0.421 | 1.025 | 0.557 |
| | 336 | 0.637 | 0.520 | 0.467 | 0.459 | 0.764 | 0.528 | 0.382 | 0.399 | 0.317 | 0.339 | 0.436 | 0.496 | 0.692 | 0.377 | 1.298 | 0.589 |
| | 720 | 0.672 | 0.577 | 0.977 | 0.664 | 0.683 | 0.522 | 0.466 | 0.446 | 0.359 | 0.368 | 0.908 | 0.715 | 0.725 | 0.393 | 1.146 | 0.548 |
| | Avg | 0.677 | 0.532 | 0.576 | 0.492 | 0.745 | 0.519 | 0.404 | 0.404 | 0.310 | 0.336 | 0.423 | 0.456 | 0.792 | 0.405 | 1.369 | 0.586 |
| TimeKAN | 96 | 0.485 | 0.448 | 0.321 | 0.363 | 0.580 | 0.467 | 0.221 | 0.298 | 0.203 | 0.245 | 0.112 | 0.243 | 0.720 | 0.403 | 0.780 | 0.486 |
| | 192 | 0.484 | 0.453 | 0.417 | 0.421 | 0.489 | 0.430 | 0.312 | 0.356 | 0.288 | 0.304 | 0.275 | 0.384 | 0.650 | 0.365 | 1.175 | 0.586 |
| | 336 | 0.514 | 0.462 | 0.451 | 0.447 | 0.517 | 0.451 | 0.341 | 0.368 | 0.293 | 0.310 | 0.507 | 0.518 | 0.662 | 0.370 | 0.816 | 0.490 |
| | 720 | 0.522 | 0.482 | 0.428 | 0.441 | 0.498 | 0.452 | 0.415 | 0.405 | 0.360 | 0.351 | 0.911 | 0.723 | 0.681 | 0.380 | 0.440 | 0.366 |
| | Avg | 0.501 | 0.461 | 0.404 | 0.418 | 0.521 | 0.450 | 0.323 | 0.357 | 0.286 | 0.302 | 0.451 | 0.467 | 0.678 | 0.379 | 0.803 | 0.482 |
| FiLM | 96 | 0.472 | 0.431 | 0.307 | 0.350 | 0.609 | 0.463 | 0.238 | 0.306 | 0.247 | 0.262 | 0.115 | 0.248 | 0.905 | 0.521 | 1.932 | 0.747 |
| | 192 | 0.486 | 0.444 | 0.387 | 0.397 | 0.460 | 0.415 | 0.281 | 0.329 | 0.274 | 0.286 | 0.181 | 0.303 | 1.335 | 0.741 | 1.347 | 0.673 |
| | 336 | 0.510 | 0.458 | 0.425 | 0.430 | 0.467 | 0.425 | 0.320 | 0.349 | 0.314 | 0.315 | 0.322 | 0.409 | 1.413 | 0.775 | 0.820 | 0.569 |
| | 720 | 0.500 | 0.478 | 0.435 | 0.446 | 0.509 | 0.450 | 0.421 | 0.406 | 0.378 | 0.356 | 0.901 | 0.722 | 1.470 | 0.794 | 0.595 | 0.526 |
| | Avg | 0.492 | 0.453 | 0.389 | 0.406 | 0.511 | 0.438 | 0.315 | 0.348 | 0.303 | 0.305 | 0.380 | 0.421 | 1.281 | 0.708 | 1.173 | 0.629 |
| RobustTSF | 96 | 0.654 | 0.501 | 0.413 | 0.424 | 0.920 | 0.539 | 0.306 | 0.351 | 0.504 | 0.365 | 0.127 | 0.261 | 1.027 | 0.482 | 2.459 | 0.818 |
| | 192 | 0.613 | 0.506 | 0.459 | 0.454 | 0.725 | 0.496 | 0.329 | 0.367 | 0.477 | 0.373 | 0.193 | 0.329 | 0.937 | 0.446 | 2.280 | 0.816 |
| | 336 | 0.591 | 0.505 | 0.507 | 0.489 | 0.673 | 0.497 | 0.366 | 0.394 | 0.452 | 0.383 | 0.297 | 0.412 | 0.856 | 0.425 | 1.961 | 0.751 |
| | 720 | 0.574 | 0.528 | 0.634 | 0.564 | 0.637 | 0.505 | 0.446 | 0.444 | 0.450 | 0.407 | 1.026 | 0.741 | 0.885 | 0.447 | 1.583 | 0.658 |
| | Avg | 0.608 | 0.510 | 0.503 | 0.483 | 0.739 | 0.509 | 0.362 | 0.389 | 0.471 | 0.382 | 0.411 | 0.435 | 0.926 | 0.450 | 2.071 | 0.761 |
| TameR | 96 | 0.421 | 0.416 | 0.303 | 0.348 | 0.366 | 0.377 | 0.182 | 0.260 | 0.187 | 0.231 | 0.183 | 0.308 | 0.503 | 0.306 | 0.356 | 0.328 |
| | 192 | 0.453 | 0.436 | 0.374 | 0.390 | 0.401 | 0.393 | 0.240 | 0.296 | 0.229 | 0.265 | 0.238 | 0.350 | 0.510 | 0.309 | 0.315 | 0.299 |
| | 336 | 0.488 | 0.450 | 0.413 | 0.424 | 0.424 | 0.410 | 0.295 | 0.330 | 0.280 | 0.300 | 0.461 | 0.505 | 0.523 | 0.319 | 0.338 | 0.307 |
| | 720 | 0.485 | 0.467 | 0.425 | 0.441 | 0.473 | 0.435 | 0.394 | 0.389 | 0.355 | 0.347 | 0.803 | 0.677 | 0.664 | 0.344 | 0.349 | 0.297 |
| | Avg | 0.461 | 0.442 | 0.379 | 0.401 | 0.416 | 0.404 | 0.278 | 0.319 | 0.263 | 0.286 | 0.421 | 0.460 | 0.550 | 0.319 | 0.340 | 0.308 |

*Table 14.* Detailed results of multivariate time series forecasting task on unperturbed data.

| Models | F | ETTh1 | | ETTh2 | | ETTm1 | | ETTm2 | | Weather | | Exchange | | Traffic | | Solar | |
|---|---|---|---|---|---|---|---|---|---|---|---|---|---|---|---|---|---|
| | | MSE | MAE | MSE | MAE | MSE | MAE | MSE | MAE | MSE | MAE | MSE | MAE | MSE | MAE | MSE | MAE |
| DLinear | 96 | 0.379 | 0.386 | 0.293 | 0.343 | 0.331 | 0.350 | 0.183 | 0.258 | 0.208 | 0.233 | 0.098 | 0.221 | 0.688 | 0.364 | 0.280 | 0.289 |
| | 192 | 0.429 | 0.416 | 0.384 | 0.404 | 0.376 | 0.372 | 0.245 | 0.302 | 0.244 | 0.268 | 0.158 | 0.288 | 0.646 | 0.340 | 0.312 | 0.306 |
| | 336 | 0.472 | 0.441 | 0.460 | 0.458 | 0.407 | 0.394 | 0.307 | 0.348 | 0.287 | 0.306 | 0.263 | 0.382 | 0.649 | 0.343 | 0.356 | 0.322 |
| | 720 | 0.490 | 0.490 | 0.673 | 0.577 | 0.469 | 0.432 | 0.409 | 0.413 | 0.346 | 0.355 | 0.755 | 0.656 | 0.678 | 0.362 | 0.366 | 0.316 |
| | Avg | 0.443 | 0.433 | 0.452 | 0.446 | 0.396 | 0.387 | 0.286 | 0.330 | 0.271 | 0.290 | 0.318 | 0.387 | 0.665 | 0.352 | 0.328 | 0.308 |
| TimeMixer | 96 | 0.382 | 0.389 | 0.287 | 0.331 | 0.326 | 0.352 | 0.175 | 0.253 | 0.167 | 0.201 | 0.095 | 0.214 | 0.469 | 0.289 | 0.274 | 0.290 |
| | 192 | 0.441 | 0.422 | 0.374 | 0.390 | 0.374 | 0.372 | 0.239 | 0.296 | 0.215 | 0.246 | 0.186 | 0.307 | 0.485 | 0.299 | 0.378 | 0.343 |
| | 336 | 0.496 | 0.460 | 0.413 | 0.421 | 0.411 | 0.393 | 0.300 | 0.334 | 0.276 | 0.290 | 0.329 | 0.414 | 0.496 | 0.304 | 0.356 | 0.335 |
| | 720 | 0.523 | 0.484 | 0.420 | 0.437 | 0.481 | 0.435 | 0.399 | 0.392 | 0.358 | 0.344 | 0.913 | 0.723 | 0.532 | 0.325 | 0.327 | 0.317 |
| | Avg | 0.461 | 0.439 | 0.374 | 0.395 | 0.398 | 0.388 | 0.278 | 0.319 | 0.254 | 0.270 | 0.381 | 0.414 | 0.495 | 0.304 | 0.334 | 0.321 |
| CycleNet | 96 | **0.374** | **0.382** | 0.287 | 0.332 | 0.319 | 0.346 | 0.167 | 0.245 | 0.176 | 0.212 | 0.084 | 0.202 | 0.471 | 0.283 | 0.258 | 0.253 |
| | 192 | 0.422 | **0.410** | 0.373 | 0.385 | 0.367 | 0.369 | 0.233 | 0.288 | 0.225 | 0.255 | 0.190 | 0.308 | 0.480 | 0.286 | 0.282 | 0.262 |
| | 336 | 0.461 | **0.430** | 0.416 | 0.422 | 0.397 | 0.391 | 0.294 | 0.328 | 0.279 | 0.294 | 0.372 | 0.441 | 0.491 | 0.292 | 0.309 | 0.273 |
| | 720 | **0.461** | **0.449** | 0.424 | 0.438 | 0.460 | 0.425 | 0.396 | 0.387 | 0.352 | 0.344 | 0.915 | 0.722 | 0.516 | 0.308 | 0.319 | 0.277 |
| | Avg | **0.429** | **0.418** | 0.375 | 0.394 | 0.386 | 0.383 | 0.273 | 0.312 | 0.258 | 0.276 | 0.390 | 0.419 | 0.489 | 0.292 | 0.292 | 0.266 |
| Crossformer | 96 | 0.444 | 0.443 | 0.644 | 0.547 | 0.367 | 0.389 | 0.228 | 0.315 | **0.150** | **0.189** | 0.172 | 0.304 | 0.519 | 0.261 | 0.196 | **0.185** |
| | 192 | 0.455 | 0.450 | 0.758 | 0.600 | 0.391 | 0.399 | 0.301 | 0.376 | **0.201** | 0.246 | 0.373 | 0.463 | 0.611 | 0.325 | 0.222 | **0.207** |
| | 336 | 0.522 | 0.489 | 1.039 | 0.713 | 0.428 | 0.420 | 0.631 | 0.545 | 0.274 | 0.301 | 0.699 | 0.665 | 0.581 | 0.270 | 0.258 | 0.251 |
| | 720 | 0.651 | 0.610 | 1.464 | 0.892 | 0.748 | 0.571 | 1.057 | 0.710 | 0.365 | 0.365 | 1.182 | 0.907 | 0.661 | 0.345 | 0.229 | 0.227 |
| | Avg | 0.518 | 0.498 | 0.976 | 0.688 | 0.484 | 0.445 | 0.554 | 0.486 | 0.247 | 0.275 | 0.606 | 0.585 | 0.593 | 0.300 | 0.226 | **0.218** |
| PatchTST | 96 | 0.375 | 0.387 | **0.284** | **0.329** | 0.310 | 0.340 | 0.174 | 0.251 | 0.174 | 0.210 | 0.084 | 0.200 | 0.473 | 0.283 | 0.211 | 0.243 |
| | 192 | 0.423 | 0.416 | **0.362** | **0.380** | 0.363 | 0.372 | 0.239 | 0.294 | 0.221 | 0.249 | 0.184 | 0.304 | 0.476 | 0.284 | 0.269 | 0.316 |
| | 336 | 0.467 | 0.436 | 0.408 | 0.416 | 0.400 | 0.393 | 0.299 | 0.333 | 0.276 | 0.289 | 0.324 | 0.412 | 0.497 | 0.292 | 0.324 | 0.337 |
| | 720 | 0.461 | 0.455 | 0.414 | 0.432 | 0.459 | 0.429 | 0.400 | 0.391 | 0.352 | 0.339 | 0.901 | 0.719 | 0.533 | 0.309 | 0.283 | 0.322 |
| | Avg | 0.431 | 0.424 | **0.367** | **0.389** | 0.383 | 0.383 | 0.278 | 0.318 | 0.256 | 0.272 | 0.373 | 0.409 | 0.495 | 0.292 | 0.272 | 0.305 |
| iTransformer | 96 | 0.383 | 0.397 | 0.296 | 0.341 | 0.317 | 0.346 | 0.175 | 0.252 | 0.168 | 0.202 | 0.083 | 0.202 | **0.421** | 0.270 | 0.189 | 0.213 |
| | 192 | 0.443 | 0.432 | 0.381 | 0.394 | 0.371 | 0.376 | 0.243 | 0.298 | 0.219 | 0.248 | 0.179 | 0.300 | **0.441** | 0.278 | 0.257 | 0.246 |
| | 336 | 0.476 | 0.451 | 0.422 | 0.428 | 0.409 | 0.398 | 0.301 | 0.335 | 0.277 | 0.291 | 0.343 | 0.423 | **0.458** | 0.284 | 0.248 | 0.246 |
| | 720 | 0.490 | 0.476 | 0.424 | 0.440 | 0.477 | 0.437 | 0.405 | 0.397 | 0.354 | 0.342 | 0.855 | 0.699 | **0.481** | **0.296** | 0.302 | 0.275 |
| | Avg | 0.448 | 0.439 | 0.381 | 0.401 | 0.394 | 0.389 | 0.281 | 0.321 | 0.254 | 0.271 | 0.365 | 0.406 | **0.450** | 0.282 | 0.249 | 0.245 |
| Leddam | 96 | 0.378 | 0.387 | 0.291 | 0.333 | 0.317 | 0.342 | 0.171 | 0.249 | 0.157 | 0.203 | 0.084 | 0.204 | 0.463 | **0.260** | **0.187** | 0.233 |
| | 192 | 0.428 | 0.417 | 0.366 | 0.381 | 0.367 | 0.367 | 0.236 | 0.291 | 0.207 | 0.250 | 0.176 | 0.298 | 0.477 | **0.273** | 0.216 | 0.253 |
| | 336 | 0.467 | 0.435 | 0.415 | 0.421 | 0.404 | 0.392 | 0.298 | 0.331 | 0.263 | 0.291 | 0.346 | 0.423 | 0.497 | 0.280 | 0.234 | 0.265 |
| | 720 | 0.474 | 0.462 | **0.413** | **0.430** | 0.472 | 0.432 | 0.404 | 0.395 | 0.343 | 0.345 | 0.841 | 0.689 | 0.527 | 0.300 | 0.241 | 0.273 |
| | Avg | 0.437 | 0.425 | 0.371 | 0.391 | 0.390 | 0.383 | 0.277 | 0.317 | 0.243 | 0.272 | 0.362 | 0.403 | 0.491 | **0.278** | **0.220** | 0.256 |
| MICN | 96 | 0.388 | 0.402 | 0.298 | 0.351 | **0.307** | 0.341 | 0.175 | 0.263 | 0.187 | 0.225 | 0.089 | 0.209 | 0.575 | 0.296 | 0.196 | 0.229 |
| | 192 | **0.413** | 0.423 | 0.379 | 0.402 | 0.362 | 0.373 | 0.242 | 0.310 | 0.230 | 0.262 | **0.151** | **0.280** | 0.595 | 0.300 | 0.244 | 0.249 |
| | 336 | **0.448** | 0.444 | 0.464 | 0.460 | 0.396 | 0.399 | 0.302 | 0.349 | 0.262 | 0.289 | **0.246** | **0.374** | 0.622 | 0.311 | 0.321 | 0.293 |
| | 720 | 0.493 | 0.494 | 0.710 | 0.589 | 0.458 | 0.439 | 0.419 | 0.426 | **0.312** | **0.322** | 0.751 | 0.654 | 0.684 | 0.336 | 0.372 | 0.347 |
| | Avg | 0.435 | 0.441 | 0.463 | 0.450 | 0.381 | 0.388 | 0.285 | 0.337 | 0.248 | 0.274 | **0.309** | **0.379** | 0.619 | 0.311 | 0.283 | 0.279 |
| TimesNet | 96 | 0.419 | 0.420 | 0.319 | 0.356 | 0.336 | 0.364 | 0.180 | 0.256 | 0.162 | 0.204 | 0.108 | 0.234 | 0.662 | 0.338 | 0.292 | 0.295 |
| | 192 | 0.451 | 0.437 | 0.412 | 0.411 | 0.376 | 0.387 | 0.244 | 0.300 | 0.223 | 0.256 | 0.195 | 0.320 | 0.669 | 0.347 | 0.385 | 0.357 |
| | 336 | 0.518 | 0.481 | 0.456 | 0.441 | 0.404 | 0.405 | 0.313 | 0.341 | 0.283 | 0.301 | 0.379 | 0.442 | 0.674 | 0.348 | 0.359 | 0.322 |
| | 720 | 0.572 | 0.520 | 0.480 | 0.466 | 0.506 | 0.457 | 0.419 | 0.401 | 0.357 | 0.351 | 1.037 | 0.779 | 0.711 | 0.360 | 0.421 | 0.371 |
| | Avg | 0.490 | 0.464 | 0.417 | 0.419 | 0.406 | 0.403 | 0.289 | 0.325 | 0.256 | 0.278 | 0.430 | 0.444 | 0.679 | 0.348 | 0.364 | 0.336 |
| FEDformer | 96 | 0.377 | 0.404 | 0.328 | 0.370 | 0.450 | 0.462 | 0.227 | 0.317 | 0.210 | 0.272 | 0.124 | 0.255 | 0.637 | 0.376 | 0.331 | 0.409 |
| | 192 | 0.427 | 0.434 | 0.407 | 0.419 | 0.562 | 0.518 | 0.295 | 0.355 | 0.258 | 0.307 | 0.255 | 0.361 | 0.637 | 0.369 | 0.442 | 0.502 |
| | 336 | 0.463 | 0.458 | 0.482 | 0.476 | 0.607 | 0.536 | 0.375 | 0.406 | 0.347 | 0.375 | 0.477 | 0.505 | 0.648 | 0.373 | 0.995 | 0.826 |
| | 720 | 0.492 | 0.486 | 0.452 | 0.460 | 0.658 | 0.566 | 0.468 | 0.453 | 0.410 | 0.401 | 1.261 | 0.865 | 0.675 | 0.388 | 0.424 | 0.460 |
| | Avg | 0.439 | 0.446 | 0.417 | 0.431 | 0.569 | 0.520 | 0.341 | 0.383 | 0.306 | 0.339 | 0.529 | 0.497 | 0.649 | 0.376 | 0.548 | 0.549 |
| FreTS | 96 | 0.408 | 0.401 | 0.308 | 0.347 | 0.330 | 0.353 | 0.183 | 0.261 | 0.167 | 0.204 | 0.086 | 0.207 | 0.500 | 0.297 | 0.230 | 0.247 |
| | 192 | 0.460 | 0.431 | 0.388 | 0.399 | 0.407 | 0.394 | 0.258 | 0.312 | 0.207 | **0.243** | 0.164 | 0.295 | 0.498 | 0.296 | 0.260 | 0.263 |
| | 336 | 0.534 | 0.467 | 0.508 | 0.480 | 0.429 | 0.415 | 0.329 | 0.364 | **0.259** | 0.283 | 0.427 | 0.489 | 0.515 | 0.302 | 0.279 | 0.270 |
| | 720 | 0.643 | 0.551 | 0.961 | 0.651 | 0.493 | 0.453 | 0.429 | 0.423 | 0.329 | 0.336 | 0.899 | 0.710 | 0.562 | 0.323 | 0.282 | 0.265 |
| | Avg | 0.511 | 0.462 | 0.541 | 0.469 | 0.415 | 0.404 | 0.300 | 0.340 | **0.240** | **0.267** | 0.394 | 0.425 | 0.519 | 0.304 | 0.263 | 0.261 |
| TimeKAN | 96 | 0.383 | 0.386 | 0.293 | 0.336 | 0.316 | 0.347 | 0.178 | 0.256 | 0.170 | 0.208 | 0.085 | 0.204 | 0.654 | 0.360 | 0.262 | 0.260 |
| | 192 | 0.443 | 0.426 | 0.375 | 0.388 | 0.379 | 0.371 | 0.240 | 0.296 | 0.221 | 0.252 | 0.188 | 0.307 | 0.617 | 0.341 | 0.289 | 0.277 |
| | 336 | 0.465 | 0.432 | 0.417 | 0.424 | 0.399 | 0.392 | 0.301 | 0.334 | 0.264 | **0.283** | 0.395 | 0.455 | 0.613 | 0.340 | 0.351 | 0.305 |
| | 720 | 0.491 | 0.465 | 0.426 | 0.439 | 0.459 | 0.427 | 0.405 | 0.397 | 0.343 | 0.335 | 0.909 | 0.721 | 0.649 | 0.357 | 0.384 | 0.322 |
| | Avg | 0.445 | 0.427 | 0.378 | 0.397 | 0.388 | 0.384 | 0.281 | 0.321 | 0.249 | 0.269 | 0.394 | 0.422 | 0.633 | 0.349 | 0.321 | 0.291 |
| FiLM | 96 | 0.382 | 0.386 | 0.296 | 0.340 | 0.334 | 0.350 | 0.183 | 0.258 | 0.208 | 0.231 | 0.115 | 0.248 | 0.811 | 0.477 | 0.298 | 0.291 |
| | 192 | 0.435 | 0.417 | 0.385 | 0.395 | 0.380 | 0.373 | 0.247 | 0.299 | 0.250 | 0.266 | 0.180 | 0.302 | 1.281 | 0.717 | 0.337 | 0.324 |
| | 336 | 0.478 | 0.439 | 0.421 | 0.426 | 0.412 | 0.394 | 0.308 | 0.338 | 0.299 | 0.302 | 0.321 | 0.409 | 1.386 | 0.765 | 0.383 | 0.368 |
| | 720 | 0.481 | 0.468 | 0.427 | 0.440 | 0.477 | 0.430 | 0.408 | 0.394 | 0.370 | 0.348 | 0.886 | 0.711 | 1.455 | 0.789 | 0.491 | 0.466 |
| | Avg | 0.444 | 0.427 | 0.382 | 0.400 | 0.401 | 0.387 | 0.286 | 0.322 | 0.282 | 0.287 | 0.376 | 0.417 | 1.233 | 0.687 | 0.377 | 0.362 |
| RobustTSF | 96 | 0.378 | 0.386 | 0.292 | 0.343 | 0.331 | 0.352 | 0.180 | 0.259 | 0.206 | 0.233 | **0.078** | **0.196** | 0.666 | 0.371 | 0.294 | 0.324 |
| | 192 | 0.440 | 0.429 | 0.384 | 0.405 | 0.375 | 0.374 | 0.243 | 0.303 | 0.242 | 0.269 | **0.149** | 0.280 | 0.616 | 0.347 | 0.326 | 0.343 |
| | 336 | 0.479 | 0.451 | 0.459 | 0.458 | 0.406 | 0.396 | 0.304 | 0.349 | 0.285 | 0.310 | 0.262 | 0.380 | 0.621 | 0.349 | 0.369 | 0.355 |
| | 720 | 0.487 | 0.484 | 0.604 | 0.546 | 0.469 | 0.437 | 0.408 | 0.417 | 0.346 | 0.360 | 1.007 | 0.727 | 0.654 | 0.370 | 0.377 | 0.348 |
| | Avg | 0.446 | 0.438 | 0.435 | 0.438 | 0.395 | 0.390 | 0.284 | 0.332 | 0.270 | 0.293 | 0.374 | 0.396 | 0.639 | 0.359 | 0.342 | 0.343 |
| TameR | 96 | 0.377 | 0.389 | 0.286 | 0.332 | 0.312 | **0.339** | **0.164** | **0.242** | 0.163 | 0.202 | 0.083 | 0.200 | 0.488 | 0.290 | 0.191 | 0.214 |
| | 192 | 0.425 | 0.417 | 0.365 | 0.382 | 0.360 | 0.366 | 0.229 | 0.286 | 0.212 | 0.247 | 0.192 | 0.313 | 0.501 | 0.298 | 0.228 | 0.236 |
| | 336 | 0.462 | 0.433 | **0.406** | 0.418 | 0.392 | 0.388 | 0.287 | 0.323 | 0.268 | 0.286 | 0.343 | 0.422 | 0.514 | 0.308 | 0.272 | 0.262 |
| | 720 | 0.462 | 0.453 | 0.419 | 0.435 | 0.454 | 0.422 | 0.387 | 0.384 | 0.348 | 0.340 | 0.813 | 0.679 | 0.647 | 0.334 | 0.280 | 0.263 |
| | Avg | 0.432 | 0.423 | 0.369 | 0.392 | 0.379 | 0.379 | 0.267 | 0.309 | 0.248 | 0.269 | 0.358 | 0.403 | 0.537 | 0.308 | 0.243 | 0.244 |

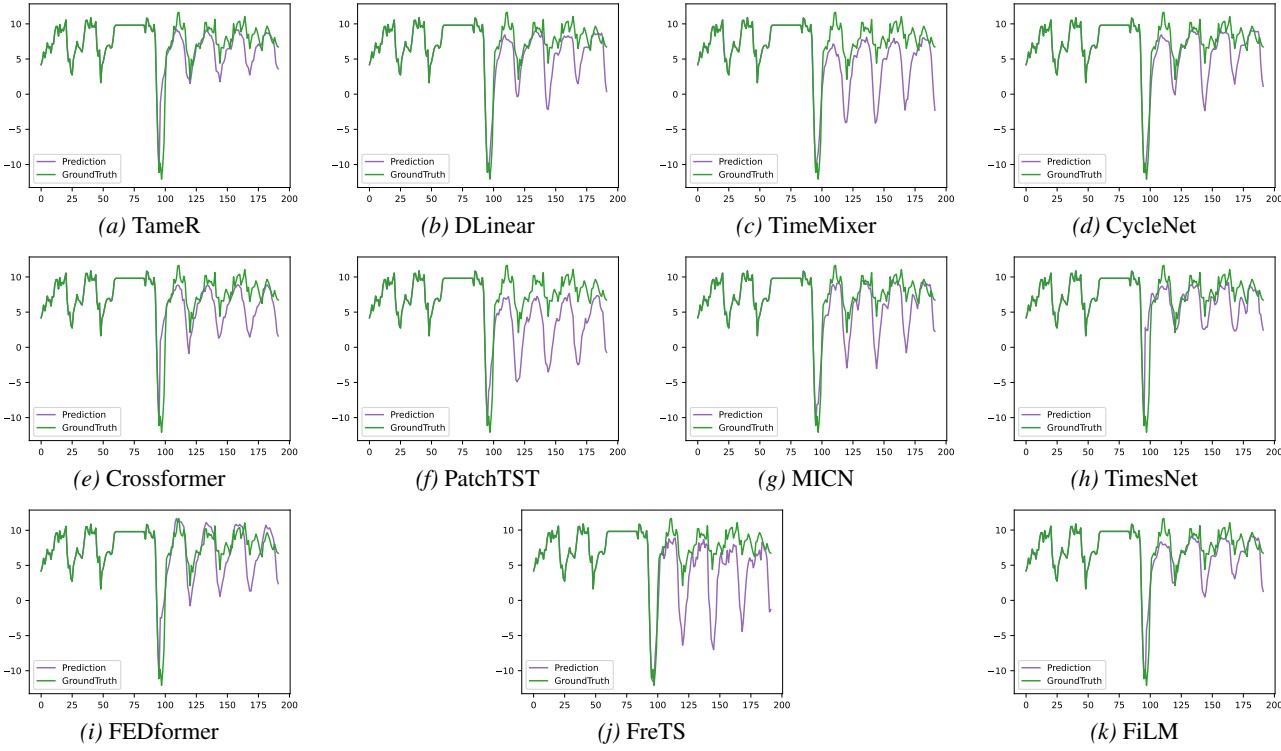

*Figure 10.* Prediction cases from ETTh1 by different models with historical length $H = 96$ and prediction length $F = 96$. Green lines are the ground truths and purple lines are the model predictions.

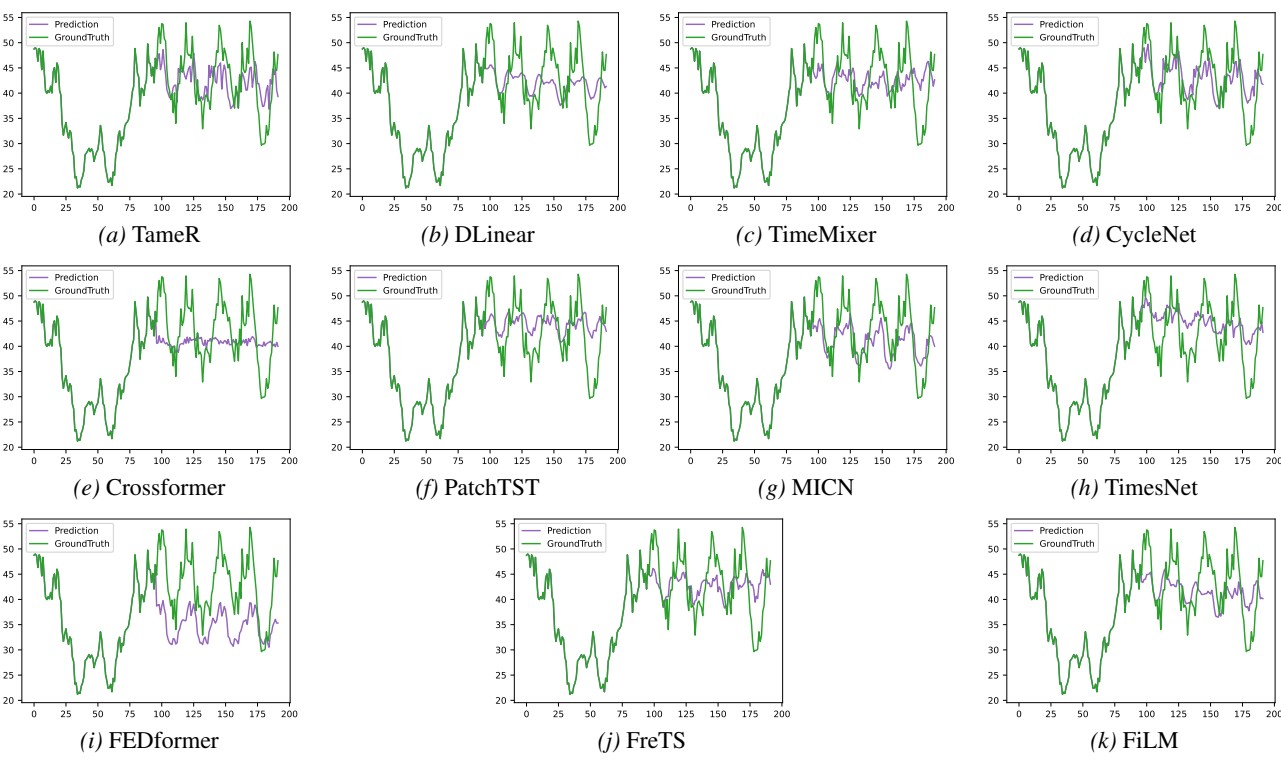

*Figure 11.* Prediction cases from ETTh2 by different models with historical length $H = 96$ and prediction length $F = 96$.

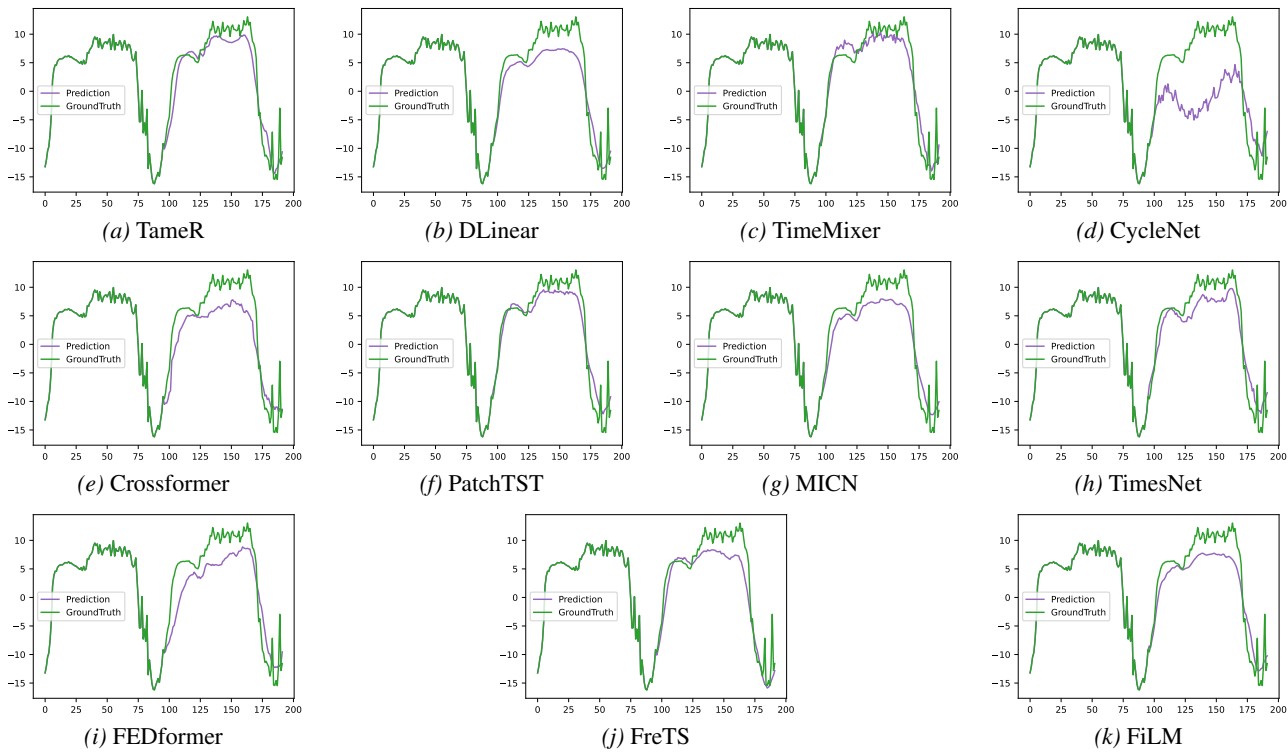

*Figure 12.* Prediction cases from ETTm1 by different models with historical length $H = 96$ and prediction length $F = 96$.

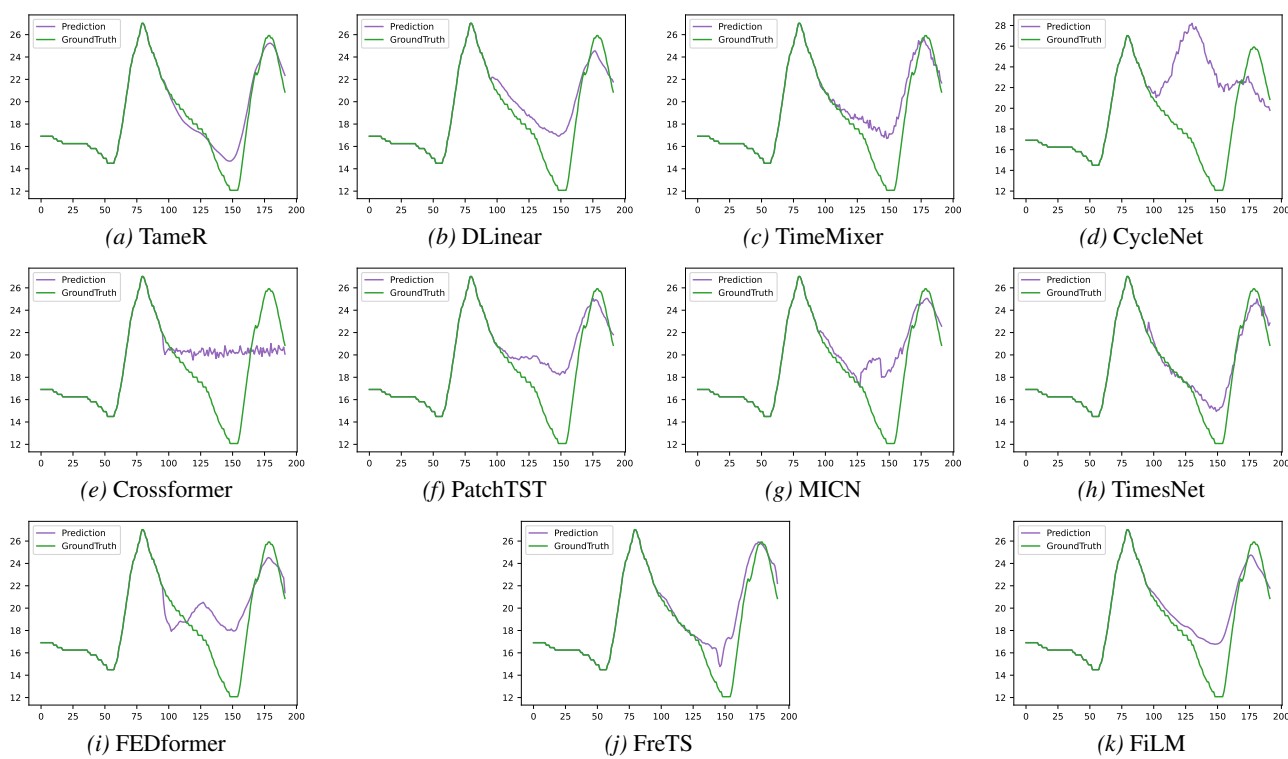

*Figure 13.* Prediction cases from ETTm2 by different models with historical length $H = 96$ and prediction length $F = 96$.

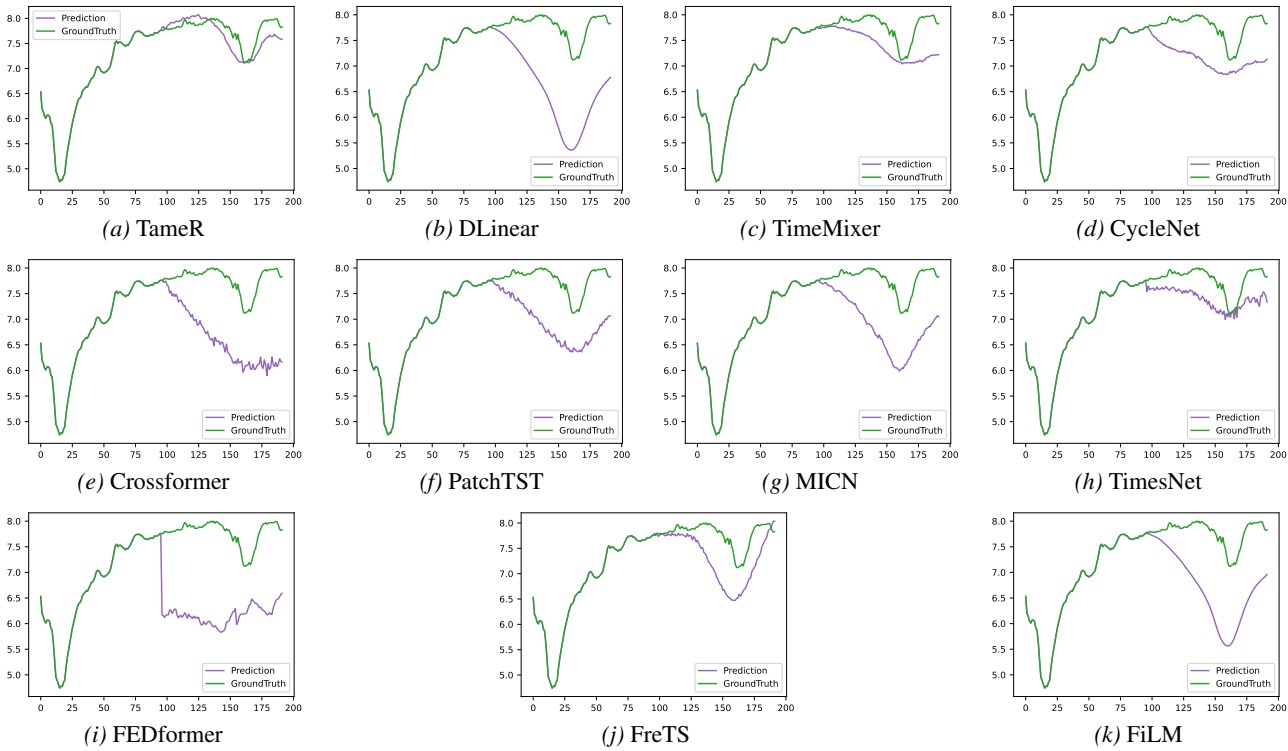

*Figure 14.* Prediction cases from Weather by different models with historical length $H = 96$ and prediction length $F = 96$.

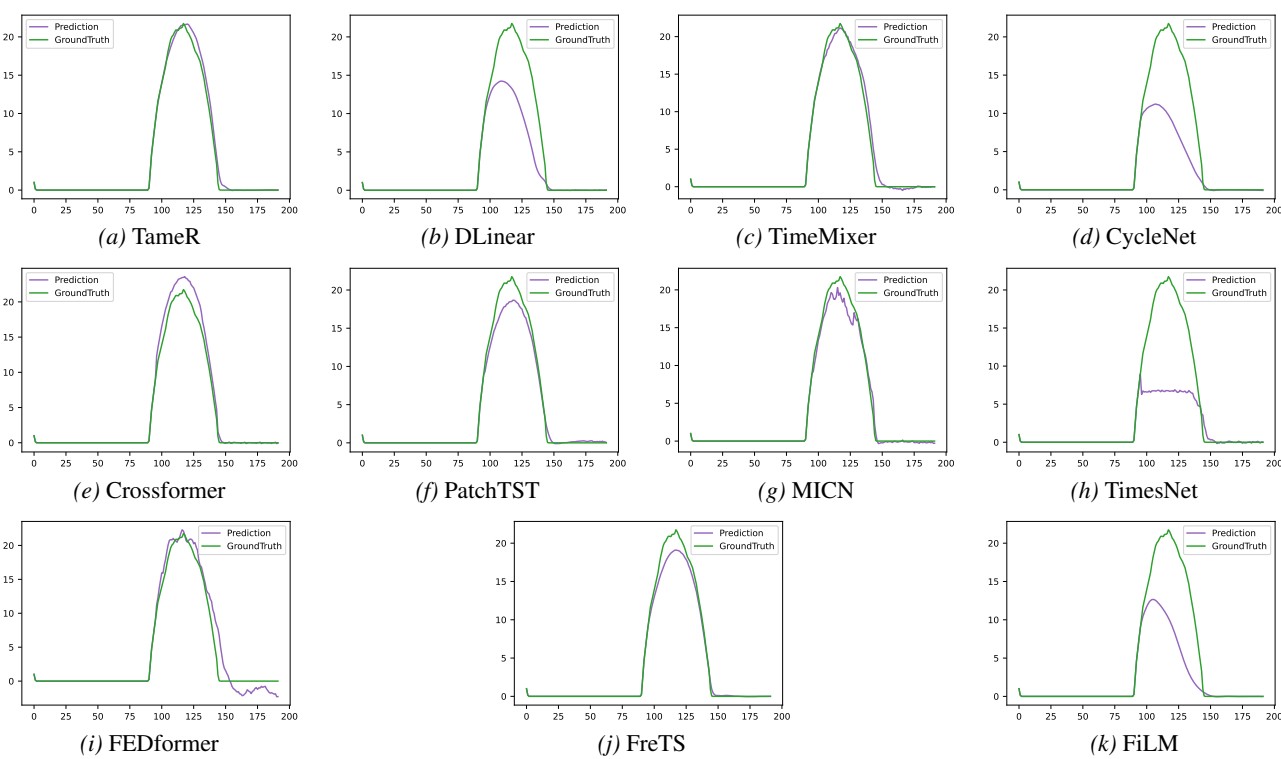

*Figure 15.* Prediction cases from Solar by different models with historical length $H = 96$ and prediction length $F = 96$.

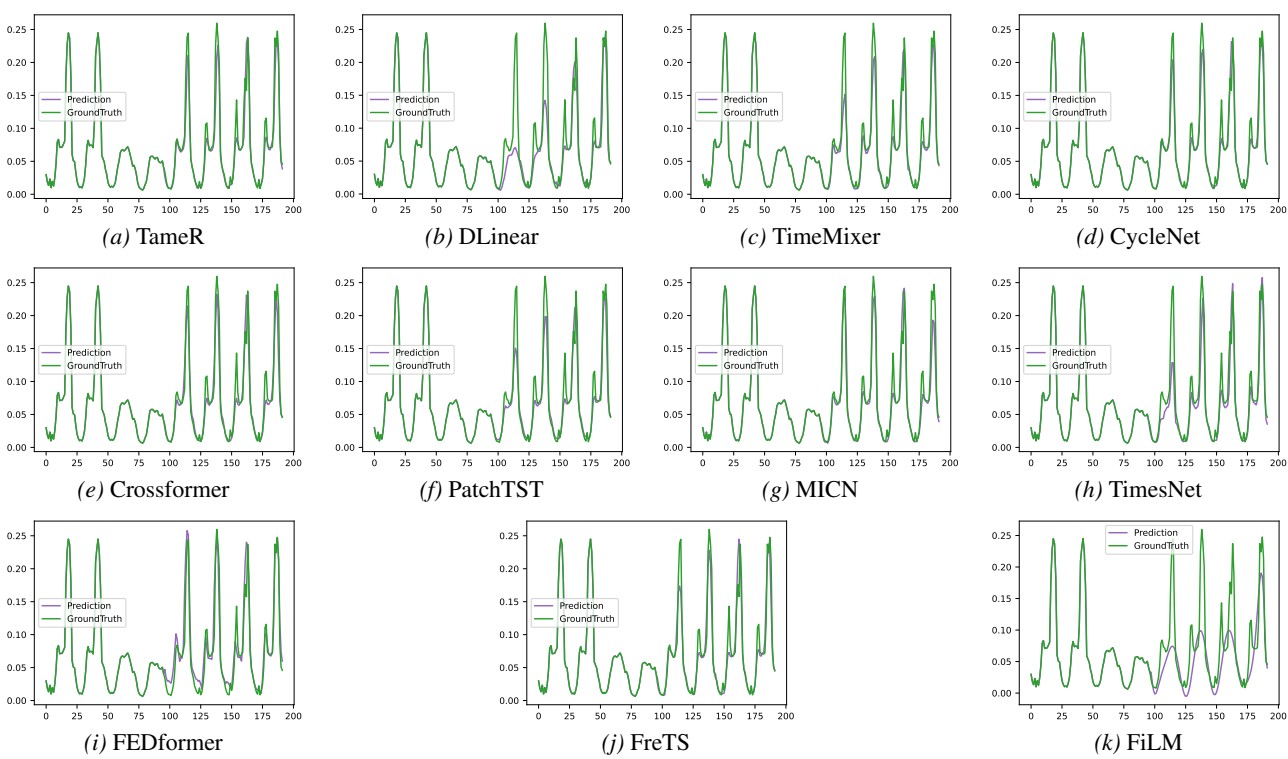

*Figure 16.* Prediction cases from Traffic by different models with historical length $H = 96$ and prediction length $F = 96$.

