# OpenReview forum: "Taming the Recent-Data Bias: Towards Robust Time Series Forecasting with Global Context"
_ICML.cc/2026/Conference — ICML 2026 regular_

### Official Review · Reviewer_xkiW · 2026-03-02

**Soundness:** 2
**Presentation:** 3
**Significance:** 3
**Originality:** 3
**Overall Recommendation:** 3
**Confidence:** 4

**Summary:**

This paper conducts a study on the recent-data bias problem that is prevalent in time series prediction models, pointing out that this bias leads to insufficient robustness of the models when dealing with recent data anomalies. To address this issue, the authors propose TameR, which consists of two core modules: Basis-Aligned Randomized Sampling (BARS) and Learnable Periodicity Extraction (LPE). BARS forces the model to utilize global context rather than local dependencies through random sampling during training. LPE extracts key periodic information before sampling to avoid information loss. Experimental results show that TameR improves prediction robustness in various data perturbation scenarios, especially in the presence of recent anomalies, while maintaining competitive accuracy on clean data.

**Compliance With Llm Reviewing Policy:**

Affirmed.

**Final Justification:**

I would like to keep my initial score.

**Key Questions For Authors:**

1. Can the model still be effective for time series data that lacks strong periodicity?
2. The random sampling rate ρ is a critical hyperparameter. Given that the sampling strategy during inference (using full data) differs from training, does this distribution shift lead to performance loss? Furthermore, Fig. 9a shows a significant drop in robustness at ρ=0.9. Does this imply that the model tends to revert to relying on recent data when information becomes too abundant?
3. The authors mention using MAE as the loss function for robustness. Were the baseline models also retrained using MAE for a fair comparison in the robustness evaluation?

**Limitations:**

The authors have discussed limitations regarding channel independence and fixed sampling rates. However, they could further elaborate on the model's limitations concerning highly non-stationary or non-periodic time series, where the reliance on fixed trigonometric basis functions might be suboptimal.

**Strengths And Weaknesses:**

Strengths
1. The research problem addressed is highly valuable and practically relevant.
2. The authors designed comprehensive perturbation experiments, covering various anomaly types.
3. The results show that TameR outperforms baseline models under specific perturbation scenarios, proving the effectiveness of the method.

Weaknesses
1. While the authors claim competitive accuracy on clean data, Table 2 shows that TameR's performance on many datasets does not surpass strong baselines like PatchTST or iTransformer. Although a trade-off for robustness is understandable, the extent of this sacrifice in accuracy needs to be more objectively scrutinized, especially for high-precision forecasting scenarios.
2. TameR relies heavily on the LPE module and BARS, introducing a strong inductive bias towards periodicity. This may limit its expressiveness for non-stationary data, time series lacking obvious periodicity, or those with severe distribution shifts (e.g., the Exchange dataset), as evidenced by its relatively weaker performance in such cases.
3. The BARS module utilizes fixed Fourier bases (sine/cosine). Why not employ learnable basis functions? Additionally, fixing the frequency set to 179 might be too rigid to cover the diverse characteristics across all datasets, potentially constraining the model's generalization ability.

---

> ### Author Rebuttal · Authors · 2026-03-30
>
> **Thanks for your constructive review!**
>
> **W1: Accuracy on clean data.**
>
> A1: Existing literature shows that **robustness against perturbations and accuracy on clean data represent a trade-off [1]**. Despite this, TameR globally outperforms baselines on clean data, achieving the best Average Rank (1.82) and $1^{st}$ Count (16) (Table 2). The sacrifice in clean-data accuracy is minimal compared to the gains in SOTA robustness.
>
> Besides, TameR offers flexibility to improve clean-data accuracy by adjusting the sampling rate $\rho$. As shown in our parameter sensitivity analysis (Fig9.a), increasing $\rho$ can improve clean-data accuracy, albeit with a decrease in robustness to perturbations.
>
> **W2 & Q1: Effectiveness on non-periodic data (e.g., Exchange).**
>
> A2: **TameR is not limited to periodic data.** For non-periodic datasets, we can explicitly set an extremely large cycle length ($W\gg H$). Thus, the LPE component tends to learn the cycle as a constant value, gracefully degrading TameR to operate on the raw sequence via BARS.
>
> Besides, literature highlights that without periodicity, forecasting is extremely challenging [2, 3]. On the non-periodic and shifted Exchange, almost all advanced models cannot outperform simply copying the last data point [3] (MSE 0.344). Thus, TameR's slightly weaker performance on Exchange is expected, while its overall performance remains competitive.
>
> **W3: Fixed Fourier bases vs. learnable basis functions.**
>
> A3: We carefully design the fixed set of 179 frequencies to explicitly cover the **most common sampling frequencies** in TSF across varing domains (sampling interval: minutely, hourly, daily, weekly). For datasets with other characteristics (e.g., second-level sampling), this frequency range can be easily adjusted to cover higher or lower frequencies.
>
> Learnable basis functions actually degrade both **accuracy** and **efficiency**:
>
> - **Accuracy**: Without physically meaningful frequency constraints, learnable bases tend to fit spurious periods and make optimization less stable, leading to weaker modeling of either short-term fluctuations or long-term trends. Thus, they do not outperform the inductive bias of our fixed frequency set.
> - **Efficiency**: BARS uses a closed-form least-squares solution. Differentiating through a matrix inverse dynamically at every step increases computational costs.
>
> To prove this, we design a learnable variant (TameR-LF) with 256 learnable frequencies and a dynamic gating mechanism to control the activated frequency count. We test it across datasets with different sampling frequencies:
>
> |**Dataset**|**Model**|**MSE**|**Training time (s/iter)**|
> |:-|:-|:-|:-|
> |ETTh2|TameR-LF|0.392|27.4|
> ||TameR|**0.369**|**9.0**|
> |ETTm2|TameR-LF|0.295|121.3|
> ||TameR|**0.267**|**38.8**|
> |Weather|TameR-LF|0.259|278.0|
> ||TameR|**0.248**|**61.7**|
>
> As demonstrated, our fixed frequency set provides optimal, stable capacity, being **more accurate** and 3-4$\times$ **faster** to train.
>
> **Q2: Distribution shift at inference & Robustness drop at $\rho=0.9$.**
>
> A4:
>
> **Distribution Shift**: BARS projects irregular points onto a **continuous** set of basis functions, rather than operating on discrete time steps. Because the representation is continuous, using entire data points during inference simply samples this continuous function perfectly, causing no harmful distribution shift.
>
> **Robustness drop at $\rho=0.9$:** This is correct because high sampling rate leaves too much continuous data intact, allowing the model to easily reconstruct the input. Such phenomenon is consistent with our key idea of utilizing global context via randomized sampling. A moderate rate (e.g., $\rho=0.75$) can force the model to learn to rely on the global context, thus improving robustness. **We can adjust $\rho$ based on data characteristics and practical needs:**
>
> - **Lower $\rho$ (0.5 ~ 0.7):** Recommended when the data involves frequent anomalies, or when the practical need demands extreme robustness. This forces the model to rely heavily on the stable macro-trend.
> - **Higher $\rho$ (0.8 ~ 1.0):** Recommended for relatively clean data, or when higher accuracy is prioritized and some recent-data bias is acceptable.
> - **Moderate $\rho$ (0.7 ~ 0.8):** Serves as a universally default that strikes a balance between accuracy and perturbation robustness for general forecasting tasks.
>
> **Q3: Are baseline models retrained using MAE?**
>
> A5: **Yes.** As detailed in Appendix E.5, **all baselines were retrained using MAE**. To prevent suboptimal performance due to the loss function shift, we validate and adjust hyper-parameters, ensuring all models are optimized for MAE.
>
> **References:**
>
> [1] Chen, et al. Information bottleneck-guided mlps for robust spatial-temporal forecasting. ICML 2025.
>
> [2] Lin, et al. SparseTSF: Modeling Long-term Time Series Forecasting with 1k Parameters. ICML 2024.
>
> [3] Zeng, et al. Are transformers effective for time series forecasting?. AAAI 2023.

---

> > ### Author Rebuttal · Reviewer_xkiW · 2026-04-03
> >
> > My concerns were addressed well, and the additional results helped clarify the paper and strengthen its overall quality.

---

> > > ### Author Response · Authors · 2026-04-03
> > >
> > > Thank you for your feedback. We are encouraged that our response and the additional results have adequately addressed your concerns and strengthened the paper. In light of this, we hope you would consider re-evaluating the current score of our paper. Thank you again for your time and effort in reviewing our paper.

---

### Official Review · Reviewer_AkHm · 2026-03-03

**Soundness:** 3
**Presentation:** 2
**Significance:** 2
**Originality:** 3
**Overall Recommendation:** 4
**Confidence:** 5

**Summary:**

This paper investigates a phenomenon in time-series forecasting, named recent-data bias, indicating the forecasting model tends to weight more on recent than older data points to generate the predictions. This paper then proposes TameR, a framework that handles recent-data bias, to mitigate possible cases where the 'recent data' could possibly be missing values or anomalies. Empirical studies demonstrate the effectiveness of TameR when perturbing the time series and comparable performance when handling normal time series.

**Compliance With Llm Reviewing Policy:**

Affirmed.

**Key Questions For Authors:**

Please refer to the weakness.

**Limitations:**

Please refer to the weakness.

**Strengths And Weaknesses:**

Strength:

1. This paper investigates a phenomenon in time-series forecasting, named recent-data bias, indicating the forecasting model tends to weight more on recent than older data points to generate the predictions. The phenomenon sounds reasonable, and is supported by both empirical and theoretical analysis by this work.

2. This paper proposes TameR, a framework that handles recent-data bias, to mitigate possible cases where the 'recent data' could possibly be missing values or anomalies. Empirical studies demonstrate the effectiveness of TameR when perturbing the time series and comparable performance when handling normal time series.

Weakness:

1. The major motivation of TameR is to avoid being affected by the 'bad recent points' during the forecasting process. It sounds very similar to other popular works in handling non-stationary issues in time-series forecasting, such as RevIN, SAN, and more, where the data may also drift and their methods aim to mitigate such effects. These approaches are typically model agnostic. For TameR, it seems it can also work like a model agnostic framework, rather than a model. For example, in Figure 3, replacing the 'backbone' with SOTA forecasting models without modifying any other parts. To this end, I assume the flexibility of TameR can be more broad, and a more fair comparison could be, for example, TameR+DLinear is better than RevIN+DLinear or SAN+DLinear.

These non-stationary methods then are the baselines, where the rationale behind is that handling the anomalies or missing values is also part of handling non-stationarity in time series.

2. For the evaluations with Weather dataset, to my knowledge, Weather dataset itself contains missing values, which are filled with fixed large values, causing spikes in its original time series. It seems a perfect evaluation case even without adding artificial perturbations. However, TameR seems not performing well on Weather either in Table 1 or 2, could the author provide more intuitions on this?

Looking at the showcases in the appendix, is there any failure cases of TameR, such that lead to its overall performance worse than other methods in Table 2. For Figure 11 (a), the horizon time series are very similar to the most recent data, while the older data can significantly drift, in this case, why TameR still works better than these 'biased' models, and not affected by the drifted older historical data, even with more focus on them.

3. For anomalies in time series, one simple yet intuitive approach is through series-decomposition, while the author narratively claims it does not address perturbations in recent data. Could the author explain more on this, with intuitions and some experiments. My intuition is that moving average typically smoothen the data and such perturbations could be somehow mitigated.

4. The author could further optimize Figure 4 to make it a more clear, step-by-step pipeline with arrows. Currently, components look like simply stack together with no sequential orders, making it hard to read and understand.

---

> ### Author Rebuttal · Authors · 2026-03-30
>
> **Thanks for your constructive review!**
>
> **Q1: TameR as a model-agnostic framework vs. non-stationary methods.**
>
> A1: **We do not claim TameR is strictly model-agnostic** since our backbone operates specifically on basis function coefficients. However, it is a great insight that TameR can seamlessly integrate different backbone models suited for the basis domain (e.g., MLP, CNN).
>
> Regarding non-stationary methods, they are complementary to TameR, not mutually exclusive. TameR natively incorporates RevIN by default. RevIN normalizes _value-space_ distribution shifts but preserves relative local shapes (a recent spike remains a spike). Thus, it cannot alter _temporal-space_ recent-data bias mainly caused by model optimization dynamics, but only handles the value-space non-stationarity.
>
> **Q2: TameR's performance on Weather dataset.**
>
> A2: We appreciate your insight on the inherent missing values which are filled by `-9999.0` in Weather dataset. However, our statistical analysis reveals that **these inherent anomalies are too scarce**:
>
> |**Channel**|**Total Points**|**# Missing Values**|**Proportion**|**Continuous Segments**|
> |---|---|---|---|---|
> |wv|52,696|1|0.0019%|1|
> |max. PAR|52,696|30|0.0569%|18|
> |OT|52,696|50|0.0949%|2|
> |**All (21)**|**1,106,616**|**81**|**0.0073%**|**21**|
>
> Across all 21 channels (over 1.1 million points), there are only **81 missing values (0.0073%) and 21 continuous segments**. Due to the extremely small proportion, its impact on the average MSE over the entire dataset is highly limited. Despite this, TameR remains competitive on the raw Weather dataset (ranking 4th in MSE, 2nd in MAE, see in Table 2).
>
> **Q3: Behavior in Fig. 11(a) and failure cases.**
>
> A3:
>
> **Fig. 11(a)**: While it may appear that TameR places greater emphasis on older data for all input time series, this is not the case. TameR does not blindly amplify the importance of older data; rather, it leverages global context to **dynamically assign importance weights** based on each specific input. This mechanism enables TameR to effectively resist anomalies occurring at any position (as shown in Fig. 6b, TameR shows strong robustness to perturbations at **random** positions).
>
> **Failure Cases**: TameR may fail in the event of a **permanent distribution shift (concept drift)**. TameR's BARS module—which relies on global context for perturbation resistance—may misinterpret this persistent shift as a transient local perturbation.
>
> This reflects a trade-off between mitigating recent-data bias and adapting to persistent concept drift. Fortunately, TameR offers flexibility through adjusting sampling rate ($\rho$) in BARS, allowing it to better accommodate varying data characteristics. As indicated by our parameter sensitivity analysis (Fig9.a), increasing $\rho$ can enhance the model’s capacity to handle concept drift and improve accuracy on clean data, albeit with a slight reduction in robustness to local perturbations. **Based on data characteristics and practical needs, we provide a simple yet intuitive guideline for setting $\rho$:**
>
> - **Lower $\rho$ (0.5 ~ 0.7):** Recommended when the data involves frequent anomalies, or when the practical need demands extreme robustness. A lower $\rho$ forces the model to rely mainly on the stable macro-trend.
> - **Higher $\rho$ (0.8 ~ 1.0):** Recommended for relatively clean data or when permanent distribution shifts occur frequently. Here, the practical need prioritizes adaptation to the "new normal" and high-frequency details for acheiving high clean-data accuracy.
> - **Moderate $\rho$ (0.7 ~ 0.8):** Serves as a universally default that strikes a balance between accuracy and perturbation robustness for general forecasting tasks.
>
> **Q4: Simple series-decomposition (moving average) fails on recent perturbations.**
>
> A4: While moving average (MA) acts as a low-pass filter, it fails to address recent-data perturbations due to two structural flaws:
>
> - **Edge Distortion**: MA relies on sliding windows. A massive anomaly at the end of the historical window causes severe "edge distortion" due to required padding [1], abruptly distorting the decomposition and forcing a false trajectory.
> - **Periodicity Contamination**: MA smooths the data but leaves high-frequency anomalies within the remaining periodic component, causing the model to still overfit to recent perturbations.
>
> In fact, if we use MA for decomposition and apply a model to predict the seasonal and residual terms, it conceptually reverts to the **DLinear** architecture. As shown in Table 1, DLinear performs poorly under recent perturbations (e.g., MSE 0.566 on ETTh1), proving that MA decomposition is fundamentally insufficient for robustness.
>
> **Q5: Fig. 4 Optimization.**
>
> A5: We will redesign Fig. 4 into a **clear, step-by-step pipeline with explicit directional arrows** to improve readability.
>
> **References:**
>
> [1] Lin, Shengsheng, et al. "Cyclenet: Enhancing time series forecasting through modeling periodic patterns." NeurIPS 2024.

---

> > ### Author Rebuttal · Reviewer_AkHm · 2026-04-05
> >
> > I appreciate the rebuttal from the author, I currently have no further concerns.

---

> > > ### Author Response · Authors · 2026-04-06
> > >
> > > Thank you for your feedback. We are glad our clarifications fully addressed your concerns. We hope our response further strengthen your positive assessment of our work. We sincerely appreciate your time and constructive review.

---

### Official Review · Reviewer_vx9c · 2026-03-09

**Soundness:** 3
**Presentation:** 3
**Significance:** 4
**Originality:** 3
**Overall Recommendation:** 5
**Confidence:** 4

**Summary:**

This paper introduces TameR, a new method for robust time series forecasting. The authors identify that existing models suffer from "recent-data bias", which makes them weak against anomalies in recent data. To solve this, TameR uses two main modules. First, it uses Learnable Periodicity Extraction (LPE) to separate periodic patterns. Second, it applies Basis-Aligned Randomized Sampling (BARS) on the residual data. BARS randomly samples points and projects them onto basis functions. The proposed method is evaluated across datasets from diverse domains and demonstrate superior robustness than other state-of-the-art baselines.

**Compliance With Llm Reviewing Policy:**

Affirmed.

**Final Justification:**

The author's rebuttal has addressed my concerns. In light of this, I will maintain my original positive rating.

**Key Questions For Authors:**

see weaknesses

**Limitations:**

yes

**Strengths And Weaknesses:**

Strengths:
1. The paper provides both an empirical and theoretical perspectives over the problem of recent-data bias. It clearly explains the failure of existing models under anomalies.
2. The paper proposes an intuitive and simple method to mitigate the reliance of forecasting models on recent data. The algorithm is well supported by rigor theoretical proofs.
3. The paper offers a comprehensive evaluation of the proposed method against baselines. In depth analysis of the components, key parameters, and different perturbation scenarios are investigated.

Weaknesses:
1. The paper currently adopts a pre-defined and fixed set of frequencies as basis functions. It is better to also discuss their effect on the performance of the proposed method.

---

> ### Author Rebuttal · Authors · 2026-03-30
>
> **Thanks for your kind and constructive review!** We agree that the choice of basis frequencies is an important design aspect of BARS. In the revision, we will add a dedicated frequency ablation and discussion (Appendix) to clarify how the predefined frequency set impacts both clean forecasting accuracy and robustness under recent perturbations.
>
> **Q: Discussion on the effect of using a pre-defined and fixed set of frequencies as basis functions.**
>
> A: We conduct additional ablations on (1) different fixed frequency sets and (2) a learnable-frequency variant.
>
> **1. Fixed frequency sets**
>
> Our default basis uses **179 frequencies** across four tiers: minute-level (12), hour-level (92), day-level (24), and week-level (51), i.e., covering periods from **1 minute up to 1 year**. We construct 10 combinations (e.g., "Minute–Day" includes minute- and hour-level frequencies, "Minute–Year" is our default 179-frequency set) and evaluate them on ETTh2 (hourly) and ETTm2 (15-min) under both clean and perturbed (recent, single anomalous point) data.
>
> |Frequency sets|ETTh2 MSE (Clean / Perturbed)|ETTm2 MSE (Clean / Perturbed)|
> |---|---|---|
> |Minute–Hour|0.392 / 0.398|0.286 / 0.291|
> |Minute–Day|0.392 / 0.394|0.271 / 0.281|
> |Minute–Week|$\underline{0.368}$ / $\underline{0.380}$|0.268 / *0.279*|
> |Minute–Year **(Default)**|*0.369* / **0.379**|**0.267** / **0.278**|
> |Hour–Day|0.405 / 0.407|0.278 / 0.294|
> |Hour–Week|0.381 / 0.386|*0.268* / 0.286|
> |Hour–Year|0.379 / 0.385|$\underline{0.268}$ / 0.284|
> |Day–Week|0.373 / 0.410|0.271 / 0.287|
> |Day–Year|**0.368** / 0.429|0.269 / 0.282|
> |Week–Year|0.370 / *0.383*|0.272 / $\underline{0.279}$|
>
> **Observations:**
>
> 1. Removing low-frequency components (e.g., Minute–Hour) tends to hurt clean-data accuracy, likely due to insufficient capacity to represent long-term trends.
> 2. Removing high-frequency components (e.g., Week–Year) tends to increase sensitivity to perturbations. A plausible explanation is that local anomaly has broad spectral support; without enough high-frequency bases, the least-squares fit may compensate by adjusting low-frequency coefficients, which can affect the global reconstruction. This intuition is consistent with our projection analysis (Proposition C.4), and we will add the detailed discussion in the appendix.
>
> Given the goal of avoiding dataset-specific frequency tuning, we use the full 179-frequency set by default, which performs consistently well across typical forecasting datasets (sampled by minutes to weeks). For datasets with much finer sampling intervals (e.g., seconds), this set can be extended straightforwardly.
>
> **2. Fixed vs. learnable frequencies**
>
> We also explore a learnable-frequency variant (TameR-LF) with 256 learnable frequencies and a gating mechanism to control the number of activated frequencies. In our experiments, learnable frequencies degrade accuracy and incur a noticeable computational overhead, since BARS relies on a closed-form least-squares projection and backpropagating through it increases training cost.
>
> |Dataset|Model|MSE (Clean / Perturbed)|Training time (s/iter)|
> |---|---|---|---|
> |ETTh2|TameR-LF|0.392 / 0.401|27.4|
> ||TameR|**0.369 / 0.379**|**9.0**|
> |ETTm2|TameR-LF|0.295 / 0.314|121.3|
> ||TameR|**0.267 / 0.278**|**38.8**|
>
> We will include these additional results in the revised appendix.

---

> > ### Author Rebuttal · Reviewer_vx9c · 2026-04-01
> >
> > The authors’ detailed rebuttal has adequately addressed my concern about the frequency set. In view of this, I will keep my rating.

---

> > > ### Author Response · Authors · 2026-04-01
> > >
> > > Thank you for your feedback on our response. We are glad that our clarifications addressed your concern and sincerely appreciate your time and effort in reviewing our paper.

---

### Official Review · Reviewer_UbZG · 2026-03-13

**Soundness:** 4
**Presentation:** 4
**Significance:** 3
**Originality:** 2
**Overall Recommendation:** 4
**Confidence:** 4

**Summary:**

This paper studies robust time series forecasting under noisy, missing, and anomalous observations. The authors identify a common weakness in existing methods, namely **recent-data bias**, where models rely too heavily on the most recent observations and thus become sensitive to local perturbations. To address this issue,  **TameR** reduces overdependence on recent data by strengthening the use of global context through randomized sampling and periodicity-aware modeling. Experiments show that TameR achieves substantially better robustness under diverse perturbations while maintaining competitive accuracy on clean data.

**Compliance With Llm Reviewing Policy:**

Affirmed.

**Final Justification:**

This paper addresses robust time series forecasting and identifies *recent-data bias* as a key issue. The proposed method leverages randomized sampling and periodicity-aware modeling to enhance the use of global context.

The paper is technically sound, with strong empirical results and clear presentation. My main concerns were about the limited originality (e.g., similarity between BARS and dropout), as well as the justification of robustness and the positioning regarding global context.

The rebuttal **adequately addressed these concerns**, clarifying the distinction from dropout, the mechanism behind robustness, and the role of optimization dynamics. While the novelty remains somewhat limited, the method is well-motivated and empirically effective.

Since my initial rating was already positive, I **maintain my original score**.

**Key Questions For Authors:**

Please refer to the Weaknesses.

**Limitations:**

Yes.

**Strengths And Weaknesses:**

> **Pros**:

1. The problem involved, i.e., recent data bias is interesting, and the empirical results are strong.
2. This paper provides an intuitive theoretical explanation for the recent-data bias phenomenon. Although not particularly deep, the analysis offers a reasonable conceptual motivation for the proposed method.
3. The paper is generally well presented, with a clear structure and coherent organization.

> **Cons**:

1. The methodological novelty appears limited, as the approach mainly integrates existing ideas. In particular, the proposed BARS is essentially similar to dropout, and the resulting irregularity issue is not adequately addressed.
2. Even with a sampling rate of $\rho = 0.75$, recent anomalous points may still be frequently included due to random sampling, meaning the recent-data issue may not be fully mitigated.
3. Transformer architectures are inherently designed to capture global context, although their learned behavior may still exhibit recency bias in practice. In addition, several periodicity-based methods also exploit global dependencies. Therefore, attributing the limitations of existing approaches primarily to a lack of global context may not be entirely accurate.

---

> ### Author Rebuttal · Authors · 2026-03-30
>
> **Thanks for your constructive review!**
>
> **Q1: Methodological novelty (BARS vs. dropout) and the irregularity issue.**
>
> A1: While BARS shares a conceptual similarity with dropout, it solves a **different** problem of recent-data bias which classic dropout cannot solve. Most baselines (e.g., PatchTST, TimeMixer) natively employ standard dropout but **still exhibit severe vulnerability to recent-data perturbations** (see Fig. 6b).
>
> **Difference with dropout**: To clarify this distinction, we summarize their mechanisms below:
>
> |**Strategy**|**Operation space**|**Core objective**|**Robustness to recent-data bias**|
> |:-|:-|:-|:-|
> |**Dropout**|latent feature space (hidden dimensions)|prevent neuron co-adaptation|insufficient (e.g., PatchTST's MSE on ETTm1 increases by 69% under recent perturbations)|
> |**BARS**|raw input space (temporal time-steps)|break temporal optimization bias|highly effective (MSE increases only 10% on ETTm1 under recent perturbations)|
>
> **Irregularity issue**: Dropping raw points destroys uniform temporal intervals. Padding or interpolation suffers from this issue and severely degrades accuracy (in Fig. 7a, Sample-0-Pad increases ETTh1 MSE from 0.432 to 0.516). However, **BARS explicitly solves this via Basis Transformation** (Sec 5.1), mathematically projecting irregular points onto a continuous set of trigonometric basis functions. Thus, BARS yields a unified, regularized representation without distorting data distributions.
>
>
> **Q2: Random sampling ($\rho=0.75$) may still frequently include recent anomalous points.**
>
> A2: You are correct that BARS does not strictly filter out anomalies. However, the robustness of BARS stems from **fully utilizing the global context**, not from anomaly exclusion.
>
> **During training**, random sampling forces the model to dynamically learn the importance of different historical points based on the specific input, rather than defaulting to a high reliance on recent data.
>
> **During inference**, though the input contains anomalous points, the trained model tends to apply low importances to them. Besides, **the basis projection spreads the local perturbation across global coefficients**, shrinking its impact by $O(1/L)$ (Proposition C.4).
>
> As our **empirical study** shows, the importance scores of baselines sharply spike at the last data point (Fig. 2). In contrast, TameR dynamically balances temporal importance across the historical context, resulting in a more evenly distributed importance with a broader range for each time step (Fig. 5).
>
>
> **Q3: Attributing failure to a "lack of global context" is inaccurate.**
>
> A3: We agree that models like transformers and periodicity-aware models possess the *architectural capacity* for global context. However, we highlight a gap between **architectural capacity** and **optimization dynamics**.
>
> **Theoretically**, standard optimization on time series forecasting drives models to favor recent points (Proposition 3.1). Gradients for distant data decay exponentially, forcing the model to rely on stronger recent-data gradients. Thus, **architectural capacity alone cannot overcome optimization dynamics**.
>
> **Empirically**, we have evaluated several models with architectural capacity to capture global context, including attention-based (PatchTST), multi-scale mixing (TimeMixer), and periodicity-based (CycleNet) architectures. Despite their advanced designs, they still suffer from recent-data bias during optimization (Fig. 2), leading to great performance degradation under recent perturbations. We present MSE and MSE$\Uparrow$ on ETTm1 under perturbations (recent, single anomalous point) as an example:
>
> |Model|Mechanism of Global Context|Perturbed MSE|MSE$\Uparrow$|
> |:-|:-|:-|:-|
> |PatchTST|self-attention|0.645|75.5%|
> |TimeMixer|multi-scale mixing|0.534|36.3%|
> |CycleNet|periodicity modeling|0.619|65.0%|
> |TameR|BARS + LPE|**0.416**|**10.3%**|
>
> Therefore, it is accurate to state that existing methods suffer from an inability to effectively **utilize global context due to optimization dynamics**, a fundamental bottleneck that TameR explicitly resolves.

---

> > ### Author Rebuttal · Reviewer_UbZG · 2026-04-02
> >
> > Thank you for the response. It has basically addressed my concerns. Since my original score was already positive, I have decided to keep it unchanged.

---

> > > ### Author Response · Authors · 2026-04-02
> > >
> > > Thank you for your feedback on our response. We are glad that our clarifications addressed your concern and sincerely appreciate your time and effort in reviewing our paper.

---

### Decision · Program_Chairs · 2026-04-30

**Decision:**

Accept (regular)

**Comment:**

Motivation and Problem Scope: This paper addresses robust time series forecasting and identifies recent-data bias as a key issue [Reviewer UbZG], The paper provides both an empirical and theoretical perspectives over the problem of recent-data bias. It clearly explains the failure of existing models under anomalies. [Reviewer vx9c], This paper investigates a phenomenon in time-series forecasting, named recent-data bias, indicating the forecasting model tends to weight more on recent than older data points to generate the predictions. The phenomenon sounds reasonable, and is supported by both empirical and theoretical analysis by this work [Reviewer AkHm], The research problem addressed is highly valuable and practically relevant [Reviewer xkiW]

Innovation/Originality: Reviewer UbZG concerns on limited originality has been adequately addressed by the authors; This paper proposes TameR, a framework that handles recent-data bias, to mitigate possible cases where the 'recent data' could possibly be missing values or anomalies [Reviewer AkHm]; To reviewer AkHm comment that TameR is very similar to other popular works in handling non-stationary issues in time-series forecasting, authors clarify that the TameR is not strictly model agnostic, but can seamlessly integrate different backbone models suited for the basis domain and also that non-stationary methods are complementary to TameR, not mutually exclusive;

Experiments: The paper offers a comprehensive evaluation of the proposed method against baselines [ Reviewer vx9c]; The results show that TameR outperforms baseline models under specific perturbation scenarios, proving the effectiveness of the method [Reviewer xkiW]; the method is well-motivated and empirically effective [Reviewer UbZG]

Ablations: In depth analysis of the components, key parameters, and different perturbation scenarios are investigated [ Reviewer vx9c]; The results show that TameR outperforms baseline models under specific perturbation scenarios, proving the effectiveness of the method [Reviewer xkiW];

To reviewer AkHm comment that TameR is very similar to other popular works in handling non-stationary issues in time-series forecasting, authors clarify that the TameR is not strictly model agnostic, but can seamlessly integrate different backbone models suited for the basis domain and also that non-stationary methods are complementary to TameR, not mutually exclusive

Authors have addressed the reviews comments regarding (i) Accuracy on clean data (ii) Effectiveness on non-periodic data (e.g., Exchange).
 (iii) using learning basis instead of Fourier bases, authors demonstrate empirically that the fixed frequency set provides optimal, stable capacity, being more accurate and 3-4x faster to train. (iv) the limitation of the method’s expressiveness for non-stationary data.